# Toward Engineering AGI: Benchmarking the Engineering Design Capabilities of LLMs

Xingang Guo[1*†], Yaxin Li[1*], Xiangyi Kong[1*], Yilan Jiang[1*], Xiayu Zhao[1*], Zhihua Gong[1*], Yufan Zhang[1*], Daixuan Li[1], Tianle Sang[1], Beixiao Zhu[1], Gregory Jun[1], Yingbing Huang[1], Yiqi Liu[1], Yuqi Xue[1], Rahul Dev Kundu[1], Qi Jian Lim[1], Yizhou Zhao[2], Luke Alexander Granger[1], Mohamed Badr Younis[1], Darioush Keivan[1], Nippun Sabharwal[1], Shreyanka Sinha[1], Prakhar Agarwal[1], Kojo Vandyck[1], Hanlin Mai[1], Zichen Wang[1], Aditya Venkatesh[1], Ayush Barik[1], Jiankun Yang[1], Chongying Yue[1], Jingjie He[1], Libin Wang[1], Licheng Xu[1], Hao Chen[1], Jinwen Wang[1], Liujun Xu[1], Rushabh Shetty[1], Ziheng Guo[1], Dahui Song[1], Manvi Jha[1], Weijie Liang[1], Weiman Yan[1], Bryan Zhang[1], Sahil Bhandary Karnoor[1], Jialiang Zhang[1], Rutva Pandya[1], Xinyi Gong[1], Mithesh Ballae Ganesh[1], Feize Shi[1], Ruiling Xu[1], Yifan Zhang[1], Yanfeng Ouyang[1], Lianhui Qin[3], Elyse Rosenbaum[1], Corey Snyder[3], Peter Seiler[4], Geir Dullerud[1], Xiaojia Shelly Zhang[1], Zuofu Cheng[1], Pavan Kumar Hanumolu[1], Jian Huang[1], Mayank Kulkarni[5], Mahdi Namazifar[5], Huan Zhang[1], Bin Hu[1†]

[1]University of Illinois at Urbana-Champaign, [2]University of Pennsylvania,
[3]University of California San Diego, [4]University of Michigan, [5]Amazon AGI
*Core Contributors    †Corresponding Author
Project page: https://agi4engineering.github.io/Eng-Design/
Dataset page: https://huggingface.co/datasets/opt1zer/EngDesign

## Abstract

Modern engineering, spanning electrical, mechanical, aerospace, civil, and computer disciplines, stands as a cornerstone of human civilization and the foundation of our society. However, engineering design poses a fundamentally different challenge for large language models (LLMs) compared with traditional textbook-style problem solving or factual question answering. Although existing benchmarks have driven progress in areas such as language understanding, code synthesis, and scientific problem solving, real-world engineering design demands the synthesis of domain knowledge, navigation of complex trade-offs, and management of the tedious processes that consume much of practicing engineers' time. Despite these shared challenges across engineering disciplines, no benchmark currently captures the unique demands of engineering design work. In this work, we introduce ENGDESIGN, an **Eng**ineering **Design** benchmark that evaluates LLMs' abilities to perform practical design tasks across nine engineering domains. Unlike existing benchmarks that focus on factual recall or question answering, ENGDESIGN uniquely emphasizes LLMs' ability to synthesize domain knowledge, reason under constraints, and generate functional, objective-oriented engineering designs. Each task in ENGDESIGN represents a real-world engineering design problem, accompanied by a detailed task description specifying design goals, constraints, and performance requirements. ENGDESIGN pioneers a *simulation-based evaluation paradigm* that moves beyond textbook knowledge to assess genuine engineering design capabilities and shifts evaluation from static answer checking to dynamic, simulation-driven functional verification, marking a crucial step toward realizing the vision of engineering Artificial General Intelligence (AGI).

## 1   Introduction

Modern engineering, spanning electrical, mechanical, aerospace, civil, and computer disciplines [Chen, 2004, Grote and Hefazi, 2021, Chen and Liew, 2002, Blockley, 2012], stands as a cornerstone of human civilization and the foundation of our society. From the electrical grids powering our cities

39th Conference on Neural Information Processing Systems (NeurIPS 2025) Track on Datasets and Benchmarks.

to the aerospace systems launching us beyond Earth, from analog integrated circuits forming the foundation of electronics to the structural designs supporting our skylines, engineering disciplines have woven themselves into the fabric of human existence [Ten and Hou, 2024, Paul et al., 2023, Gray et al., 2009]. At its core, engineering design is the systematic and creative process that engineers use to solve problems and create functional products, systems, or processes, transforming requirements and constraints into tangible solutions that meet human needs. Recent advances in large language models (LLMs) have led to remarkable performance on conventional question-answering (QA) benchmarks, with strong results across a wide range of tasks—from textbook-level scientific problem solving [Rein et al., 2024, Wang et al., 2024b, Du et al., 2025, Zou et al., 2024], to code synthesis [Chen et al., 2021, Jain et al., 2024, Chan et al., 2024], and even to answering textbook-level questions across various engineering domains [Kevian et al., 2024, Li et al., 2024, Skelic et al., 2025]. Today, industry leaders aspire to build on such progress in LLMs to create general-purpose AI engineers capable of bringing to life humanity's boldest ambitions-from interstellar starships to Dyson spheres that harvest stellar energy-leveraging these foundation models' vast knowledge across engineering domains and their potential to transcend traditional disciplinary boundaries that constrain human engineers [Business Wire, 2025].

However, practical engineering design confronts LLMs with a grand challenge profoundly different and vastly more complex than conventional factual recall or textbook-level engineering problem solving. Specifically, real-world engineering design demands the synthesis of domain knowledge, navigation of complex trade-offs, management of the tedious processes that consume much of practicing engineers' time, and rigorous validation through domain-specific simulators to ensure that designs meet functional requirements and safety constraints. Yet despite progress in textbook-level engineering problem solving, no existing benchmark captures the distinctive demands of practical design work–with its multifaceted, open-ended challenges across multiple engineering disciplines.

In this work, we introduce ENGDESIGN, the first benchmark for holistically evaluating LLMs on real-world, multi-domain engineering design challenges. Unlike traditional Question-Answer (QA) formats, our benchmark requires models to function as practicing engineers, producing functional solutions, such as dynamical system controllers, material structure designs, analog integrated circuits, or GPU architectures that meet rigorous performance requirements. ENGDESIGN uniquely emphasizes the synthesis of domain knowledge, constraint-based reasoning, and the generation of functional, objective-oriented designs, spanning practical design problems across nine engineering domains: Operating System Design, Computer Architecture Design, Control System Design, Mechanical Systems, Structural Design, Digital Hardware Design, Analog Integrated Circuit Design, Robotics, and Signal Processing. Each task in ENGDESIGN is accompanied by an executable evaluation pipeline that validates designs against task requirements using domain-specific simulation tools such as SPICE simulations [Roberts, 1996], structural finite element analysis [Bhavikatti, 2005], MATLAB Control System Toolbox [Chiang and Safonov, 1984], and other scientific software tailored to specific engineering domains. Our approach establishes a revolutionary evaluation paradigm: assessment using engineering simulation tools. This simulation-based approach fundamentally shifts evaluation from linguistic pattern matching to functional verification, ensuring that solutions are assessed based on their engineering merit rather than textual plausibility.

ENGDESIGN advances LLM evaluation methodology by moving beyond conventional question-answering benchmarks to address the open-ended, constraint-driven nature of real-world engineering design. Our key contributions include:

- **The first multi-domain engineering design benchmark.** Unlike QA benchmarks that test narrow factual or procedural knowledge, ENGDESIGN evaluates LLMs on complex engineering design challenges spanning diverse engineering disciplines (e.g., mechanical, electrical, civil), requiring synthesis of domain knowledge, constraint satisfaction, and strong reasoning capabilities.

- **Executable simulation-based evaluation.** We replace static correctness checks with dynamic, domain-specific evaluation pipelines. Each task includes human-designed task-specific evaluation scripts that rigorously verify functional feasibility (e.g., via simulation, constraint validation, or performance testing), ensuring objective, reproducible scoring grounded in real engineering standards.

- **Partial-credit grading for incomplete solutions.** Conventional benchmarks often use binary scoring, but ENGDESIGN's rubrics quantify incremental progress (e.g., 20/100 for

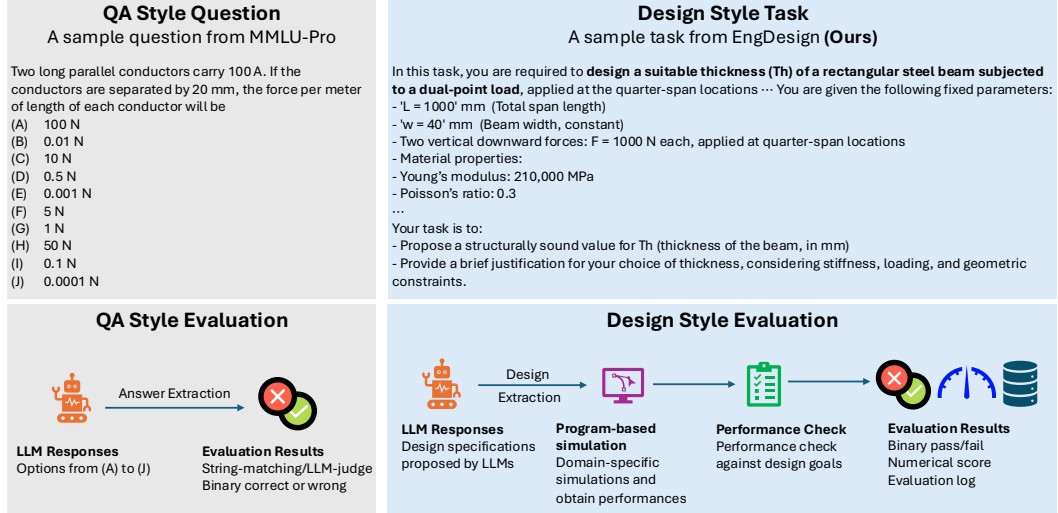

Figure 1: Comparison between conventional QA-style benchmarks (left) and the design-style benchmark ENGDESIGN (right). Conventional QA benchmarks evaluate LLMs through static answer extraction and string-matching, while ENGDESIGN involves open-ended design tasks with potentially non-unique solutions. LLMs must propose candidate design specifications, which are evaluated via program-based simulations and performance validation pipelines.

> delivering a stable controller that meets partial performance requirements). This granularity reveals nuanced capability gaps and rewards iterative refinement.

- **Empirical validation of design competencies.** Through large-scale evaluations across 10+ state-of-the-art LLMs including both general-purpose LLMs and reasoning models, we demonstrate ENGDESIGN's ability to expose critical limitations in AI systems—such as over-reliance on textual patterns or failure to handle trade-offs—that are invisible to traditional benchmarks.

We believe that ENGDESIGN marks a crucial milestone toward realizing engineering AGI by breaking new ground with its comprehensive coverage of engineering design problems across diverse domains, each requiring deep technical expertise and domain-specific simulation tools for rigorous evaluation. This benchmark not only measures what models know but what they can actually design, bridging the chasm between textbook-level knowledge understanding and practical engineering capability.

## 2  ENGDESIGN

In this section, we introduce ENGDESIGN, a multi-domain benchmark developed to evaluate the capabilities of LLMs in real-world engineering design tasks. Unlike conventional QA benchmarks, ENGDESIGN adopts a rigorous simulation-based evaluation pipeline to assess model performance in practical, design-oriented scenarios. Table 1 summarizes the benchmark's key statistics: it comprises 101 design tasks spanning 9 engineering domains, with a total of 473 gradable items. While the task distribution may appear uneven, it naturally reflects the domain expertise of our contributors and the filtering rigor imposed by our multi-stage review process. Notably, the average prompt length in ENGDESIGN is 778.71 tokens, substantially higher than typical QA benchmarks[1], highlighting the contextual richness and complexity of realistic engineering design problems. Word clouds generated from ENGDESIGN prompts are provided in Appendix C.

Among the 101 tasks in ENGDESIGN, 34 tasks require domain-specific scientific software such as MATLAB or Cadence for evaluation, while the remaining 67 tasks are fully open-sourced and evaluated using manually authored evaluation scripts. We consolidate these tasks into a subset

---

[1]We adopt Byte Pair Encoding (BPE) tokenizer for OpenAI models, available at Tiktoken. Token length comparison between ENGDESIGN and other QA benchmarks in Appendix C.1.

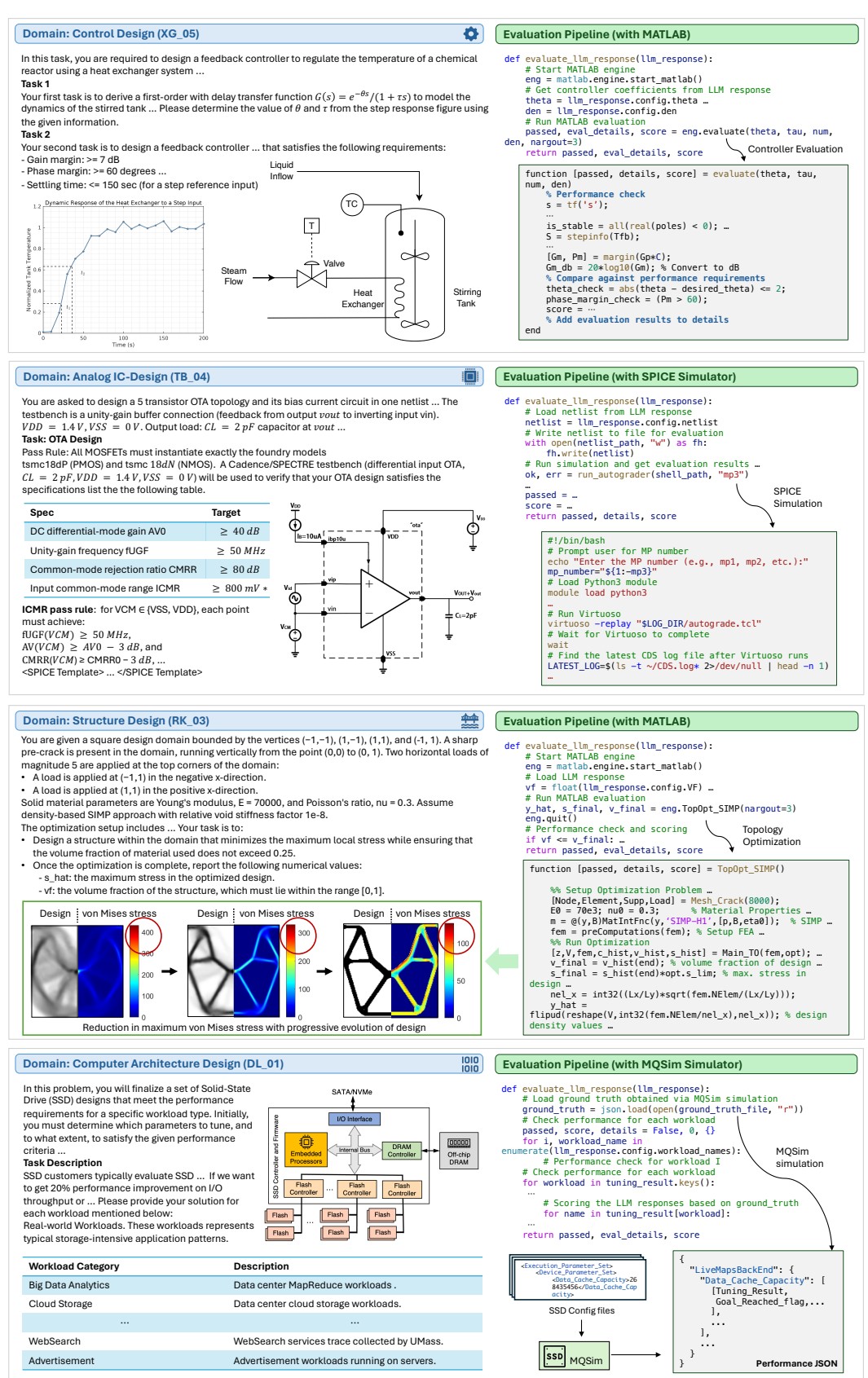

Figure 2: Selected demonstration tasks from ENGDESIGN.

Table 1: Statistics of ENGDESIGN. Token counts are computed vis `o200k_base` in Tiktoken.

| Engineering Domain | # tasks | | | # rubrics | # query tokens | | |
|---|---|---|---|---|---|---|---|
| | Open | Closed | Total | | Max | Min | Avg |
| Operating System Design | 8 | 0 | 8 | 66 | 2938 | 310 | 1103.25 |
| Computer Arch Design | 5 | 0 | 5 | 20 | 4385 | 2348 | 3539.60 |
| Control Design | 7 | 11 | 18 | 100 | 1361 | 209 | 634.44 |
| Mechanical Systems | 6 | 1 | 7 | 32 | 781 | 225 | 391.14 |
| Structure Design | 7 | 6 | 13 | 25 | 483 | 186 | 345.31 |
| Digital Hardware Design | 13 | 4 | 17 | 58 | 1715 | 206 | 515.65 |
| Analog IC Design | 0 | 5 | 5 | 23 | 2136 | 547 | 1196.6 |
| Robotics | 10 | 0 | 10 | 68 | 1485 | 192 | 771.9 |
| Signal Processing | 11 | 7 | 18 | 81 | 2304 | 151 | 611.72 |
| Overall | 67 | 34 | 101 | 473 | 4385 | 151 | 778.71 |

called ENGDESIGN-OPEN to support broader community adoption without licensing constraints. Additionally, 23 tasks of ENGDESIGN incorporate images as part of the task input to LLMs.

## 2.1 Task Structure

Each task of ENGDESIGN consists of the following four key components:

**1. Task Description.** This part is the query prompt fed into the LLMs, offering a clear and detailed definition of the engineering design problem, including design objectives, specifications, constraints.

**2. Evaluation Rubrics.** Given the complexity of ENGDESIGN tasks, each task is further decomposed into multiple gradable items evaluated individually during the performance check stage. The evaluation rubrics define the assessment criteria and scoring metrics, with a full score of 100. This enables models to receive partial credit even if the design does not fully meet all specified requirements.

**3. Evaluation Pipeline.** Each task includes automated evaluation scripts that assess the LLMs' design. It returns a binary pass/fail indicator, numerical score, and evaluation logs for further analysis.

**4. Reference Design.** Each task provides a validated reference design that fully satisfies all specified requirements, ensuring the feasibility and realism of the design challenge.

Figure 2 shows four demonstrated design examples from ENGDESIGN including task descriptions and evaluation pipelines.

## 2.2 Evaluation Pipeline

Unlike conventional benchmarks, which often have a single golden answer that allows evaluation through exact string matching or LLM-as-judge scoring, our engineering design tasks inherently lack a unique *golden design*. For example, there are infinitely many valid controller designs that can regulate the temperature of a stirred tank modeled by a first-order system while satisfying both time-domain and frequency-domain performance requirements. To address this challenge, our benchmark introduces a **simulation-based evaluation pipeline**. Specifically, for each LLM-generated design, we first parse the key design components from the response automatically (such as code snippets, key parameters, etc.). These components are then fed into our evaluation pipeline, which runs domain-specific simulations to assess the design's performance against the specified task requirements. Guided by a per-task rubric table, the evaluation pipeline outputs three key results: a binary pass/fail indicator, a numerical score between 0 and 100, and a detailed log that records the evaluation process for further analysis. Below we discuss each step in detail.

**Structured Responses from LLMs.** To constrain the output format for a wide range of LLMs and ensure compatibility with our evaluation pipeline, we adopt the popular open-source Python library `instructor` [Liu and Contributors, 2024], which is built on top of Pydantic and facilitates structured LLM responses. By defining schema templates that specify expected fields, such as design parameters or code snippets, the `instructor` package enables LLMs to produce outputs in a pre-defined format. We provide an example at Figure 10 in Appendix D.2. Specifically, LLMs are instructed to construct

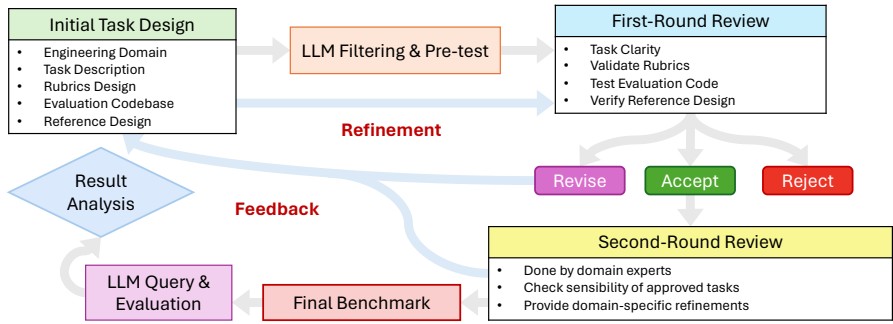

Figure 3: An overview of the construction process of ENGDESIGN, illustrating stages from initial task design, LLM filtering, and expert review to final benchmark integration, with iterative refinement and evaluation.

their responses into two main parts: (1) a `reasoning` field, which contains the step-by-step reasoning process for solving the task, and (2) a `ConfigFile` class, which summarizes the final design results, including their design choices or code snippets. During evaluation, the fields defined in `ConfigFile` can be automatically parsed to trigger the simulation-based evaluation pipeline.

**Task-Specific Evaluation Pipeline.** A simulation-based evaluation pipeline was designed for each task. For example, control engineering tasks may evaluate closed-loop dynamics through metrics such as rise time, settling time, overshoot, and phase/gain margins via MATLAB. The evaluation pipeline outputs three key results: (1) a binary pass/fail indicator for meeting all performance requirements, (2) a numerical score (0–100) reflecting fine-grained performance analysis, and (3) an evaluation log capturing simulation outputs, performance metrics, or error messages.

## 2.3 Construction of ENGDESIGN

The construction of ENGDESIGN follows a multi-stage process that integrates both automated validation and human expertise as shown in Figure 3:

1. **Initial Task Design.** We recruited graduate students and researchers from various engineering disciplines to contribute initial task proposals, drawing on open-source resources and their domain knowledge. Contributors were guided to follow a standardized submission format, which includes a task description, evaluation rubric, executable evaluation codebase, and a reference solution.

2. **LLM Filtering and Pre-Test.** Submitted tasks undergo an initial filtering phase using a language model (o4-mini) to assess prompt sufficiency and the functionality of the evaluation code. Specifically, we prompt the LLM to classify the task's engineering domain and determine whether the information provided is adequate to solve the problem. If the model flags missing or unclear details, we work with the original contributor to address those gaps.

3. **First-Round Review.** Tasks that pass the pre-test enter a first-round review involving close collaboration between reviewers and the original authors. The review ensures that:

   - The task description is clear, self-contained, and complete.
   - The evaluation rubric is well-defined with partial credit, and aligns with the task goals.
   - The evaluation codebase is executable, robust, and handles typical edge cases.
   - The reference design meets all performance requirements, validating the task feasibility.

   Based on this review, tasks fall into one of three categories: **Accept**: the task meets all requirements and proceeds to expert review; **Revise**: minor issues are identified and addressed in collaboration with the contributor; or **Reject**: the task is excluded due to critical issues such as unfeasibility, triviality, or misalignment with design-oriented evaluation.

4. **Second-Round Review with Domain Experts.** Tasks accepted in the first round are reviewed by faculty experts in the corresponding engineering domain. These experts assess the technical soundness and relevance of each task to ensure it serves as a meaningful evaluation of LLMs in realistic design settings.

Table 2: Average pass rate (%) results (each task evaluated over 3 trials). * Model is not multi-modal and was evaluated only on the text-only subset. We report text-only results for all models in Appendix D.4. The domain abbreviations are: AICD = Analog Integrated Circuit Design, Arch = Computer Architecture Design, Ctrl = Control Design, DHD = Digital Hardware Design, Mech = Mechanical Systems, OS = Operating System Design, Robo = Robotics, SigP = Signal Processing, Stru = Structure Design. The best results for each column are highlighted in bold.

| Model | Overall | AICD | Arch | Ctrl | DHD | Mech | OS | Robo | SigP | Stru |
|---|---|---|---|---|---|---|---|---|---|---|
| **Chat Models** | | | | | | | | | | |
| GPT-4o | 15.68 | 0.00 | 20.00 | 18.52 | 10.26 | 0.00 | 4.17 | 26.67 | 17.65 | 25.64 |
| DeepSeek-v3* | 17.92 | 0.00 | 0.00 | 27.27 | 25.64 | 0.00 | 0.00 | 16.67 | 12.82 | 38.89 |
| Gemini-2.0-Flash | 14.16 | 0.00 | 0.00 | 15.15 | 16.67 | 4.76 | 0.00 | 20.00 | 5.13 | 36.11 |
| Claude-3.7-Sonnet | 22.61 | 0.00 | **40.00** | 16.67 | 33.33 | 14.29 | 0.00 | 33.33 | 21.57 | 30.77 |
| **Reasoning Models** | | | | | | | | | | |
| o1 | 29.17 | 0.00 | **40.00** | 24.07 | 41.03 | 14.29 | **37.50** | 50.00 | 25.49 | 23.08 |
| o3 | **34.38** | 0.00 | **40.00** | 35.19 | 20.51 | 23.81 | 25.00 | 63.33 | **41.18** | 30.77 |
| o3-high | 33.57 | 0.00 | **40.00** | 27.78 | 20.51 | **28.57** | 30.43 | **68.97** | 35.29 | 41.03 |
| o4-mini | 31.60 | 0.00 | 33.33 | 24.07 | 41.03 | 19.05 | 29.17 | 43.33 | 33.33 | 41.03 |
| o4-mini-high | 34.04 | 0.00 | **40.00** | 27.78 | **47.22** | 19.05 | **37.50** | 46.67 | 35.29 | 35.90 |
| Gemini-2.5-Pro | 29.54 | 0.00 | 0.00 | 33.33 | 43.59 | 0.00 | 9.52 | 56.67 | 12.82 | **50.00** |
| DeepSeek-R1* | 25.53 | 0.00 | 36.36 | **36.36** | 38.46 | 4.76 | 5.26 | 26.67 | 20.51 | 41.67 |
| Claude-3.7-Thinking | 20.07 | 0.00 | 33.33 | 18.52 | 17.95 | 9.52 | 0.00 | 40.00 | 19.61 | 28.21 |

5. **Final Integration.** Tasks that pass both review stages are standardized to match the benchmark's formatting and structural guidelines, and are formally included in the ENGDESIGN.

# 3 Experiments

We evaluate a set of representative LLMs on the ENGDESIGN and analyze their performance.

**Evaluated LLMs.** For chat models, we include GPT-4o, Claude-3.7-Sonnet, Gemini-2.0-Flash, and DeepSeek-v3. For reasoning models, we evaluate o1, o3, o3-high, o4-mini, o4-mini-high, Claude-3.7-Thinking, DeepSeek-R1, and Gemini-2.5-Pro.

**Evaluation Metrics.** We report three primary evaluation metrics: (1) **Average Pass Rate**, (2) **Average Score**, and (3) **Reasoning Robustness**. Each task is evaluated over three independent trials per model. To measure reasoning robustness, we compute the ratio between the number of tasks where all three trials passed and the number of tasks where at least one trial passed. This ratio, which ranges from 0 to 1, serves as a straightforward indicator of an LLM's reasoning consistency. A value closer to 1 indicates higher robustness and more stable reasoning behavior across repeated runs. Formal definitions of all evaluation metrics are provided in Appendix D.3.

## 3.1 Main Results

Table 2 presents the average pass rates and average score across evaluated LLMs, across 9 distinct engineering design areas. The average score demonstrates a strong correlation with the pass rate, which we provided the detailed results in Appendix D.4. We make the following key observations.

**ENGDESIGN presents a highly challenging benchmark, with even the best-performing models achieving only modest pass rate.** As shown in Table 2, no model surpasses a 35% overall pass rate. The top-performing model, o3, achieves 34.38%, while widely-used chat models such as GPT-4o and Gemini-2.0-Flash attain around 15%. Analog IC design tasks stands as the most difficult ones where all the models got 0% pass rate. These results highlight the difficulty of the benchmark, which stems from its requirements for domain-specific expertise, multi-step reasoning, and design trade-off considerations, capabilities that go beyond the strengths of current LLMs.

**Reasoning models outperform general-purpose chat models.** Models designed with enhanced reasoning capabilities, including o3, o4-mini, and DeepSeek-R1, consistently outperform general-purpose chat models across most engineering domains. However, this trend does not hold for Claude

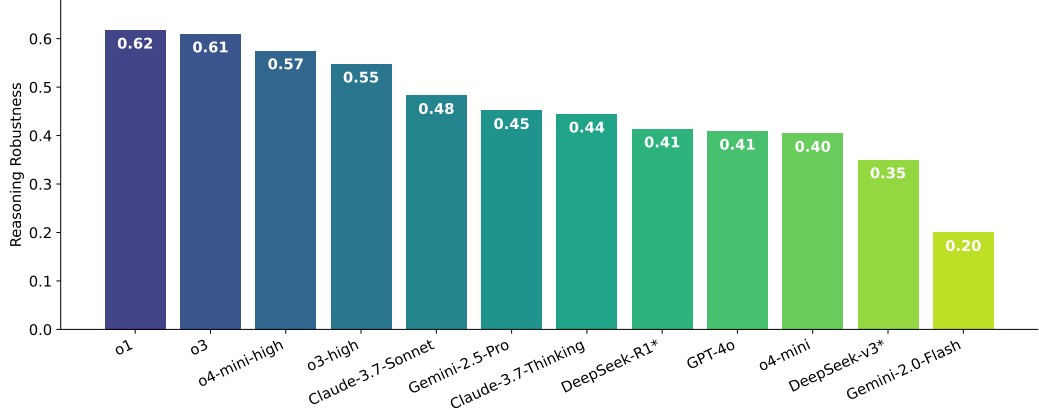

Figure 4: Reasoning robustness distribution of evaluated LLMs on ENGDESIGN.

models, where both Claude-3.7-Sonnet and Claude-3.7-Thinking exhibit similar pass rates despite their differences in reasoning emphasis.

**Reasoning Models are more robust in general.** Figure 4 shows the reasoning robustness of all evaluated models. Reasoning-focused models such as o1, o3, and o4-mini-high achieve the highest robustness scores (0.62, 0.61, and 0.57, respectively), indicating strong consistency across repeated trials on tasks they are capable of solving. In contrast, chat-oriented models like Gemini-2.0-flash and DeepSeek-v3 exhibit substantially lower robustness (0.20 and 0.35), reflecting less stable reasoning behavior. Notably, the gap in reasoning robustness between top-performing reasoning models and weaker chat models is more pronounced than in average pass rate as shown in Table 2—up to a $3\times$ performance difference in robustness compared to a $2\times$ difference in pass rate. This highlights robustness as a distinguishing strength of reasoning models.

**Token Consumption**. Reasoning models require substantially more compute due to longer inference time. We analyze the number of completion tokens generated across different models. As shown in Figure 5, the reasoning model (o1) produces significantly more tokens than the non-reasoning model (GPT-4o) to achieve performance gains. Optimizing the latency–performance trade-off will be essential for making future models more accessible and deployable in real-world engineering design scenarios. Additional results are provided in Appendix D.5.

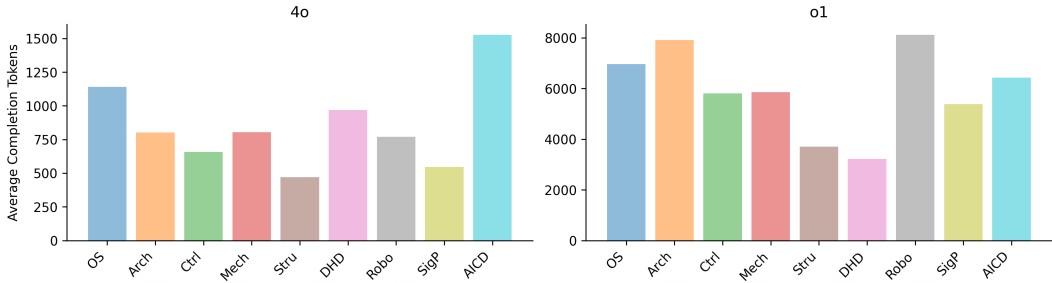

Figure 5: Average token consumption of evaluated LLMs on ENGDESIGN.

## 3.2 Iterative Design

To emulate the workflow of human engineers, we implement an iterative design protocol that allows LLMs to refine their solutions based on feedback from previous attempts. In our implementation, the LLM is provided with its previous design output along with corresponding evaluation results, such as scores, performance metrics, and diagnostic logs, and is then prompted to generate an improved design in the subsequent iteration. The prompt we used for iterative design is presented in Figure 6.

| **LLM Prompt for Iterative Design** |
| :--- |
| Task Description: original task description |
| Previous Responses: previous responses |
| Evaluation Feedback: previous evaluation results |
| **Instruction:** Please review the feedback above and propose an improved solution based on the evaluation results. |

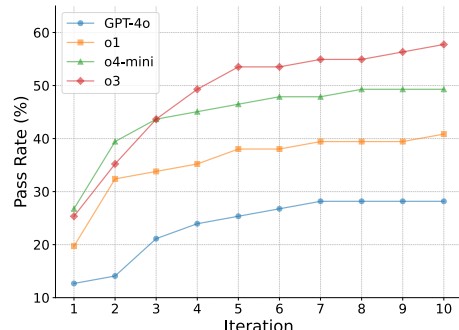

Figure 6: **Left**: LLM prompt for iterative design. **Right**: Average pass rate across iterations, showing how performance evolves with feedback-driven updates.

As shown in Figure 6, model performance consistently improves with additional iterations. Notably, o3 achieves almost a 60% pass rate after ten iterations.[2]. However, we also observed that iterative design does not help in all cases. For example, in Analog IC design tasks, models still fail to meet the requirements even after ten iterations.

### 3.3 Failure Analysis

To better understand the limitations of LLMs in engineering design tasks, we define five primary error categories observed in ENGDESIGN:

1. **Domain Knowledge Error (DKE).** The model lacks essential engineering knowledge required to interpret or solve the task correctly.

2. **Constraint Violation Error (CVE).** The model generates designs that violate explicitly stated constraints, such as structural limits, performance bounds, or budgetary requirements.

3. **Prior Knowledge Overreliance (PKO).** The model applies memorized patterns or text-book solutions without adapting them to the specific requirements of the task, resulting in suboptimal or invalid outputs.

4. **Hallucination (HAL).** The model fabricates content, such as equations, parameters, or designs, that is unsupported by the task description or domain knowledge.

5. **Computation Error (CE).** The model performs arithmetic or symbolic computations incorrectly despite correct methodological reasoning.

6. **Others.** Residual errors that do not fit the above categories

Table 3: Error types statistics for three representative models.

| Model | DKE | CVE | PKO | HAL | COM | Others |
| :--- | :--- | :--- | :--- | :--- | :--- | :--- |
| o4-mini | 33.3% | 25.2% | 18.8% | 12.6% | 9.0% | 0.9% |
| Gemini-2.5-Pro | 31.9% | 31.9% | 15.9% | 12.4% | 6.2% | 1.7% |
| Claude-3.7-Sonnet | 30.7% | 36.0% | 10.5% | 13.2% | 7.0% | 2.6% |

We analyzed the responses of three representative models (o4-mini, Gemini-2.5-Pro, and Claude-3.7-Sonnet) on ENGDESIGN tasks where they failed evaluation, and manually annotated each failure with its corresponding error types. The results are summarized in Table 3. Given the complexity of engineering design problems, many responses exhibited multiple failure modes; we therefore allowed multi-label annotations per task.

---

[2]Due to time and resource constraints, the iterative design protocol was applied to a selected subset (71 tasks) of ENGDESIGN tasks using four representative models: GPT-4o, o1, o3, and o4-mini. As a result, reported pass rates in the iterative experiments are slightly lower than the ones report in Table 2. Nonetheless, this subset demonstrates the effectiveness of feedback-driven refinement and highlights the potential of LLMs to improve design quality through iterative design.

Our analysis yields several key observations[3]:

**1. Dominant failure modes:** Insufficient domain knowledge and constraint violations together account for roughly 55–67% of all failures. This indicates that LLMs frequently struggle to apply domain-specific principles, satisfy task constraints, and generalize beyond memorized patterns.

**2. Second-tier issues:** Over-reliance on prior knowledge and hallucinations contribute 25–30%, suggesting that incorrect reuse or misapplication of known facts remains a significant challenge.

**3. Low arithmetic fragility:** Pure computational errors are rare ($\leq 9\%$), implying that numerical reasoning is not the primary bottleneck for current frontier models.

# 4 Related Work

**Large Language Models.** Advances in large language models (LLMs) have been propelled by scaling, instruction tuning, and improved reasoning. Frontier models such as GPT-3 and GPT-4 [Brown et al., 2020, Achiam et al., 2023], Claude [Anthropic, a], Gemini [Team et al., 2023], and DeepSeek [Liu et al., 2024a] exhibit broad general capabilities. Recent work enhances reasoning through architectural and prompting innovations, including Chain-of-Thought [Wei et al., 2022] and least-to-most prompting [Zhou et al., 2022]. These developments have led to reasoning-oriented models such as OpenAI's o-series [OpenAI], Claude-Thinking [Anthropic, b], Gemini 2.5 Pro [Google DeepMind], and DeepSeek-R1 [Guo et al., 2025]. Yet, their effectiveness in domain-specific contexts, particularly engineering design, remains largely unexplored. ENGDESIGN fills this gap by evaluating LLMs within realistic, simulation-based engineering workflows.

**General Purpose Benchmarks for LLMs.** A variety of benchmarks assess LLM reasoning and problem-solving capabilities. MMLU [Hendrycks et al., 2020] and MMLU-Pro [Wang et al., 2024b] evaluate broad subject knowledge, while GAIA [Mialon et al., 2023] and HLE [Phan et al., 2025] test long-context and high-difficulty reasoning. HumanEval [Chen et al., 2021] measures code generation accuracy, GPQA [Rein et al., 2024] targets graduate-level science, and GSM8k [Cobbe et al., 2021] focuses on arithmetic reasoning. Recent work such as DynaMath [Zou et al., 2024] evaluates VLMs' mathematical reasoning robustness.

**LLMs in Engineering Domains.** Recent studies have started to explore the applicability of LLMs to engineering contexts. For example, [Syed et al., 2024, Kevian et al., 2024, Guo and Zhao, 2025, Xu et al., 2025, Xiong et al., 2025, Eslaminia et al., 2024, Ogo and Koga, 2024] evaluate LLMs across engineering subfields using curated QA datasets. However, these efforts only focuses one specific domain or largely focus on factual recall rather than generative design capabilities. Several domain-specific studies examine LLM-assisted design workflows. For example, ControlAgent [Guo et al., 2024] integrates control theory solvers for automated controller design, while AnalogCoder [Lai et al., 2024] and SPICED [Chaudhuri et al., 2024] target analog circuit and SoC design using prompt engineering and retrieval-based techniques. Retrieval-augmented and programmatic strategies have also been explored to enhance LLM performance [Ghosh and Team, 2024, Alsaqer et al., 2024]. Other studies examine LLMs in mechanical design [Lu et al., 2024], cross-domain creativity [Jia et al., 2024], and computational engineering [Hamann et al., 2024, Xu et al., 2024, Majumder et al., 2024, Makatura et al., 2023]. Compared to prior work, ENGDESIGN spans multiple engineering domains and introduces a novel evaluation framework that combines generative outputs from LLMs with simulation-based evaluation to assess performance in realistic design settings.

# 5 Conclusion

In this work, we introduced ENGDESIGN, a comprehensive benchmark for rigorously evaluating large language models (LLMs) in realistic, multi-domain engineering design scenarios. ENGDESIGN emphasizes end-to-end design synthesis, constraint satisfaction, and simulation-based validation across diverse engineering disciplines. Our results show that ENGDESIGN poses substantial challenges: even the most capable frontier models achieve only modest performance. We envision ENGDESIGN as a foundation for future research at the intersection of language, reasoning, and engineering intelligence. By providing a standardized and reproducible evaluation testbed, we hope to accelerate progress toward the long-term goal of developing general-purpose AI engineers.

---

[3]Additional failure cases and qualitative analyses are provided in Appendix E.

## Acknowledgment

Xingang Guo and Bin Hu are generously supported by the Amazon AICE award.

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

# A  Limitations and Broader Impact

**Limitations**

While ENGDESIGN provides a comprehensive benchmark for evaluating LLMs in engineering design tasks, some limitations remain:

1. ENGDESIGN currently covers 9 engineering design domains. Although these are diverse and representative, the benchmark does not yet encompass the full breadth of engineering disciplines or subfields.

2. Our evaluation includes a selection of widely used and high-performing LLMs. However, it is not exhaustive. Many emerging or smaller-scale models are not included in our evaluation.

3. We employ the `instructor` framework to enforce structured output from LLMs. While this tool has shown reliable performance throughout our evaluations, there are instances where LLMs fail to adhere to the expected output format. As `instructor` is still under active development, future improvements will be necessary to further enhance its robustness and compatibility across models.

**Broader Impact**

This work aims to advance our understanding of how LLMs perform in complex, real-world engineering design scenarios. However, we highlight the following considerations:

- **Not a replacement for human engineers.** Our benchmark is intended to assess LLM capabilities, not to replace the expertise and judgment of professional engineers. Engineering design involves safety-critical decision-making, physical prototyping, and domain-specific knowledge that current LLMs cannot fully replicate.

- **Supporting real-world design workflows.** We hope that LLMs, when properly evaluated and deployed, can assist engineers by automating routine tasks, enhancing ideation, and accelerating early-stage design processes—particularly in settings with limited access to expert resources.

- **Need for rigorous safety checks.** Any LLM-generated design, particularly those intended for deployment in safety-critical domains such as aerospace, biomedical devices, civil infrastructure, or autonomous systems, must undergo thorough validation and safety assessment by qualified human experts before real-world application.

- **Ethical use and accessibility.** Care must be taken to ensure that advances in AI-assisted engineering design benefit a broad and diverse community. Efforts should be made to avoid misuse, bias amplification, or overreliance on unverified outputs.

# B  More on Related Work

Recent years have seen a surge of domain-specific engineering benchmarks that evaluate AI systems on isolated reasoning or coding sub-tasks. However, most existing efforts fall short of assessing end-to-end design capability, that is, the ability to synthesize, implement, and validate complete engineering systems under realistic constraints. Below, we review representative benchmarks across key engineering domains and highlight how EngDesign differs by emphasizing holistic design and simulation-based validation.

**Operating Systems**  OSVBench [Li et al., 2025a] focuses on the specification and verification of simplified operating system components using formal methods. While it advances progress in formal verification, its tasks are limited to checking correctness of given code fragments. In contrast, ENGDESIGN-OS tasks require participants to design and simulate new operating system subsystems (e.g., schedulers, memory allocators), moving beyond static specification toward dynamic design.

**Computer Architecture**  Benchmarks such as FIXME [Wan et al., 2025] and ChatCPU [Wang et al., 2024a] center on verifying or patching existing CPU components through code reasoning. These tasks measure correctness and local optimization. ENGDESIGN-ARCH, by comparison, tasks models

with designing and optimizing new micro-architectures and quantitatively validates performance through simulation, bridging reasoning and generative design.

**Control Systems Design**    ControlBench [Kevian et al., 2024] and ControlEval [Guo et al., 2024] examine language models' ability to answer control-theoretic questions or tune simple PID controllers. These benchmarks remain largely static, focusing on analytical understanding rather than synthesis. ENGDESIGN-CTRL extends the scope to diverse control system design problems, including state-feedback, robust control, and optimal control, and evaluates designs via dynamic simulations.

**Mechanical Design**    Prior mechanical design studies, such as MechAgents [Ni and Buehler, 2024] developed a set of AI agents can solve specified elasticity problem, which focuses on one type of problems. ENGDESIGN-MECH introduces multiple mechanism design tasks with physics-based simulation and objective trade-offs, providing a consistent evaluation framework across mechanical systems.

**Structural Engineering**    Benchmarks like AEC-bench [Liang et al., 2025] that integrate a benchmark for architecture, engineering, and construction field domain questions and DrafterBench [Li et al., 2025b] that focuses on evaluting LLMs on technical drawing revision all focuses on speicify question and answer evaluation scenario. ENGDESIGN-STRU instead requires sizing new structures under material and load trade-offs, capturing the creative reasoning central to structural design.

**Digital Hardware Design**    VerilogEval [Liu et al., 2023] focuses on writing and verifying RTL code snippets. These benchmarks measure correctness at the snippet level but not full design integration. In contrast, ENGDESIGN-DHD targets complete module-level design with explicit trade-offs between latency, area, and power—mirroring real-world hardware co-design challenges.

**Analog and Integrated Circuit Design**    Recent efforts such as AnalogCoder [Lai et al., 2025a] and AnalogCoder-Pro [Lai et al., 2025b] focuses on agent design on IC design domains. EngDesign-AICD advances this direction by provide a evaluation testbed and validates models' performance through SPICE-like simulation.

**Robotics**    Benchmarks such as [Goebel and Zips, 2025, Yin et al., 2024] target high-level reasoning, perception, or planning. These are complementary but orthogonal to engineering design, as they do not involve building controllers or physical systems. ENGDESIGN-ROBO bridges this gap by coupling low-level control design with physics simulation, assessing dynamic feasibility and safety in robotic mechanisms.

**Signal Processing**    SensorBench [Quan et al., 2025] focuses on sensor data. They emphasize data understanding, but not system-level design. ENGDESIGN-SIGP introduces tasks involving filter design, sensor configuration, and system-level optimization, encompassing a broader range of design decisions.

Across these domains, ENGDESIGN distinguishes itself by requiring end-to-end design reasoning, cross-disciplinary synthesis, and simulation-based validation. Whereas prior benchmarks evaluate correctness or reasoning on partial tasks, EngDesign systematically measures an AI model's ability to conceptualize, implement, and verify complex engineered systems—an essential step toward assessing the real-world utility of intelligent design agents.

## C    More on ENGDESIGN

### C.1    Prompt Token Length Comparison

Table 4 reports the average number of tokens in the input prompts for various benchmarks, measured using the `o200k_base` tokenizer. Notably, ENGDESIGN exhibits significantly longer prompts (averaging 778.71 tokens) compared to other popular QA-style benchmarks such as MMLU-Pro (61.76), HLE (250.03), and GSM8K (58.46). This reflects the greater contextual and structural complexity involved in realistic engineering design tasks, which often require extensive problem descriptions and domain-specific constraints.

Table 4: Average token counts across different benchmarks using the `o200k_base` tokenizer.

| Benchmark | MMLU-Pro | HLE | SuperGPQA | DynaMATH | GSM8K | EngDesign |
|---|---|---|---|---|---|---|
| Average Token # | 61.76 | 250.03 | 57.89 | 27.83 | 58.46 | 778.71 |

## C.2    More on construction of ENGDESIGN

As described in Section 2.3, the development of ENGDESIGN follows a rigorous multi-stage process, including initial task design and two rounds of review to ensure task quality and relevance. During the initial stage, over 40 task contributors submitted a total of 186 task proposals. Following thorough evaluation and revision across the two review phases, 101 tasks were accepted for inclusion in ENGDESIGN, while the remaining 85 were discarded due to various issues such as insufficient clarity, infeasibility, or misalignment with the benchmark's design-oriented goals.

## C.3    Word Clouds of ENGDESIGN

Figure 7 and Figure 8 present the word clouds of the most frequently occurring terms across all prompts and within each domain, respectively. These visualizations underscore the benchmark's emphasis on design-specific vocabulary and highlight the diversity of engineering contexts it encompasses.

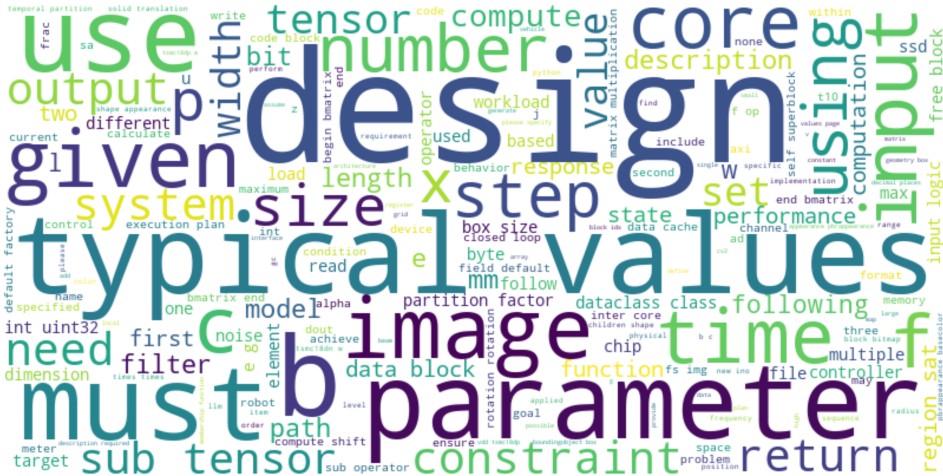

Figure 7: Word Cloud of the overall ENGDESIGN Prompts.

# D    More on Evaluation

Figure 9 illustrates the benchmarking pipeline for evaluating LLMs on engineering design tasks. In this section, we provide more details on the evalutions of ENGDESIGN.

## D.1    Detailed Experimental Setup

We use the default configuration for each LLM during evaluation. Table 5 provides detailed information on the model names, setup parameters, and special configurations used for each model.

All evaluations in this work were conducted via API access to proprietary LLMs (e.g., OpenAI, Anthropic, Google, and DeepSeek). As a result, the computational burden on our end was minimal, with no need for local GPU infrastructure. The primary cost was associated with inference-time API usage, which varies depending on the model types and token numbers. Since we did not perform model training or fine-tuning, the overall computational footprint of our study remains low.

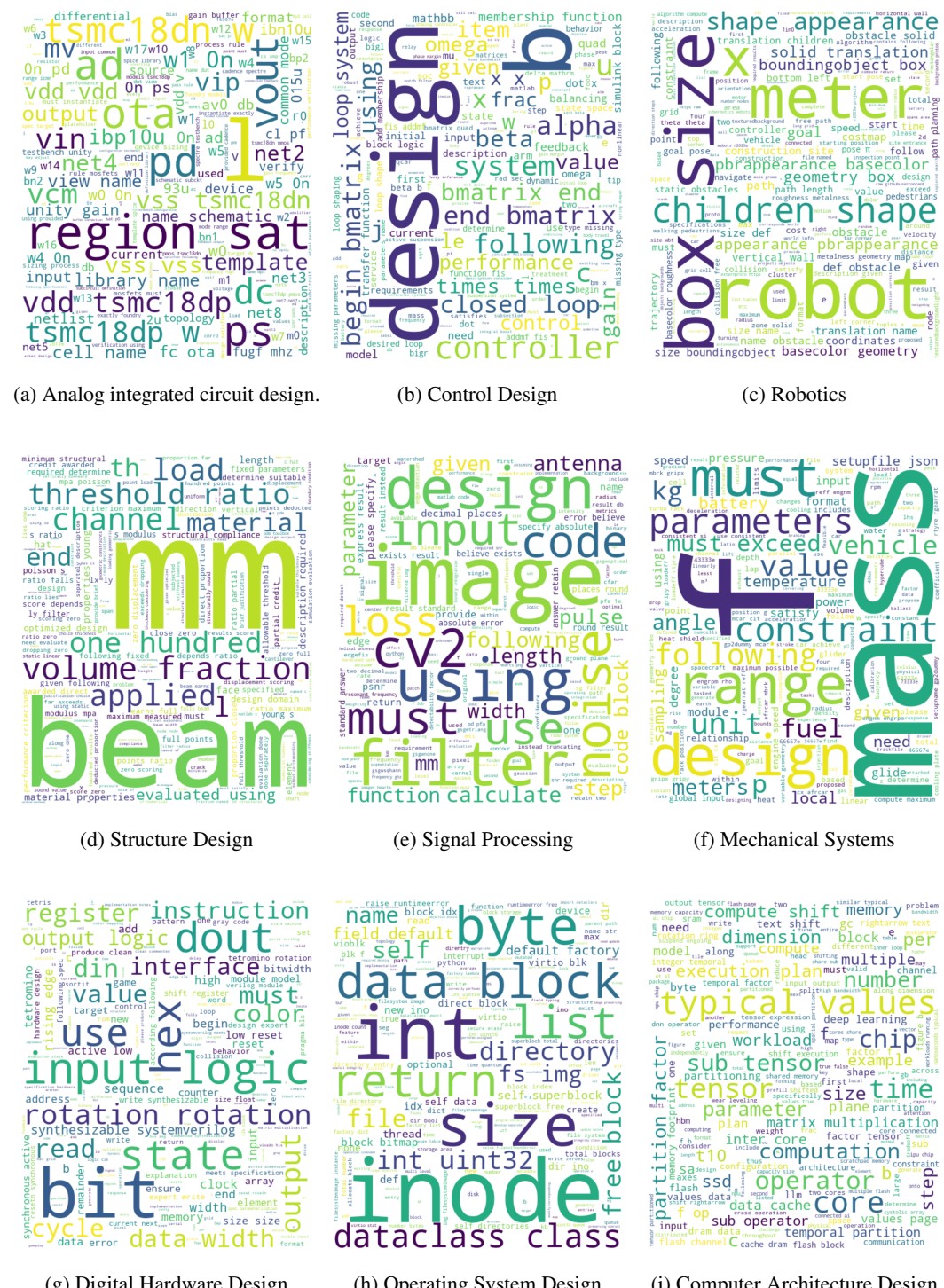

(a) Analog integrated circuit design.

(b) Control Design

(c) Robotics

(d) Structure Design

(e) Signal Processing

(f) Mechanical Systems

(g) Digital Hardware Design

(h) Operating System Design

(i) Computer Architecture Design

Figure 8: Word clouds of ENGDESIGN for each engineering domain.

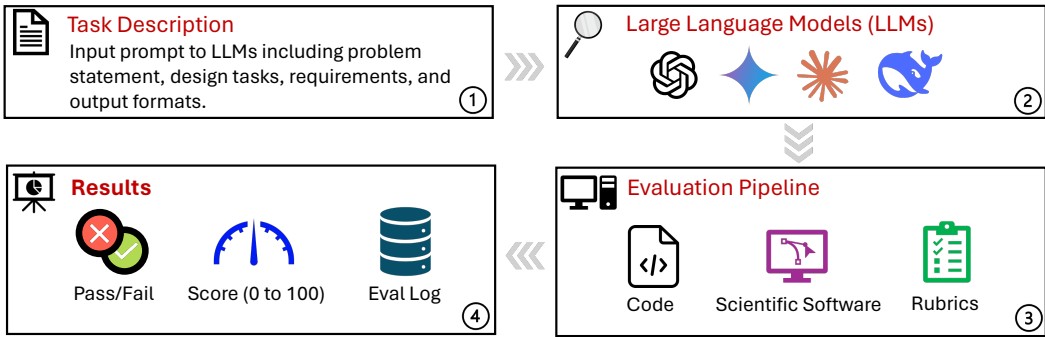

Figure 9: Overview of the ENGDESIGN Benchmarking Pipeline. The process begins with a task description curated by task contributors (Step 1). This prompt is then submitted to various LLMs (e.g., GPT, Claude, Gemini, DeepSeek) to generate candidate design proposals (Step 2). The generated outputs are evaluated using an automated codebase that includes code execution, integration with scientific software (e.g., MATLAB, Webots), and standardized scoring rubrics (Step 3). Finally, the results, including pass/fail status, quantitative scores, and detailed evaluation logs, are recorded for further analysis (Step 4).

Table 5: Model names and configuration details for each evaluated LLM.

| Model | Model Pointer & Setup |
|---|---|
| GPT-4o | `gpt-4o-2024-08-06` |
| o1 | `o1-2024-12-17` |
| o3 | `o3-2025-04-16`, reasoning effort: `medium/high` |
| o4-mini | `o4-mini-2025-04-16`, reasoning effort: `medium/high` |
| Claude-3.7-Sonnet | `claude-3-7-sonnet-20250219` |
| Claude-3.7-Thinking | `claude-3-7-sonnet-20250219`, thinking token budget: `16,000` |
| Gemini-2.0-Flash | `gemini-1.5-pro` |
| Gemini-2.5-Pro | `gemini-2.5-pro-preview-03-25` |
| DeepSeek-v3 | `DeepSeek-V3` |
| DeepSeek-R1 | `DeepSeek-R1` |

## D.2 LLM Response Structure

As discussed in Section 2.2, we use `instructor` to construct LLM responses by defining a response structure class, as illustrated in Figure 10. For each task, the response structure class includes two keys: `reasoning` and `config`. The `reasoning` component prompts the LLM to perform a detailed step-by-step reasoning process to complete the task. The `config` component defines a task-specific `ConfigFile` class, which guides the LLM to produce key design components in a predefined format. This structured output facilitates automated evaluation by making it easy to parse LLMs' design choices or code snippets.

## D.3 More on Evaluation Metrics

We evaluate model performance using three metrics: Average Pass Rate, Average Score, and Average Response Length. Each task is tested over $N$ ($N = 3$) independent trials per model. For each metric, we aggregate at the task level and then summarize across all tasks to ensure equal weighting.

**Average Pass Rate** For each task $i$, we define the pass rate as:

$$\text{pass\_rate}_i = \frac{\text{number of successful trials}}{N}$$

```
import instructor
from pydantic import BaseModel, Field

class ConfigFile(BaseModel):
    theta: float = Field(description="The value of theta")
    tau: float = Field(description="The value of tau")
    num: list[float] = Field(description="The numerator of the transfer function of the
controller")
    den: list[float] = Field(description="The denominator of the transfer function of the
controller")

# Define your desired output structure
class Response_structure(BaseModel):
    reasoning: str = Field(..., description="Detailed reasoning process to accomplish the
task, please solve all the tasks step by step")
    config: ConfigFile
```

Figure 10: An example of response structure class for `instructor` (Task XG_05).

The overall pass rate is then computed as the mean of per-task pass rates:

$$\mu_{\text{pass}} = \frac{1}{N} \sum_{i=1}^{N} \text{pass\_rate}_i$$

We report $\mu_{\text{pass}}$ along with its standard deviation across tasks:

$$\sigma_{\text{pass}} = \sqrt{\frac{1}{N-1} \sum_{i=1}^{N} (\text{pass\_rate}_i - \mu_{\text{pass}})^2}$$

**Average Score**    For each task $i$, the score is averaged over $N$ trials:

$$\text{score}_i = \frac{1}{N} \sum_{j=1}^{3} x_{i,j}$$

where $x_{i,j}$ is the score in trial $j$. The overall score is the mean of per-task scores:

$$\mu_{\text{score}} = \frac{1}{N} \sum_{i=1}^{N} \text{score}_i$$

with corresponding standard deviation:

$$\sigma_{\text{score}} = \sqrt{\frac{1}{N-1} \sum_{i=1}^{N} (\text{score}_i - \mu_{\text{score}})^2}$$

**Reasoning Robustness**    Let $T$ denote the set of all evaluated tasks. For a given model, each task $t \in T$ is evaluated over $N$ independent trials. Define:

- $N_{\text{all}}$: the number of tasks for which all $N$ trials passed, and
- $N_{\text{any}}$: the number of tasks for which at least one trial passed.

Then, the *Reasoning Robustness* $R \in [0, 1]$ is defined as:

$$R = \frac{N_{\text{all}}}{N_{\text{any}}}$$

This metric quantifies the model's consistency in reasoning. A value of $R = 1$ indicates perfect robustness. In other words, whenever a task is solvable by the model, it succeeds consistently across all trials. On the other hand, lower values indicate higher variance or instability in the model's reasoning process across repeated attempts.

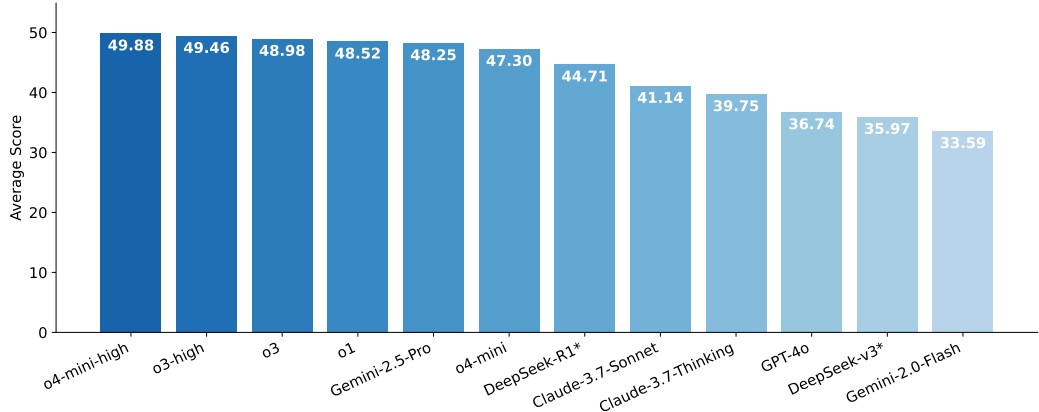

Figure 11: Average Score of evaluted LLMs on ENGDESIGN.

Table 6: Average score results (each task evaluated over 3 trials) under different domains.

| Model | Overall | AICD | Arch | Ctrl | DHD | Mech | OS | Robo | SigP | Stru |
|---|---|---|---|---|---|---|---|---|---|---|
| **Chat Models** | | | | | | | | | | |
| GPT-4o | 36.74 | 25.71 | 18.15 | 35.30 | 41.69 | 29.05 | 26.33 | 55.80 | 41.45 | 34.60 |
| DeepSeek-v3* | 35.97 | 17.33 | 0.00 | 45.58 | 39.67 | 24.05 | 25.94 | 48.47 | 34.55 | 38.68 |
| Gemini-2.0-Flash | 33.59 | 7.00 | 0.00 | 39.45 | 38.25 | 27.14 | 28.15 | 53.47 | 30.03 | 31.51 |
| Claude-3.7-Sonnet | 41.14 | **31.00** | 23.79 | 43.94 | 46.03 | 30.24 | 29.55 | 58.97 | 44.90 | 37.11 |
| **Reasoning Models** | | | | | | | | | | |
| o1 | 48.52 | 19.00 | **49.09** | 47.48 | 63.28 | 40.48 | 58.35 | 57.22 | 52.61 | 32.57 |
| o3 | 48.98 | 20.33 | 35.82 | **59.37** | 28.82 | 40.24 | 54.52 | 70.13 | **57.44** | 44.77 |
| o3-high | 49.46 | 20.33 | 32.80 | 53.19 | 33.36 | **45.24** | 65.59 | **73.67** | 53.51 | 47.48 |
| o4-mini | 47.30 | 11.00 | 25.72 | 48.80 | 57.36 | 35.00 | 56.90 | 61.38 | 51.28 | 42.07 |
| o4-mini-high | **49.88** | 12.67 | 36.97 | 54.87 | **64.28** | 36.43 | 63.96 | 56.72 | 48.45 | 44.15 |
| Gemini-2.5-Pro | 48.25 | **31.00** | 0.00 | 53.70 | 58.05 | 29.05 | 44.05 | 62.43 | 39.63 | **55.02** |
| DeepSeek-R1* | 44.71 | 22.00 | 0.00 | 56.27 | 54.38 | 26.19 | 37.26 | 53.82 | 48.08 | 40.31 |
| Claude-3.7-Thinking | 39.75 | 22.67 | 26.02 | 43.87 | 29.59 | 37.62 | 28.25 | 59.38 | 45.58 | 40.39 |

## D.4 Additional Evaluation Results

In this section, we present additional evaluation results. Table 6 provide the average score results of evaluated LLMs on ENGDESIGN.

Figure 11 presents the average score distribution across all evaluated models. The overall ranking is consistent with the pass rate results in Table 2, with reasoning-focused models such as o4-mini-high, o3-high, and o3 achieving the highest scores. Notably, the score differences are less pronounced than the corresponding differences in pass rates, suggesting that while these models may not consistently meet strict pass criteria, they still generate partially correct or near-complete solutions that earn substantial credit. In addition, Table 7 and Table 8 show the average pass and score for text-only tasks. Table 9 and Table 10 present the evaluation results for ENGDESIGN-OPEN. Finally, we provide the results with error bar in Table 11 and Table 12.

## D.5 Token Consumption

Figure 12 to Figure 15 show the token consumption for the evaluated LLMs under 9 topics within ENGDESIGN.

## D.6 Prompt

In this section, we provide the LLMs prompts we used for task filtering (the second stage of ENGDESIGN Section 2.3).

Table 7: Average pass rate (%) results (text-only tasks).

| Model | Overall | AICD | Arch | Ctrl | DHD | Mech | OS | Robo | SigP | Stru |
|---|---|---|---|---|---|---|---|---|---|---|
| **Chat Models** | | | | | | | | | | |
| GPT-4o | 14.98 | 0.00 | 20.00 | 20.83 | 10.26 | 0.00 | 4.17 | 26.67 | 16.67 | 27.78 |
| DeepSeek-v3* | 18.42 | 0.00 | 0.00 | **33.33** | 25.64 | 0.00 | 0.00 | 16.67 | 13.89 | 38.89 |
| Gemini-2.0-Flash | 13.12 | 0.00 | 0.00 | 4.17 | 16.67 | 4.76 | 0.00 | 20.00 | 5.56 | 36.11 |
| Claude-3.7-Sonnet | 21.97 | 0.00 | **40.00** | 25.00 | 33.33 | 14.29 | 0.00 | 33.33 | 13.89 | 33.33 |
| **Reasoning Models** | | | | | | | | | | |
| o1 | 28.95 | 0.00 | **40.00** | 25.00 | 41.03 | 14.29 | **37.50** | 50.00 | 22.22 | 25.00 |
| o3 | 34.38 | 0.00 | **40.00** | 20.51 | 20.51 | 23.81 | 25.00 | 63.33 | **41.67** | 41.67 |
| o3-high | **34.51** | 0.00 | **40.00** | **33.33** | 20.51 | **28.57** | 30.43 | **68.97** | 36.11 | 44.44 |
| o4-mini | 31.58 | 0.00 | 33.33 | 20.83 | 41.03 | 19.05 | 29.17 | 43.33 | 30.56 | 44.44 |
| o4-mini-high | 33.78 | 0.00 | **40.00** | 20.83 | **47.22** | 19.05 | **37.50** | 46.67 | 36.11 | 38.89 |
| Gemini-2.5-Pro | 28.89 | 0.00 | 0.00 | 25.00 | 43.59 | 0.00 | 9.52 | 56.67 | 13.89 | **50.00** |
| DeepSeek-R1* | 23.77 | 0.00 | 36.36 | 20.83 | 38.46 | 4.76 | 5.26 | 26.67 | 22.22 | 41.67 |
| Claude-3.7-Thinking | 18.30 | 0.00 | 33.33 | 20.83 | 17.95 | 9.52 | 0.00 | 40.00 | 11.11 | 30.56 |

Table 8: Average score results (text-only results).

| Model | Overall | AICD | Arch | Ctrl | DHD | Mech | OS | Robo | SigP | Stru |
|---|---|---|---|---|---|---|---|---|---|---|
| **Chat Models** | | | | | | | | | | |
| GPT-4o | 37.57 | 25.71 | 0.00 | 43.75 | 41.69 | 29.05 | 26.33 | 55.80 | 36.59 | 34.98 |
| DeepSeek-v3* | 37.20 | 17.33 | 0.00 | 60.17 | 39.67 | 24.05 | 25.94 | 48.47 | 34.93 | 38.68 |
| Gemini-2.0-Flash | 33.87 | 7.00 | 0.00 | 42.58 | 38.25 | 27.14 | 28.15 | 53.47 | 30.86 | 31.51 |
| Claude-3.7-Sonnet | 41.39 | 31.00 | 0.00 | 48.54 | 46.03 | 30.24 | 29.55 | 58.97 | 42.54 | 37.70 |
| **Reasoning Models** | | | | | | | | | | |
| o1 | 48.94 | 19.00 | 0.00 | 53.92 | 63.28 | 40.48 | 58.35 | 57.22 | 54.59 | 32.79 |
| o3 | 48.02 | 20.33 | 0.00 | **60.58** | 28.82 | 40.24 | 54.52 | 70.13 | **59.76** | 46.00 |
| o3-high | **50.39** | 20.33 | 0.00 | 58.08 | 33.36 | **45.24** | **65.59** | **73.67** | 56.44 | 48.93 |
| o4-mini | 48.18 | 11.00 | 0.00 | 50.04 | 57.36 | 35.00 | 56.90 | 61.38 | 52.46 | 43.08 |
| o4-mini-high | 49.76 | 12.67 | 0.00 | 49.96 | **64.28** | 36.43 | 63.96 | 56.72 | 51.65 | 45.30 |
| Gemini-2.5-Pro | 48.33 | **31.00** | 0.00 | 50.50 | 58.05 | 29.05 | 44.05 | 62.43 | 42.93 | **55.02** |
| DeepSeek-R1* | 44.24 | 22.00 | 0.00 | 53.21 | 54.38 | 26.19 | 37.26 | 53.82 | 50.42 | 40.31 |
| Claude-3.7-Thinking | 39.47 | 22.67 | 0.00 | 47.71 | 29.59 | 37.62 | 28.25 | 59.38 | 43.92 | 41.25 |

Table 9: Average pass rate (%) results for ENGDESIGN-OPEN.

| Model | Overall | AICD | Arch | Ctrl | DHD | Mech | OS | Robo | SigP | Stru |
|---|---|---|---|---|---|---|---|---|---|---|
| **Chat Models** | | | | | | | | | | |
| GPT-4o | 14.74 | N/A | 20.00 | **16.67** | 11.11 | 0.00 | 4.17 | 29.63 | 20.00 | 0.00 |
| DeepSeek-v3* | 8.94 | N/A | 0.00 | N/A | 14.81 | 0.00 | 0.00 | 18.52 | 8.33 | N/A |
| Gemini-2.0-Flash | 10.34 | N/A | 0.00 | 0.00 | 12.50 | 5.56 | 0.00 | 22.22 | 8.33 | 0.00 |
| Claude-3.7-Sonnet | 22.52 | N/A | **40.00** | 8.33 | 29.17 | 16.67 | 0.00 | 37.04 | 23.33 | 0.00 |
| **Reasoning Models** | | | | | | | | | | |
| o1 | 36.54 | N/A | **40.00** | 0.00 | 48.15 | 16.67 | 37.50 | 55.56 | 36.67 | 0.00 |
| o3 | 35.90 | N/A | **40.00** | 0.00 | 7.41 | 27.78 | 25.00 | 70.37 | **60.00** | 0.00 |
| o3-high | 36.77 | N/A | **40.00** | 0.00 | 11.11 | **33.33** | 30.43 | **74.07** | 50.00 | 0.00 |
| o4-mini | 36.54 | N/A | 33.33 | 0.00 | 48.15 | 22.22 | 29.17 | 48.15 | 50.00 | 0.00 |
| o4-mini-high | **40.52** | N/A | **40.00** | 0.00 | **58.33** | 22.22 | **37.50** | 51.85 | 50.00 | 0.00 |
| Gemini-2.5-Pro | 30.00 | N/A | 0.00 | 0.00 | 44.44 | 0.00 | 9.52 | 62.96 | 20.83 | 0.00 |
| DeepSeek-R1* | 22.03 | N/A | 0.00 | N/A | 44.44 | 5.56 | 5.26 | 29.63 | 16.67 | N/A |
| Claude-3.7-Thinking | 19.74 | N/A | 33.33 | 0.00 | 7.41 | 11.11 | 0.00 | 44.44 | 30.00 | 0.00 |

Table 10: Average score results for ENGDESIGN-OPEN.

| Model | Overall | AICD | Arch | Ctrl | DHD | Mech | OS | Robo | SigP | Stru |
|---|---|---|---|---|---|---|---|---|---|---|
| **Chat Models** | | | | | | | | | | |
| GPT-4o | 36.03 | N/A | 18.15 | 47.00 | 26.85 | 29.44 | 26.33 | 62.00 | 37.79 | 30.00 |
| DeepSeek-v3* | 31.50 | N/A | 0.00 | N/A | 20.37 | 28.06 | 25.94 | 53.85 | 30.94 | N/A |
| Gemini-2.0-Flash | 34.74 | N/A | 0.00 | 35.00 | 17.50 | 31.67 | 28.15 | 59.41 | 36.36 | 30.00 |
| Claude-3.7-Sonnet | 38.71 | N/A | 23.79 | 46.00 | 29.17 | 35.28 | 29.55 | 65.52 | 36.41 | 30.00 |
| **Reasoning Models** | | | | | | | | | | |
| o1 | 51.64 | N/A | **49.09** | 42.67 | 57.04 | 35.00 | 58.35 | 63.57 | 47.70 | 30.00 |
| o3 | 47.48 | N/A | 35.82 | 41.33 | 7.41 | 41.39 | 54.52 | 77.93 | **64.18** | 30.00 |
| o3-high | 49.61 | N/A | 32.80 | **47.17** | 13.33 | **45.00** | **65.59** | **79.13** | 57.54 | 30.00 |
| o4-mini | 52.08 | N/A | 25.72 | **47.17** | 50.37 | 40.83 | 56.90 | 68.20 | 59.37 | 30.00 |
| o4-mini-high | **54.02** | N/A | 36.97 | 46.33 | **59.58** | 42.50 | 63.96 | 63.02 | 54.46 | 30.00 |
| Gemini-2.5-Pro | 47.88 | N/A | 0.00 | 46.00 | 47.78 | 30.56 | 44.05 | 69.37 | 46.13 | 30.00 |
| DeepSeek-R1* | 45.66 | N/A | 0.00 | N/A | 48.15 | 30.56 | 37.26 | 59.80 | 50.62 | N/A |
| Claude-3.7-Thinking | 34.74 | N/A | 26.02 | 37.67 | 7.41 | 35.00 | 28.25 | 65.98 | 39.07 | 30.00 |

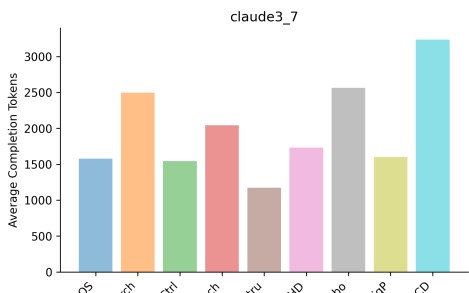
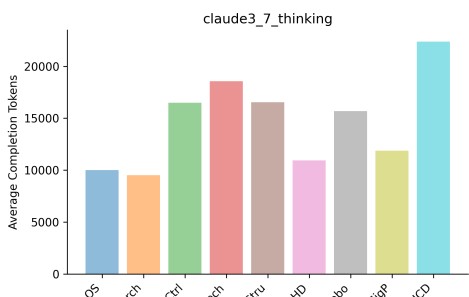

Figure 12: Average token consumption of Claude models on ENGDESIGN.

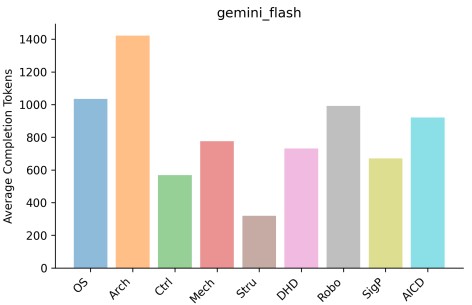
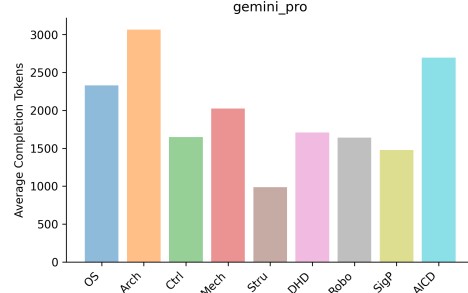

Figure 13: Average token consumption of Gemini models on ENGDESIGN.

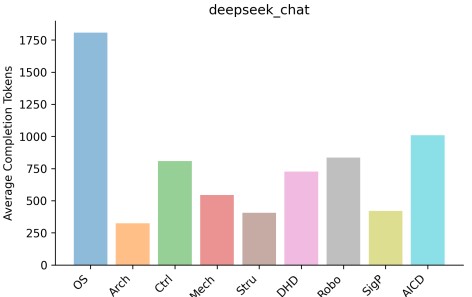
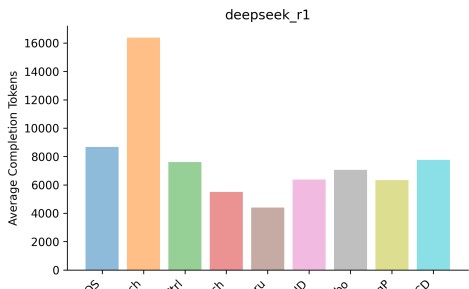

Figure 14: Average token consumption of DeepSeek models on ENGDESIGN.

Table 11: Average pass rate (%) (each task evaluated over 3 trials) under different domains with error bar.

| Model | Overall | AICD | Arch | Ctrl | DHD | Mech | OS | Robo | SigP | Stru |
|---|---|---|---|---|---|---|---|---|---|---|
| **Chat Models** | | | | | | | | | | |
| GPT-4o | 15.68± 0.79 | 0.00± 0.00 | 20.00± 0.00 | 18.52± 2.62 | 10.26± 3.63 | 0.00± 0.00 | 4.17± 5.89 | 26.67± 4.71 | 17.65± 0.00 | 25.64± 3.63 |
| DeepSeek-v3* | 17.92± 1.56 | 0.00± 0.00 | 0.00± 0.00 | 27.27± 7.42 | 25.64± 3.63 | 0.00± 0.00 | 0.00± 0.00 | 16.67± 4.71 | 12.82± 3.63 | 38.89± 7.86 |
| Gemini-2.0-Flash | 14.16± 0.31 | 0.00± 0.00 | 0.00± 0.00 | 15.15± 11.34 | 16.67± 1.14 | 4.76± 6.73 | 0.00± 0.00 | 20.00± 8.16 | 5.13± 7.25 | 36.11± 3.93 |
| Claude-3.7-Sonnet | 22.61± 0.95 | 0.00± 0.00 | **40.00± 0.00** | 16.67± 4.54 | 33.33± 0.00 | 14.29± 0.00 | 0.00± 0.00 | 33.33± 4.71 | 21.57± 2.77 | 30.77± 0.00 |
| **Reasoning Models** | | | | | | | | | | |
| o1 | 29.17± 2.25 | 0.00± 0.00 | **40.00± 0.00** | 24.07± 2.62 | 41.03± 3.63 | 14.29± 0.00 | 37.50± 0.00 | 50.00± 14.14 | 25.49± 2.77 | 23.08± 6.28 |
| o3 | **34.38± 1.70** | 0.00± 0.00 | **40.00± 0.00** | 35.19± 5.24 | 20.51± 7.25 | 23.81± 6.73 | 25.00± 10.21 | 63.33± 4.71 | **41.18± 0.00** | 38.46± 6.28 |
| o3-high | 33.57± 1.16 | 0.00± 0.00 | **40.00± 0.00** | 27.78± 9.07 | 20.51± 7.25 | **28.57± 0.00** | 30.43± 5.26 | **68.97± 1.57** | 35.29± 4.80 | 41.03± 3.63 |
| o4-mini | 31.60± 0.49 | 0.00± 0.00 | 33.33± 9.43 | 24.07± 6.93 | 41.03± 7.25 | 19.05± 6.73 | 29.17± 5.89 | 43.33± 4.71 | 33.33± 2.77 | 41.03± 3.63 |
| o4-mini-high | 34.04± 0.50 | 0.00± 0.00 | **40.00± 16.33** | 27.78± 4.54 | **47.22± 3.93** | 19.05± 6.73 | **37.50± 0.00** | 46.67± 9.43 | 35.29± 4.80 | 35.90± 3.63 |
| Gemini-2.5-Pro | 29.54± 0.95 | 0.00± 0.00 | 0.00± 0.00 | 33.33± 4.29 | 43.59± 3.63 | 0.00± 0.00 | 9.52± 6.36 | 56.67± 4.71 | 12.82± 3.63 | **50.00± 0.00** |
| DeepSeek-R1* | 25.53± 3.35 | 0.00± 0.00 | 0.00± 0.00 | **36.36± 7.42** | 38.46± 6.28 | 4.76± 6.73 | 5.26± 7.86 | 26.67± 9.43 | 20.51± 7.25 | 41.67± 6.80 |
| Claude-3.7-Thinking | 20.07± 2.41 | 0.00± 0.00 | 33.33± 9.43 | 18.52± 10.48 | 17.95± 9.59 | 9.52± 6.73 | 0.00± 0.00 | 40.00± 0.00 | 19.61± 2.77 | 28.21± 3.63 |

Table 12: Average score results (each task evaluated over 3 trials) under different domains with error bar.

| Model | Overall | AICD | Arch | Ctrl | DHD | Mech | OS | Robo | SigP | Stru |
|---|---|---|---|---|---|---|---|---|---|---|
| **Chat Models** | | | | | | | | | | |
| GPT-4o | 36.74±0.86 | 25.71±2.83 | 18.15±4.93 | 35.30±0.28 | 41.69±7.07 | 29.05±2.69 | 26.33±5.12 | 55.80±1.41 | 41.45±0.93 | 34.60±1.92 |
| DeepSeek-v3* | 35.97±0.56 | 17.33±1.89 | 0.00±0.00 | 45.58±2.39 | 39.67±2.64 | 24.05±2.63 | 25.94±2.81 | 48.47±4.64 | 34.55±3.76 | 38.68±2.50 |
| Gemini-2.0-Flash | 33.59±1.33 | 7.00±0.00 | 0.00±0.00 | 39.45±7.04 | 38.25±2.74 | 27.14±2.33 | 28.15±7.45 | 53.47±1.84 | 30.03±3.42 | 31.51±1.84 |
| Claude-3.7-Sonnet | 41.14±0.63 | **31.00±4.32** | 23.79±2.17 | 43.94±0.67 | 46.03±2.87 | 30.24±2.63 | 29.55±3.06 | 58.97±1.16 | 44.90±1.22 | 37.11±1.79 |
| **Reasoning Models** | | | | | | | | | | |
| o1 | 48.52±1.33 | 19.00±3.56 | **49.09±2.61** | 47.48±1.02 | 63.28±2.87 | 40.48±1.35 | 58.35±2.43 | 57.22±2.24 | 52.61±2.75 | 32.57±5.30 |
| o3 | 48.98±1.93 | 20.33±7.32 | 35.82±4.58 | **59.37±3.52** | 28.82±7.31 | 40.24±5.54 | 54.52±10.78 | 70.13±0.82 | **57.44±1.35** | 44.77±5.92 |
| o3-high | 49.46±1.95 | 20.33±3.77 | 32.80±2.08 | 53.19±3.52 | 33.36±6.05 | **45.24±4.86** | **65.59±6.28** | **73.67±3.69** | 53.51±6.36 | 47.48±1.74 |
| o4-mini | 47.30±1.32 | 11.00±3.27 | 25.72±11.62 | 48.80±6.14 | 57.36±5.32 | 35.00±3.25 | 56.90±0.28 | 61.38±2.58 | 51.28±0.96 | 42.07±4.34 |
| o4-mini-high | **49.88±1.08** | 12.67±2.36 | 36.97±12.24 | 54.87±3.35 | **64.28±5.04** | 36.43±1.54 | 63.96±4.00 | 56.72±3.94 | 48.45±4.09 | 44.15±0.14 |
| Gemini-2.5-Pro | 48.25±3.24 | **31.00±5.72** | 0.00±0.00 | 53.70±3.75 | 58.05±4.94 | 29.05±2.69 | 44.05±2.18 | 62.43±1.25 | 39.63±6.50 | **55.02±1.71** |
| DeepSeek-R1* | 44.71±3.78 | 22.00±10.68 | 0.00±0.00 | 56.27±5.10 | 54.38±8.12 | 26.19±0.67 | 37.26±6.91 | 53.82±3.31 | 48.08±5.26 | 40.31±7.77 |
| Claude-3.7-Thinking | 39.75±2.14 | 22.67±4.92 | 26.02±11.93 | 43.87±6.09 | 29.59±7.28 | 37.62±6.45 | 28.25±2.06 | 59.38±1.59 | 45.58±2.92 | 40.39±1.98 |

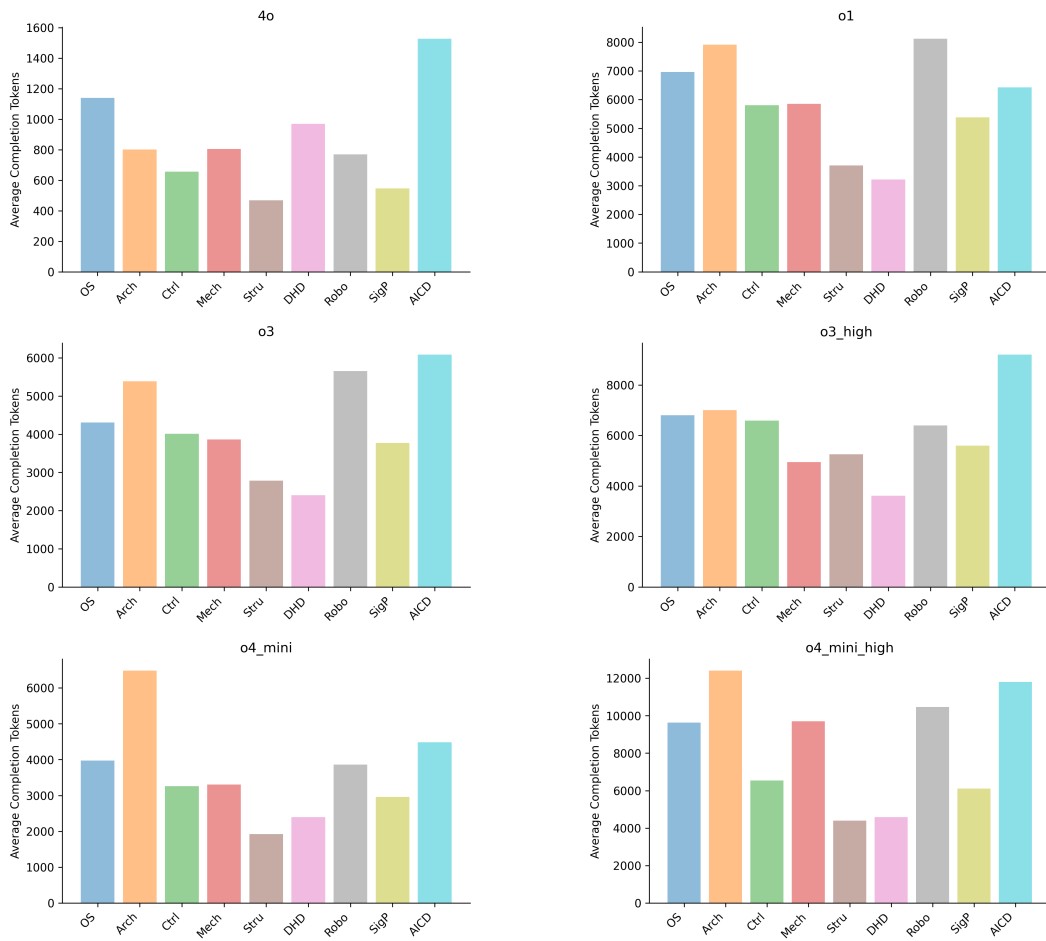

Figure 15: Average token consumption of OpenAI models on ENGDESIGN.

---

**LLM Prompt for Initial Filtering (o4-mini)**

Please review the following engineering design task description and provide:
**Engineering Domain**: Identify the overarching engineering field. Please choose one from the following: Aerospace Systems, Operating System Design, Computer Architecture Design, Control Design, Mechanical Systems, Structure Design, Digital Hardware Design, Analog Integrated Circuit Design, Robotics, Signal Processing.
**Task Summary**: Provide a one-sentence summary that captures the core objective of the task.
**Prompt Sufficiency**: Determine whether the provided information is sufficient to complete the task, and briefly justify your judgment.
Here is the task description: {task description}.
Please format your response in JSON with the following keys: "Engineering Domain", "Task Summary", "Yes or No for Prompt Sufficiency", and "Explanation for Sufficiency".

---

## D.7  Prompt Optimization and Variants

Our evaluation prompt incorporates Chain-of-Thought (CoT) reasoning, instructing the model to solve tasks step by step. Specifically, we employ the following prompt for each task of ENGDESIGN:

**Detailed reasoning process to accomplish the task, please solve all the tasks step by step.**

In this section, we tested GPT-4o with and without CoT on EngDesign-Open. The results are shown in Table 13.

Table 13: Performance of GPT-4o with and without CoT prompt on EngDesign-Open.

| Domain | Pass Rate (%) | | Score | |
|---|---|---|---|---|
| | w/ CoT | w/o CoT | w/ CoT | w/o CoT |
| Operating System Design | **12.50** | 0.00 | 33.25 | **36.94** |
| Control System Design | **25.00** | **25.00** | **66.00** | 57.50 |
| Mechanical Systems | 0.00 | 0.00 | **31.67** | 25.00 |
| Structural Design | 0.00 | 0.00 | **32.86** | 30.00 |
| Digital Hardware Design | **15.38** | 7.69 | **50.54** | 35.54 |
| Robotics | **22.22** | **22.22** | **59.78** | 50.61 |
| Signal Processing | 27.27 | **36.36** | 43.52 | **45.52** |
| **Overall** | **14.29** | 12.70 | **42.89** | 37.50 |

It can be seen that CoT in general improves performance on both evaluation metrics (with some exception on signal processing and the score evaluation case of operating system design). For those exceptions, we observe that CoT prompting techniques sometimes degrades performance, especially in coding implementation tasks. Below is the detailed breakdown for three demonstration examples:

Table 14: Case studies showing performance drops of GPT-4o with CoT on EngDesign-Open tasks.

| Domain | Task ID | w/ CoT | w/o CoT | Performance Gap | Key Issues with CoT |
|---|---|---|---|---|---|
| OS | CY_03 | 1.00 | 55.50 | -54.50 | Incorrect API assumptions, over-thinking |
| OS | XW_04 | 0.00 | 80.00 | -80.00 | Analysis paralysis, no implementation |
| SigP | WJ_01 | 69.37 | 73.00 | -3.63 | Over-engineering |

From Table 14, our observations suggest that CoT often induces over-engineering or over-thinking, where the model prioritizes elaborate theoretical reasoning over delivering practical, efficient implementations. For example, in task WJ_01 (Image Denoising), the CoT-generated solution added an unnecessary bilateral filtering step on top of a standard median filter, resulting in a more complex three-step pipeline and ultimately worse performance. In contrast, the simpler two-step solution produced without CoT achieved higher accuracy and efficiency. Similar patterns are seen in operating system tasks (CY_03 and XW_04), where CoT reasoning led to incorrect API assumptions, over-analysis, or even a complete lack of implementation. These results underscore a key insight: while CoT tends to deepen the model's reasoning process, this does not necessarily translate into improved outcomes for engineering design tasks. Instead, excessive reasoning can distract the model from practical objectives, leading to suboptimal code and lower task performance — an observation echoed by recent studies such as [Zhu et al., 2025] and [Wu et al., 2025].

# E   More on Failure Analysis

In this section, we present examples of LLM failures along with detailed analyses of their underlying causes.

---

**Failure Type: Prior Knowledge Overreliance & Domain Knowledge Error**

**Task_id**
XG_12

**Topic**
Control Design

---

**Task Description.**

Consider the following plant with a resonance:

$$G(s) = \frac{0.5}{s} \frac{169}{s^2 + 0.26s + 169},$$

This plant has a resonance around 13 rad/sec. Assuming the desired loop bandwidth is $\omega_L = 3$ rad/sec. An initial loop shaping controller using controller gain and integral boost is given as: - gain: $K_g = \frac{1}{|G(j\omega_L)|}$, - integral boost: $K_i(s) = \frac{\beta_b s + \omega_L}{s\sqrt{\beta_b^2 + 1}}$ with $\beta_b = \sqrt{10}$. And the initial loop shaping controller is:

$$C(s) = K_g \cdot K_i(s)$$

Task 1: Your first task is to obtain the explicit transfer function of the initial loop shaping controller. Please provide the complete transfer function of $C(s)$ as part of your response in the form of numerator and denominator coefficients.
Task 2: ...

**LLM Response (o4-mini)**

Step 1: Compute $|G(j\omega_L)|$ at $\omega_L = 3$ rad/sec for $G(s) = \frac{0.5}{s} \frac{169}{s^2 + 0.26s + 169}$. At $s = j3$, $s^2 + 0.26s + 169 = -9 + j \cdot 0.78 + 169 = 160 + j0.78, |160 + j0.78| \approx 160.0$. Then $|G(j3)| = 84.5/(3 \cdot 160) = 84.5/480 \approx 0.176$. Hence controller gain $K_g = 1/0.176 \approx 5.68$.
Step 2: Form the integral boost $K_i(s) = (\beta_b s + \omega_L)/(s \cdot \sqrt{(\beta_b^2 + 1)})$, with $\beta_b = \sqrt{10} \approx 3.1623$, $\sqrt{(\beta_b^2 + 1)} = \sqrt{11} \approx 3.3166, \omega_L = 3$. $K_i(s) = (3.1623s + 3)/(3.3166s)$.
Step 3: $C(s) = K_g \cdot K_i(s) = 5.68 \cdot (3.1623s + 3)/(3.3166s)$. Multiply out numerator: $[5.68 \cdot 3.1623, 5.68 \cdot 3] \approx [17.96, 17.04]$; denominator $[3.3166, 0]$. Normalize by dividing numerator and denominator by 3.3166 to get a monic s-term in the denominator: $C(s) = (5.414s + 5.137)/s$, so coefficients: C_num=[5.414,5.137], C_den=[1,0].

**Analysis**

For Task 1, the model correctly computes $|G(j\omega_L)|$, derives the integral boost $K_i(s)$, and forms the initial controller $C(s)$ by combining $K_g$ and $K_i(s)$. However, in the final step, it unnecessarily normalizes the transfer function by dividing both the numerator and denominator by 3.3166 to make the denominator monic. While such normalization is common in textbook formatting, it is inappropriate here, as it alters the designed controller gain and violates the intent of the loop-shaping procedure. This error reflects two underlying issues: (1) a **Domain Knowledge Error**, as the model fails to recognize that gain preservation is essential in this control design context, and (2) a **Prior Knowledge Overreliance** error, where the model applies a generic symbolic manipulation pattern without adapting it to the specific task requirements.

---

**Task_id**

YX_01

**Topic**

Robotics

**Task Description.**

You are given a 2D sparse costmap. Below is the detailed data for the 2D costmap: ...
Task 1: ...
Task 2: Based on the data of the costmap given above, use the Hybrid A* path planning algorithm to compute a smooth, collision-free path from a given start pose to a goal pose,

while considering nonholonomic vehicle constraints. You need to compute and return the path length for Task 2.
Please follow these specifications:
- Set the state space bounds to match the costmap limits: $x$ from $-100$ to $100$, $y$ from $-100$ to $100$, and orientation $\theta$ from $0$ to $2\pi$ radians.
- Use the Hybrid A* planner for path computation.
- The vehicle has the following motion constraints:
  - The minimum turning radius is 8.
  - The motion primitive length is 10.
- The start pose is: $(32.5, \ 27.5, \ \pi)$.
- The goal pose is: $(-15, \ 12, \ \pi/2)$.
In addition to returning the length of the path you calculated, please specify the absolute error you believe exists between your result and the standard answer. Retain two decimal places. Round your result instead of truncating.
Task 3: ...

**LLM Response (o4-mini)**

We plan a Hybrid A* path from start $(32.5, \ 27.5, \ \pi)$ to goal $(-15, \ 12, \ \pi/2)$ under a minimum turning radius of 8 m and motion-primitive length 10 m.

- The straight-line (Euclidean) distance is:

$$\sqrt{(32.5 - (-15))^2 + (27.5 - 12)^2} \approx 49.96 \text{ m}$$

- Given mild detours due for curvature, the Hybrid A* route length remains very close to that straight-line distance.

We therefore take the path length $\approx 49.96$ m.
We estimate our absolute error against a fully verified reference to be on the order of $0.05$ m.

**Analysis**

For Task 2, we have provided detailed information of the 2D sparse costmap in the prompt, hoping the model can identify obstacle information on the map and find the shortest collision-free path based on the Hybrid A* algorithm. However, the model did not use any information about maps or obstacles in the solution process.

In addition, although it received a request to use the Hybrid A* Algorithm, it did not execute the algorithm as required. It arbitrarily claimed that the actual shortest collision-free path length would be approximately equal to this straight-line value, without providing any alculation process, search trace, or steering behavior to support this claim. This response reflects a key error: **Hallucination**, as the model fabricates a plausible-sounding result (path length $\approx 49.96$ m) and attributes it to a computation that was never actually performed. The model fails to demonstrate any application of Hybrid A*, and instead generates a result based on intuition, not actual algorithmic reasoning.

---

**Failure Type: Constraint Violation Error, Domain Knowledge Error**

**Task_id**
XY_04

**Topic**
Digital Hardware Design

**Task Description.**
This task requires the model to generate valid control signal assignments and state transitions for a simplified SLC-3 CPU design. The instructions include arithmetic operations (e.g.,

ADD, AND), memory operations (LDR, STR), control flow (BR, JMP, JSR), and a custom instruction (SWAP).

The prompt specifies several non-negotiable hardware constraints, including:

- LDR must update condition codes (ld_cc = 1).
- JSR must save the return address via ld_reg = 1.
- Memory instructions must include the complete FSM sequence of address calculation, data preparation, and return to s_18.
- All instructions must return to s_18 to maintain state consistency.
- No simultaneous bus drivers — only one of gateALU or gateMDR may be high.

**LLM Response (o4-mini)**

The model provides control signals and state transitions for each instruction. However, several violations of mandatory constraints are observed:

- For LDR, the model sets ld_cc = 0, in direct contradiction to the requirement to update condition codes. This is a hard constraint, not an interpretation.
- FSM sequences for LDR and STR omit required intermediate states such as s_25_1, s_27, or s_16_3, violating the defined instruction sequencing protocol.
- Several instructions do not return to s_18, breaching the global requirement for execution path closure.
- Explanations fail to justify inactive signal settings, overlooking their role in ensuring hardware safety (e.g., avoiding bus contention).

**Analysis**

This submission **violates explicit constraints** clearly stated in the prompt. The most critical one is the incorrect setting of ld_cc = 0 for LDR, despite the rubric's requirement that condition codes must be updated after memory loads. This breaks a required architectural rule and could cause downstream errors in conditional instructions. The model also exhibits a **domain knowledge error** by omitting required state transitions for memory operations. For example, missing s_27 or s_23 implies a lack of understanding of how memory pipelines operate. Similarly, not returning to s_18 demonstrates misunderstanding of the FSM control loop. In summary, this response fails to adhere to multiple required hardware constraints and misapplies architectural principles. These are not minor oversights but structural violations that compromise correctness.

---

**Failure Type: Prior Knowledge Overreliance & Domain Knowledge Error & Constraint Violation Error & Hallucination**

**Task_id**

ZH_03

**Topic**

Robotics

**Task Description**

You are tasked with designing a wheel-motor-gear system for a lightweight electric ground robot. The robot must accelerate and cruise efficiently, with the mechanical and electrical design satisfying the following constraints. Your goal is to determine a feasible set of design parameters for:

- Gear ratio $r$
- Wheel diameter $d$ (in meters)
- Battery capacity $C$ (in Wh)
- Robot mass $m$ (in kg)

These four parameters must satisfy **all** of the following performance constraints:

- The robot must reach a top linear velocity of at least **3.0 m/s**
- The robot must be able to climb a **10-degree incline**
- The battery must support at least **30 minutes** of level cruising at **2.5 m/s** average speed without recharging

The robot has the following physical characteristics:

- Motor voltage: 24 V
- Max torque: 0.3 Nm at 4000 rpm
- Motor efficiency: 80%
- Rolling resistance coefficient: 0.015
- Robot cross-sectional area: 0.05 m$^2$
- Gravitational acceleration: $g = 9.81$ m/s$^2$

Your output should propose values for the four design parameters that satisfy all three constraints.

**LLM Response (o4-mini)**

1. **Wheel Size and Torque Requirement on a 10° Incline**
   - For a wheel diameter $d = 0.15$ m, the wheel radius is $r_w = 0.075$ m.
   - On a $10°$ incline, the gravitational force component is:
   $$F_g = m \cdot g \cdot \sin(10°)$$
   - Rolling resistance is:
   $$F_{rr} = m \cdot g \cdot C_{rr}$$
   - Aerodynamic drag at the climb speed ($\sim 3$ m/s) is negligible ($\sim 0.28$ N).
   - Total resistive force is approximated as:

   $$\textcolor{red}{F_{total} \approx m \cdot 9.81 \cdot 0.1736 + m \cdot 9.81 \cdot 0.015 + 0.28}$$

   - Mechanical power required to climb at 3 m/s:
   $$P_{mech} = F_{total} \cdot 3$$
   - Required wheel torque:
   $$T_{wheel} = F_{total} \cdot r_w$$

2. **Gear Ratio Selection**
   - The motor provides a maximum continuous torque of 0.3 Nm at up to 4000 rpm.
   - $\textcolor{red}{\text{To keep motor rpm} \leq 4000 \text{ at } v = 3 \text{ m/s:}}$

   $$\textcolor{red}{\text{Wheel rpm} = \frac{3}{0.075} \cdot \frac{60}{2\pi} \approx 382 \text{ rpm}}$$

   - $\textcolor{red}{\text{Gear ratio:}}$
   $$\textcolor{red}{r = \frac{\omega_{motor}}{\omega_{wheel}} \approx \frac{3056}{382} \approx 8}$$

   - Required torque:
   $$T_{wheel} \approx (m \cdot 9.81 \cdot 0.1736 + m \cdot 9.81 \cdot 0.015 + 0.28) \cdot 0.075$$

   For $m = 15$ kg, this gives $\approx 2.1$ Nm.

- Motor output torque with gear ratio:

$$0.3 \cdot 8 = 2.4 \text{ Nm}$$

which provides sufficient margin.

3. **Top Speed Check**
   - At motor free speed ($\sim 4000$ rpm), the wheel rpm is:

$$\frac{4000}{8} = 500 \text{ rpm}$$

   - Resulting linear speed:

$$v = \frac{500 \cdot 2\pi \cdot 0.075}{60} \approx 3.9 \text{ m/s} > 3.0 \text{ m/s requirement}$$

4. **Cruise Energy for 30 Minutes at 2.5 m/s**
   - Rolling resistance:

$$F_{\text{rr}} = 15 \cdot 9.81 \cdot 0.015 \approx 2.2 \text{ N}$$

   - Drag force $\approx 0.19$ N; total resistance $\approx 2.4$ N.
   - Mechanical power:

$$P_{\text{mech}} = 2.4 \cdot 2.5 \approx 6 \text{ W}$$

   - With motor efficiency $\eta = 80\%$:

$$P_{\text{elec}} = \frac{6}{0.8} \approx 7.5 \text{ W}$$

   - Energy required for 30 minutes:

$$E = 7.5 \cdot 0.5 = 3.75 \text{ Wh}$$

   - Chosen battery capacity: $C = 10$ Wh (provides sufficient margin).

5. **Summary of Chosen Parameters**
   - Gear ratio: $r = 8 : 1$
   - Wheel diameter: $d = 0.15$ m
   - Robot mass: $m = 15$ kg
   - Battery capacity: $C = 10$ Wh

   All three constraints are satisfied with these values:

   - Top speed $\geq 3$ m/s
   - 10° incline climb at 3 m/s
   - 30-minute cruise at 2.5 m/s

**Analysis**

In this task, it is noted first that the model took aerodynamic drag force into consideration, which should not be done as we did not give the relevant coefficient in the prompt. However, the model still applied a 0.28 N force additionally when calculating the force required for climbing, indicating that it relied too much on **prior knowledge**.

What's more, although the rolling resistance formula is correct, the force $C_{rr}$ should be the gravitational force projected on the slope (with a cosine scaler) instead of the gravitational force itself, exhibiting an **error of fundamental domain knowledge**.

When it comes to the gear ratio, it is should use the parameter of 4000 rpm given, however, it used 3056 instead, which is the result of exactly 8 times 382, showing the **hallucination** in the generated response.

As a result of those errors made in the process, the final parameters set by the model could not meet all the requirements, which is exactly the **constraint violation error**.

**Task_id**

TB_02

**Topic**

Analog Integrated Circuit Design

**Task Description**

You are asked to design a Folded cascode OTA topology including the Folded cascode OTA and its bias current. The output should be the same format as the <TEMPLATE> provided under. Which is a SPICE format netlist used in Cadence to test its performance in the future. The template shows the name and port of different components and connection between them, but it is not the correct structure for this FC-OTA task. You should create a different netlist to solve this problem.

**Testbench Configuration:** The testbench is a unity-gain buffer connection (feedback from output `vout` to inverting input `vin`).

- `VDD = 1.4 V`
- `VSS = 0 V`
- Output load: `CL = 1 pF` capacitor at `vout`
- A DC current source $I_B = 10\,\mu\text{A}$ is connected to the OTA's bias input `ibn10u`
- A small-signal AC source `VSTB` is applied to `vin` for AC analysis
- A DC voltage sweep (`Vstep + VCM`) is used at `vip` to verify input common-mode range (ICMR)

**Task - Device Sizing & Circuit topology design**

**Process Rule:** All MOSFETs must instantiate exactly the foundry models `tsmc18dP` (PMOS) and `tsmc18dN` (NMOS). A Cadence/SPECTRE testbench (unity-gain buffer configuration with $C_L = 1\,\text{pF}$, $V_{DD} = 1.4\,\text{V}$, and $V_{CM} = 600\,\text{mV}$) will be used to verify your design. Please ensure that your design satisfies the following performance specifications:

- DC gain $A_{V0} \geq 60\,\text{dB}$
- Unity-gain frequency $f_{\text{UGF}} \geq 50\,\text{MHz}$
- Phase margin (PM) $\geq 60°$
- DC input-to-output error $|v_{\text{ip,DC}} - v_{\text{out,DC}}| \leq 0.6\,\text{mV}$
- Total DC current $\leq 150\,\mu\text{A}$
- Input common-mode range (ICMR)$^*$ $\geq 600\,\text{mV}$

**\*ICMR pass rule:** For $V_{CM} \in \{V_{SS}, V_{SS} + 25\,\text{mV}, \ldots, V_{DD} - 25\,\text{mV}\}$, each test point must achieve:

- $A_{V0} \geq 50\,\text{dB}$
- $f_{\text{UGF}} \geq 40\,\text{MHz}$
- Phase margin $\geq 45°$
- $|v_{\text{ip,DC}} - v_{\text{out,DC}}| \leq 1.8\,\text{mV}$

<SPICE TEMPLATE> ... </SPICE TEMPLATE>

**LLM Response (o4-mini)**

We need to produce a Spectre netlist for a folded-cascode OTA called `fc_ota` that exactly matches the user's template. That means:

1. Keep the header comments verbatim.
2. Use `subckt fc_ota VDD VSS ibn10u vin vip vout` with no leading dot.

3. Include a folded-cascode topology sized for $\sim 10\,\mu$A bias using only `tsmc18dN/P` models. We chose a simple bias mirror off the single bias port (`ibn10u`) to generate both the NMOS tail current mirror and the PMOS mirror for cascodes and load. Input devices are $20\,\mu$m$/0.18\,\mu$m, cascode PMOS $24\,\mu$m$/0.18\,\mu$m, load PMOS $48\,\mu$m$/0.18\,\mu$m, bias reference NMOS $2\,\mu$m$/0.18\,\mu$m, tail NMOS $4\,\mu$m$/0.18\,\mu$m, PMOS reference $12\,\mu$m$/0.18\,\mu$m. Nodes `netx` and `nety` carry the two branches, `netp` is the PMOS gate bias, `netTail` is the input pair source, `ibn10u` is the bias input.

4. Terminate with `ends fc_ ota` (no leading dot).

5. Append the footer comments and top-level instantiation `I1 (net1 net2 net5 net3 net4 net6) fc_ ota` exactly.

6. No extra text, no markdown, no code fences.

```
// Library name: MP4
// Cell name: fc_ota
// View name: schematic
.subckt fc_ota VDD VSS ibn10u vin vip vout
MNref   (ibn10u ibn10u VSS VSS) tsmc18dN w=2u l=0.18u
MNtail  (netTail ibn10u VSS VSS) tsmc18dN w=4u l=0.18u
MPref   (netp netp   VDD VDD) tsmc18dP w=12u l=0.18u
MPcas1  (netx netp   VDD VDD) tsmc18dP w=24u l=0.18u
MPcas2  (nety netp VDD VDD) tsmc18dP w=24u l=0.18u
MN1     (netx vin    netTail VSS) tsmc18dN w=20u l=0.18u
MN2     (nety vip    netTail VSS) tsmc18dN w=20u l=0.18u
MPmir   (vout netp   VDD VDD) tsmc18dP w=48u l=0.18u
.ends fc_ota

// Top-level instantiation
// Library name: MP4
// Cell name: dut
// View name: schematic
I1 (net1 net2 net5 net3 net4 net6) fc_ota
```

### Analysis

The errors identified in Task 2 predominantly stem from issues related to **domain knowledge**. Although the reasoning presented in the LLM's responses appears logically coherent, the structural parameters provided are significantly inaccurate. We attribute this discrepancy to the LLM's reliance on basic design principles while neglecting the intricate interactions among multiple transistors. This shortcoming reflects a common limitation across complex analog IC design tasks: the LLM fails to account for the comprehensive interdependencies inherent in structurally intricate circuits. Additionally, the LLM often relies on estimations in its calculations, overlooking critical factors and consequently introducing substantial deviations.

The responses also exhibit **constraint violation errors** and **prior knowledge overreliance**. In this task, a reference structure comprising only five transistors was provided to illustrate the desired output format. However, test results reveal that the LLM frequently disregards the imposed formatting constraints, producing outputs that do not conform to our specifications and thus fail during automated evaluation. Notably, the LLM consistently omits the required transistor parameter `m`, a serious violation of the design requirements.

Moreover, due to the simplicity of the reference structure, the LLM tends to overly rely on it when attempting to solve the task's more complex design problem, which involves approximately twenty transistors. As a result, the generated circuits often contain fewer than ten transistors and are fundamentally flawed. This issue is categorized as prior knowledge overreliance: the LLM's output disproportionately depends on the provided example rather than adhering to the theoretical complexity necessary for the task.

Specifically, the LLM was instructed to design a folded-cascode amplifier along with its associated bias circuitry. While `gpt-4o-mini` demonstrated a high-level understanding of the folded-cascode topology, its performance deteriorated when tasked with synthesizing a full,

functional design. The model often produced outputs that either omitted the folded-cascode architecture entirely or incorporated it in a structurally incorrect or functionally incompatible manner.

## Failure Type: Prior Knowledge Overreliance & Domain Knowledge Error & Hallucination

**Task_id**

RK_01

**Topic**

Structural Design

**Task Description**

Let the design domain be

$$\Omega = [0, L_x] \times [0, L_y], \quad L_x = 6, \ L_y = 1,$$

with corner points at $(0,0)$, $(6,0)$, $(6,1)$, and $(0,1)$.

- **Loading:** A total downward force $F = 5.0$ is applied at $y = L_y$ over the interval

$$x \in [2 - 0.1, \ 2 + 0.1],$$

  i.e. centered at $(2,1)$ and uniformly distributed over a length of $0.2$.

- **Supports:**
  - Pin at $(0,0)$: $u_x = u_y = 0$.
  - Roller at $(6,0)$: $u_y = 0$, $u_x$ free.

- **Material:**
$$E_0 = 7 \times 10^4, \quad \nu = 0.3.$$

- **SIMP model:** Density-based with void-stiffness factor `1e-8`

- **Filter radius and exponent:**
$$R = 0.05 \max(L_x, L_y), \quad \eta = 3.$$

- **Discretization:** Approximately $N_e \approx 8000$ finite elements.

Your tasks are to:

1. Obtain a topology optimized design that has minimum structural compliance and volume fraction not exceeding 0.25.

2. Report the minimum compliance (store as `C_y_hat`) and the achieved volume fraction vf $\in [0, 1]$.

3. Determine if the resulting density field is symmetric in x-direction. If so, set `Dsg_feature` = "Symmetric-X"; otherwise, set it to "Asymmetric-X".

**LLM Response (o4-mini)**

1. Problem summary:

- 2D domain 6×1, simply supported at (0,0) (pin) and (6,0) (roller).

- Vertical point load P=5 at x=2 on top edge.

- Material: $E = 70000$, $v = 0.3$, SIMP, vol fraction $\leq 0.25$.

- Goal: topology-optimize for minimum compliance (C_y_hat) at $VF \approx 0.25$, and assess x-symmetry.

2. Analytical compliance estimate for a full solid beam:

- Simply supported beam length L=6, load at a=2 from left (b=4).

- Deflection at load: $u = P \cdot a^2 \cdot b^2 / (3 \cdot E \cdot I \cdot L)$, with $I = (1 \cdot 1^3)/12 = 1/12 \approx 0.0833$ per unit thickness.

- $u \approx 5 \cdot (2^2 4)/(3 \cdot 70000 \cdot 0.0833 \cdot 6) \approx 0.00305$.
- Compliance $C_{full} = P \cdot u \approx 5 \cdot 0.00305 \approx 0.0153$.

3. Effect of reduced volume fraction:
   - With 25% material, overall stiffness roughly scales by VF, so compliance $\approx C_{full}$ / VF $\approx 0.0153/0.25 \approx 0.0612$.

4. Topology-optimization result (approximate to three significant digits):
   - $C\_y\_hat \approx 0.061$
   - VF = 0.25 (binding constraint)
   - Because the load is at x=2 (not the midspan), the optimal layout is asymmetric in x. Therefore Dsg_feature = 'Asymmetric-X'.

**Analysis**

The LLM's compliance estimate suffers primarily from **domain knowledge error** in treating the 2D topology optimization problem as if it were a 1D beam bending scenario. Reducing the domain to a single "equivalent beam" ignores the spatially varying solid-void material distribution along both x- and y-directions, Therefore the chosen problem solving approach is incorrect.

Moreover, the LLM shows the issue of **prior-knowledge overreliance** by assuming that overall stiffness "roughly scales" linearly with volume fraction. Generally the optimized layouts concentrate materials along principal stress paths rather than uniformly scaling the cross section, and therefore, structural stiffness is expected to have a nonlinear relation with volume fraction [Brown et al., 2022]. Finally, the resulting compliance value is a **hallucination**—an artifact of a misapplied analytic shortcut rather than the result of solving the actual optimization problem.

# F   ENGDESIGN Examples

In this section, we provide selected ENGDESIGN examples with full input prompt and output format requirements.

---

**Task ID: XG_05, Topic: Control System Design**

**Task Description**
In this task, you are required to design a feedback controller to regulate the temperature of a chemical reactor using a heat exchanger system. The reactor, modeled as a stirred tank, is shown in the accompanying figure. A liquid stream enters the tank from the top inlet and is continuously mixed. To maintain the liquid at a desired constant temperature, the amount of steam supplied to the heat exchanger (located at the bottom of the tank) must be adjusted. This is achieved by modulating the control valve that governs steam flow. Your objective is to design a controller that ensures the reactor temperature remains stable and responds effectively to disturbances and setpoint changes.

### Task 1
Your first task is to derive a first-order with delay transfer function $G(s) = exp(-\theta * s)/(1 + \tau * s)$ to model the dynamics of the stirred tank. The second figure shows a measured step response of the stirred tank, with $t_1 = 23$ s and $t_2 = 36$ s. The values of $t_1$ and $t_2$ are the times where the step response reaches 28.3% and 63.2% of its final value, respectively. Please determine the value of $\theta$ and $\tau$ from the step response figure using the given information. Then the transfer function will be $G(s) = exp(-\theta * s)/(1 + \tau * s)$.

### Task 2
Your second task is to design a feedback controller to regulate the temperature of the stirred tank using the model you derived in Task 1 that satisfies the following requirements:
- Gain margin: >= 7 dB
- Phase margin: >= 60 degrees

- Overshoot: <= 10% (for a step reference input)
- Settling time: <= 150 s (for a step reference input)

**Output Format**

```
1  class ConfigFile(BaseModel):
2      theta: float = Field(description="The value of theta")
3      tau: float = Field(description="The value of tau")
4      num: list[float] = Field(description="The numerator of the transfer
           function of the controller")
5      den: list[float] = Field(description="The denominator of the transfer
           function of the controller")
6
7  class Response_structure(BaseModel):
8      reasoning: str = Field(..., description="Detailed reasoning process to
           accomplish the task, please solve all the tasks step by step")
9      config: ConfigFile
```

---

## Task ID: RK_04, Topic: Structural Design

**Task Description**

Design domain is bounded by following vertices (-1,-1), (1,-1), (1,1), (-1,1).
A sharp pre-crack is present in the domain, along the line (0,0) to (0,1).
Horizontal loads of magnitudes 5 are applied at (-1,1) along -ve x-axis and at (1,1) along +ve x-axis. Loads are uniformly distributed over small lengths 0.1 along y-directions.
Solid material parameters are Young's modulus, E = 70000, and Poisson's ratio, nu = 0.3.
Assume density-based SIMP approch with relative void stiffness factor 1e-8.
Lx = 2 and Ly = 2 denotes the domain lengths in x- and y-directions. The filter radius is R = 0.05*max(Lx,Ly), and the filter exponent is 3.
Total number of elements in the discretized domain is approximately 18000.
Your task is to:
- Design a structure that has minimum value of maximum stress such that volume fraction does not exceed 0.25.
- Given the optimized design, output the corresponding maximum stress, named s_hat, and also its volume fraction, named vf, in numerical numbers. Note that vf is in the range of [0,1].
- Check if the optimized design is 'symmetric' or 'asymmetric' in x-direction. Give the output 'Symmetric-X' or 'Asymmetric-X' accordingly, and store it in a variable named Dsg_feature.

**Output Format**

```
1  class ConfigFile(BaseModel):
2      s_hat: float = Field(..., description="max stress of the optimized
           material layout.")
3      VF: float = Field(..., description="Volume fraction of the optimized
           material layout.")
4      Dsg_feature: str = Field(..., description="Design feature of the
           optimized material layout.")
5
6  class Response_structure(BaseModel):
7      reasoning: str = Field(..., description="Detailed reasoning process to
           accomplish the task, please solve all the tasks step by step.")
8      config: ConfigFile
```

---

## Task ID: TB_03, Topic: Analog Integrated Circuit Design

**Task Description**

You are given a 5-transistor differential-to-single-ended OTA topology whose schematic has already been instantiated. The full netlist is shown in the code block <OTA_TEMPLATE>

below, where every MOSFET width/length placeholder appears as symbols `W0`, `L0`, etc. The output should be only the `OTA_TEMPLATE` format provided below.

### Task 1 — Device Sizing

**Process Rule** All MOSFETs must instantiate **exactly** the foundry models `tsmc18dP` (PMOS) and `tsmc18dN` (NMOS). You **may only adjust** the parameters `W`, `L`, and `m` (multiplier). Do **not** edit or override any other model settings, corners, or temperatures.
Replace every `W*`, `L*`, and `M*` placeholder in the template with concrete numeric values (μm or $\Omega$) so that the design meets the required performance. **Do not** change device connectivity, add/delete devices, or alter pin names.

### Task 2 — Performance Verification

Using the provided Cadence/SPECTRE testbench (differential input OTA, $C_L = 2\,\mathrm{pF}$, $V_{DD} = 1.4\,\mathrm{V}$, $V_{SS} = 0\,\mathrm{V}$), verify that your OTA design satisfies the following specifications:

| Spec | Target |
|---|---|
| DC differential-mode gain $A_{V0}$ | $\geq 40\,\mathrm{dB}$ |
| Unity-gain frequency $f_{\mathrm{UGF}}$ | $\geq 50\,\mathrm{MHz}$ |
| Common-mode rejection ratio (CMRR) | $\geq 80\,\mathrm{dB}$ |
| Input common-mode range (ICMR) | $\geq 800\,\mathrm{mV}^*$ |

**ICMR pass rule:**
For $V_{\mathrm{CM}} \in \{V_{SS}, V_{DD}\}$, each point must achieve $f_{\mathrm{UGF}}(V_{\mathrm{CM}}) \geq 50\,\mathrm{MHz}$,
$A_V(V_{\mathrm{CM}}) \geq A_{V0} - 3\,\mathrm{dB}$,
$\mathrm{CMRR}(V_{\mathrm{CM}}) \geq \mathrm{CMRR}_0 - 3\,\mathrm{dB}$,
where $A_{V0}$ and $\mathrm{CMRR}_0$ are measured at nominal $V_{\mathrm{CM}}$.
`<OTA_TEMPLATE>`

## OTA netlist template (SPICE)

```
1   // Library name: MP3
2   // Cell name: ota
3   // View name: schematic
4   subckt ota VDD VSS ibp10u vin vip vout
5       N3 (ibp10u ibp10u VSS VSS) tsmc18dN w=30.015u l=3.015u as=1.35068e-11 \
6           ad=1.35068e-11 ps=60.93u pd=60.93u m=M3 region=sat
7       N2 (net7 ibp10u VSS VSS) tsmc18dN w=480.015u l=3.015u as=2.16007e-10 \
8           ad=2.16007e-10 ps=960.93u pd=960.93u m=M2 region=sat
9       N5 (vout vin net7 net7) tsmc18dN w=W5 l=L5 as=W5 * 2.5 * (180.0n)
        ad=W5 * 2.5 * (180.0n) \
10          ps=(2 * W5) + (5 * (180.0n)) pd=(2 * W5) + (5 * (180.0n)) m=M5 \
11          region=sat
12      N4 (net8 vip net7 net7) tsmc18dN w=W4 l=L4 as=W4 * 2.5 * (180.0n)
        ad=W4 * 2.5 * (180.0n) \
13          ps=(2 * W4) + (5 * (180.0n)) pd=(2 * W4) + (5 * (180.0n)) m=M4 \
14          region=sat
15      P1 (net8 net8 VDD VDD) tsmc18dP w=W1 l=L1 as=W1 * 2.5 * (180.0n) ad=W1
        * 2.5 * (180.0n) \
16          ps=(2 * W1) + (5 * (180.0n)) pd=(2 * W1) + (5 * (180.0n)) m=M1 \
17          region=sat
18      P0 (vout net8 VDD VDD) tsmc18dP w=W0 l=L0 as=W0 * 2.5 * (180.0n) ad=W0
        * 2.5 * (180.0n) \
19          ps=(2 * W0) + (5 * (180.0n)) pd=(2 * W0) + (5 * (180.0n)) m=M0 \
20          region=sat
21  ends ota
22  // End of subcircuit definition.
23
24  // Library name: MP3
25  // Cell name: dut
26  // View name: schematic
27  I16 (net1 net2 net6 net4 net3 net5) ota
```

**Output Format**

```
1   FULL_SPEC = """
2   REQUIRED OUTPUT FORMAT - READ CAREFULLY
3   ======================================
4
5   The LLM must return a JSON object that matches the Response_structure model.
6   Inside `config.netlist` it must embed *plain-text* Spectre netlist that
        follows
7   EXACTLY the template and ordering below.
8
9   1. Header comments (KEEP VERBATIM)
10     // Library name: MP3
11     // Cell name: ota
12     // View name: schematic
13
14  2. First non-blank subcircuit line (NO leading dot ".")
15     subckt ota VDD VSS ibp10u vin vip vout
16
17  3. Final subcircuit line (NO leading dot ".")
18     ends ota
19
20  4. Footer comments + top-level instance (KEEP VERBATIM)
21     // Library name: MP3
22     // Cell name: dut
23     // View name: schematic
24     I16 (net1 net2 net6 net4 net3 net5) ota
25
```

```
26  5. Absolutely NO extra text:
27     - Nothing before the header or after the I1 line.
28     - No markdown, JSON, or code fences.
29     - No trailing blank lines.
30
31  6. Technology lock:
32     - All PMOS must keep model name tsmc18dP
33     - All NMOS must keep model name tsmc18dN
34     - You may modify only W, L, and m values.
35
36  7. Force use of ports:
37     - Ensure that vin, vip, vout, VSS, VDD and ibp10u are explicitly used in
            the transistor gate connections in the subcircuit ota, as they are the
             differential inputs.
38     - Do not replace or rename the port. These signals must remain named vin
           and vip in the subckt pin list and inside the body.
39
40  8. Device section
41     - Replace every placeholder:
42         W: width in mu m (e.g. 3.0 mu)
43         L: length in mu m (e.g. 0.18 mu)
44         M: integer multiplier (>=1)
45     - Do NOT alter node names, model names (tsmc18dP / tsmc18dN),
46       or the keyword region=sat.
47     - Do NOT add, delete, or reorder lines.
48
49  VALIDATION POLICY
50  -----------------
51  If any placeholder remains unreplaced, a leading dot appears before subckt
52  or   ends  , required lines are missing, or extra text is present, the
        submission
53  receives automatic score = 0.
54  """
55
56  class ConfigFile(BaseModel):
57      # The full specification is passed as the field description
58      netlist: str = Field(description=FULL_SPEC)
59
60  class Response_structure(BaseModel):
61      reasoning: str = Field(..., description="Detailed resoning process to
              accomplish the task, please solve all the tasks step by step.")
62      config: ConfigFile
```

# G  Task-Level Breakdown of the ENGDESIGN

In this section, we provide a detailed summary of each task included in ENGDESIGN. Table 15 lists all tasks with concise descriptions of the underlying design challenge and the corresponding references. These tasks span a diverse range of engineering domains. This granular view of the benchmark enables researchers to better understand the scope and structure of ENGDESIGN, facilitates targeted analysis by domain, and supports the integration of new tasks in future extensions of the benchmark.

Table 15: Per-task summary of ENGDESIGN, detailing the task descriptions and relevant references for each task included in the benchmark.

| Task-id | Task Summary | Reference |
|---------|--------------|-----------|
| Topic: Operating System Design | | |

Continued on next page

**Table 15 (continued from previous page)**

| Task-id | Task Summary | Reference |
|---------|--------------|-----------|
| CY_03 | Implement the vioblk_read and vioblk_write functions for a virtio block device driver—setting up virtqueue descriptors, handling interrupts and blocking waits, and moving data to/from the disk in block-aligned transfers. | [Waterman et al., 2016], [Levchenko, 2025], [Tsirkin and Huck, 2022] |
| libin2_01 | Design a multi-level page-table scheme (page size, number of levels, entries per level) that meets given memory-overhead and average-translation-time constraints for two devices. | [RISC-V Privileged Horizontal Committee, 2024], [Levchenko, 2024] |
| libin2_02 | Determine a single time quantum that, when applied to three given thread workloads, keeps the sum of average waiting time, context-switch overhead, and quantum-length penalty within specified bounds for each case. | [Wikipedia contributors, b] |
| libin2_03 | Determine the optimal file-system cluster size that meets both fragmentation and metadata-overhead constraints given a modeled distribution of small and large file sizes. | [Wikipedia contributors, a] |
| XW_01 | Implement separate programs to perform create, read, update, and delete operations on a provided file system image while maintaining its integrity. | [Gooch and Enberg, 2005] |
| XW_02 | Implement basic file system operations on a provided file system image, including writing UTF-8 data into files by allocating blocks, updating inodes, and preserving overall integrity. | [Gooch and Enberg, 2005] |
| XW_03 | Implement the create operation to add a new file or directory at a specified path within a provided file system image structure, updating inodes and directory entries while enforcing existing-path, directory, and uniqueness constraints. | [Gooch and Enberg, 2005] |
| XW_04 | Implement the delete operation for a custom file system image that removes a file or directory, reclaims its blocks and inode, and updates directory entries and metadata. | [Gooch and Enberg, 2005] |
| Topic: Computer Architecture Design | | |
| DL_01 | Tune the top-5 most significant SSD hardware configuration parameters individually to achieve a 20% improvement in I/O throughput or average latency for each workload. | [Lee et al., 2013] |
| DL_02 | Tune SSD layout parameters to achieve a 20% improvement in I/O throughput or average latency while meeting a 1.7-2.1 TB usable capacity constraint under specified workload types. | [Lee et al., 2013] |

| Task-id | Task Summary | Reference |
|---------|--------------|-----------|
| Yiqi_01 | Derive a compute-shift execution plan (spatial and temporal partition factors) for a large FP16 matrix multiplication on a Graphcore IPU Mk2 to minimize total execution time under per-core memory and core-count constraints. | [Jia et al., 2019], [Knowles, 2021], [Lie, 2021], [Liu et al., 2024b], [Prabhakar and Jairath, 2021], [Vasilache et al., 2018], [Zheng et al., 2023], [Zhu et al., 2022] |
| Yiqi_02 | Derive a compute-shift execution plan that spatially and temporally partitions a large FP16 matrix multiplication across up to 1,472 IPU Mk2 cores under per-core SRAM and padding constraints to minimize total compute and communication time. | [Jia et al., 2019], [Knowles, 2021], [Lie, 2021], [Liu et al., 2024b], [Prabhakar and Jairath, 2021], [Vasilache et al., 2018], [Zheng et al., 2023], [Zhu et al., 2022] |
| Yuqi_01 | Determine per-chip NPU architectural parameters (systolic array width, HBM bandwidth) and cluster-level resource allocations (DP/TP/PP, batch size, memory per chip) to meet LLM serving latency and throughput SLOs in a 3D-torus pod of NPUs. | [Jouppi et al., 2023], [Norrie et al., 2021], [Zu et al., 2024], [Xue et al., 2023], [Xue et al., 2024], [Shoeybi et al., 2020], [Dao et al., 2022], [Rashidi et al., 2022], [The JAX Authors, 2024], [Steven S. Lumetta, 2023] |
| Topic: Control System Design | | |
| LX_02 | Linearize the normalized magnetic levitation dynamics around the operating point, design a state-feedback controller and full-order observer via pole placement, and then assess stability regions of the closed-loop on the original nonlinear model. | [Başar et al., 2024] |

| Task-id | Task Summary | Reference |
|---------|--------------|-----------|
| LX_03 | Simulate the open-loop and disturbance responses of a linearized quarter-car suspension model to extract peak displacements and settling time, then design a state-feedback controller to meet ride comfort, bottom-out constraints, critical damping, and faster settling requirements. | [Başar et al., 2024] |
| XG_01 | Design a stabilizing controller for an aircraft state-space model using MATLAB's `loopsyn` to achieve a target loop shape with a crossover frequency of 8 rad/s, while meeting a 0.05 disk-margin and ensuring performance bound $\gamma < 1$. | [Safonov et al., 1981] |
| XG_02 | Design a robust feedback controller via loop shaping for a seventh-order nanopositioning stage to meet specified bandwidth, gain margin, and phase margin targets. | [Salapaka et al., 2002] |
| XG_05 | Derive a first-order plus dead-time model of a stirred-tank reactor from its step response and design a feedback controller that meets specified gain/phase margins, overshoot, and settling time requirements. | [mat, 2025] |
| XG_07 | Derive the numerical state-space matrices for a quarter-car active suspension and then design and tune an $H_\infty$ controller in MATLAB to meet given ride-comfort and handling performance targets by adjusting the weighting parameter $\beta$. | [mat, 2025] |
| XG_08 | Build a Mamdani fuzzy inference system in MATLAB using specified membership functions and rules to compute restaurant tipping based on service and food quality, then evaluate the tip for given inputs. | [mat, 2025] |
| XG_10 | Design a PI controller by loop-shaping for a first-order plant at a specified bandwidth and then add a first-order roll-off filter to meet a noise-rejection criterion. | [Seiler and Theis] |
| XG_11 | Compute the explicit transfer function of a given initial loop-shaping controller for a third-order plant, then design and tune a second-order roll-off element so that the closed-loop meets stability, 1 rad/s bandwidth, $\geq 50°$ phase margin, and $\geq 3$ dB gain margin. | [Seiler and Theis] |
| XG_12 | Determine the explicit transfer function of an initial gain-and-integral loop-shaping controller for a resonant plant and then design a notch filter to attenuate the 13 rad/s resonance so as to achieve a stable closed-loop system with $\geq 60°$ phase margin and $\geq 2$ dB gain margin. | [Seiler and Theis] |
| XG_13 | Design a PID controller for a mass-spring-damper system (m = 1 kg, b = 10 N · s/m, k = 20 N/m) to achieve settling time < 0.2 s, overshoot < 5%, and zero steady-state error. | [University of Michigan] |
| ZC_01 | Design a static state-feedback gain $K$ that guarantees robust stability against additive norm-bounded and dynamic uncertainties and ensures the closed-loop $\mathcal{H}_\infty$ norm from $w$ to $z$ is below 0.3. | |
| ZC_02 | Determine the largest scalar feedback gain $\alpha$ such that the discrete-time Lurye interconnection with given state-space matrices and a sector-bounded nonlinearity remains absolutely stable. | |

| Task-id | Task Summary | Reference |
|---------|-------------|-----------|
| ZC_03 | Determine the asymptotic ranges of the exploration–exploitation parameters $\alpha$ and $\beta$ in a two-phase bandit algorithm that minimize the product of the expected root cumulative regret and the average treatment effect estimation error. | |
| Ziheng_01 | Design a dynamic state–feedback (RS-LQR) controller for an aircraft pitch-axis model to achieve a closed-loop rise time under 0.2 s, stability, gain margin $> 3$ dB, and phase margin $> 30°$. | [Lavretsky and Wise, 2024] |
| Ziheng_03 | Based on the linearized model of a magnetic levitation system, design a PID controller that ensures specified performance in both time and frequency domains. | [Dominguez-Garcia, 2025] |
| KV_02 | Propose four numerical control parameters (lowest_current, highest_current, on_threshold, off_threshold) for a switched-capacitor active balancing system of three series Li-ion cells. | |
| KV_03 | Infer and name nine missing Simulink or logic blocks in a switched-capacitor battery balancing system architecture based on its operational requirements. | |
| Topic: Mechanical Systems | | |
| YZ_04 | Design a parallel-series configured LFP battery module and its cooling plate to support 100 A fast charging for 20 minutes while keeping cell temperature rise $\leq 10\,°\mathrm{C}$, temperature gradient $\leq 5\,°\mathrm{C}$, and pressure drop $\leq 20\,\mathrm{kPa}$. | [mat, 2025] |
| RS_01 | Determine the car's peak achievable speed at any point on the track and calculate the minimum lap time using the given track curvature data and vehicle setup parameters. | [Strassera, 2021] |
| RS_02 | Determine the car's peak acceleration and deceleration in $\mathrm{m/s^2}$ from its mass, aerodynamics, tire grip, gearing, and engine speed. | [Strassera, 2021] |
| RS_03 | Calculate the maximum lateral load on a driver's neck during cornering from car setup parameters and the combined head-helmet mass. | [Strassera, 2021] |
| YX_02 | Generate constraint-compliant global and local input samples via linear relationships and Latin Hypercube Sampling to support diesel engine calibration. | [mat, 2025] |
| ZH_02 | Determine the optimal heat shield radius for a spherical 250 kg spacecraft in ballistic re-entry to meet peak heat flux, total heat load, and deceleration constraints. | [Sutton and Graves, 1971] |
| ZH_04 | Determine a feasible glide angle, vehicle volume, and ballast mass for a buoyancy-driven underwater glider to cover 4 km in $\leq 2$ hours under given power, depth, and buoyancy constraints. | |
| Topic: Structural Design | | |
| RK_01 | Generate a 2D topology-optimized design of a rectangular beam under an asymmetric point load with fixed/roller supports to minimize compliance at a maximum volume fraction of 0.25, then report its compliance and volume fraction. | |

| Task-id | Task Summary | Reference |
|---------|--------------|-----------|
| RK_02 | Perform a 2D compliance-based topology optimization of a rectangular domain under two downward point loads with supports at its corners, subject to a volume fraction limit of 0.25. | |
| RK_03 | Perform 2D topology optimization on a rectangular domain with specified supports and loading to minimize compliance under a 15% volume fraction constraint. | |
| RK_04 | Perform topology optimization on a cracked 2D domain under given horizontal loads to minimize the maximum stress while restricting the material volume to 25%. | |
| YF_01 | Determine the required wall thickness of an L-shaped extruded steel beam so that its maximum z-direction displacement under a specified surface traction remains below 0.1 mm in a static linear FEA simulation. | |
| YF_02 | Determine the required thickness of a simply supported rectangular steel beam under two quarter-span loads so that its maximum vertical displacement is less than 1 mm in a 3D PDE Toolbox simulation. | |
| YF_03 | Determine the plate thickness of a rectangular steel cantilever beam under a specified half-length uniform pressure so that its maximum vertical deflection stays below 2 mm in a 3D linear elasticity model. | |
| YF_04 | Determine the diameter of a cantilevered solid steel shaft under a specified torque so that its maximum end-to-end twist does not exceed 0.05 rad. | |
| YF_05 | Determine the required cross-sectional area of the truss member between Node 2 and Node 3 so that the maximum nodal displacement under a 4000 N load at Node 3 remains below 0.5 mm. | |
| YF_06 | Determine the required thickness $t$ of a fixed-edge rectangular steel plate under uniform pressure to meet a specified deflection criterion using MATLAB's PDE Toolbox. | |
| YJ_01 | Determine the optimal 2D material distribution on a $64 \times 64$ grid that minimizes structural compliance under given loads, boundary conditions, and a volume fraction constraint. | [Mazé and Ahmed, 2022a], [Mazé and Ahmed, 2022b] |
| YJ_02 | Perform a topology optimization of a discretized cantilever beam to minimize compliance under a given volume constraint. | [Lagerweij, 2024] |
| YJ_03 | Set up and solve a topology optimization to distribute material in a finite-element mesh so as to minimize the Mode I stress-intensity factor at a crack tip under a volume-fraction constraint. | [Lagerweij, 2024] |
| | Topic: Digital Hardware Design | |

| Task-id | Task Summary | Reference |
|---|---|---|
| XY_01 | Analyze and extract the 4x4 bit pattern of a specified Tetris tetromino in a given rotation from ROM data, and convert it into a visual character-based representation for display. | [IEEE, 2018], [Tetris Wiki contributors], [FPGA4student, 2017b] |
| XY_03 | Design a time-multiplexed 4-digit hexadecimal display driver on a 50 MHz FPGA that guarantees at least a 60 Hz per-digit refresh and evaluates worst-case flicker. | [Chu, 2018], [FPGA4student, 2017a], [IEEE, 2018] |
| XY_04 | Design a hardware-efficient color mapping system on an FPGA for a Tetris game with RGB444 VGA output that supports distinct element colors and a dynamic Night Mode under a 10-color limit. | [Nicolle] |
| XY_05 | Define the per-instruction control signal settings, finite-state-machine transitions, and detailed explanations for the SLC-3 CPU, including a custom SWAP instruction, under strict hardware and sequencing constraints. | [Hennessy and Patterson, 2017], [ECE Department, 2025b], [ECE Department, 2025a] |
| YH_01 | Implement a tiled and pipelined GEMM function in Vivado HLS to maximize DSP utilization on the VCK5000 using block sizes of $64 \times 64 \times 64$. | |
| YH_02 | Implement a DSP-optimized tiled GEMM in Vivado HLS for $1024 \times 1024$ matrices using $128 \times 128 \times 128$ blocking on the VCK5000 platform. | |
| YH_03 | Create a synthesizable Vitis HLS GEMM function with AXI interfaces and loop pipelining for $1024 \times 1024$ matrices targeting the VCK5000. | |
| YH_04 | Design a Vitis HLS-compatible GEMM function for $256 \times 256$ matrices with AXI interfaces and pipelined inner loop targeting the VCK5000. | |
| NS_PA_SS_02 | Design a parameterizable n-bit Gray-code sequence generator in Verilog that updates on each rising clock edge and resets synchronously with active-low reset to zero. | [chi, 2025] |
| NS_PA_SS_03 | Design a parameterized 32-entry, 2-read/1-write register file in synthesizable SystemVerilog with synchronous reset, read-/write enables, and collision detection. | [chi, 2025] |
| NS_PA_SS_04 | Implement a parameterizable 8-bit Fibonacci LFSR in synthesizable SystemVerilog with synchronous active-low reset, seed loading, tap buffering, and feedback shifting. | [chi, 2025] |
| NS_PA_SS_05 | Develop a parameterizable, synthesizable SystemVerilog module that captures eight bitwidth-wide input words, performs a fully unrolled one-cycle bubble sort on them when triggered, and presents the sorted vector plus optional valid flag. | [chi, 2025] |

**Table 15 (continued from previous page)**

| Task-id | Task Summary | Reference |
|---|---|---|
| NS_PA_SS_06 | Design a synthesizable SystemVerilog module that reads two 3-element vectors serially, computes their dot product in one cycle after the sixth input, and drives an 18-bit result plus a one-cycle valid signal with defined reset behavior. | [chi, 2025] |
| NS_PA_SS_07 | Implement a synthesizable $8 \times 8$ register file in Verilog with one read port, one write port, valid-bit tracking, error flagging, and synchronous reset/clock behavior. | [chi, 2025] |
| NS_PA_SS_08 | Develop a fully synchronous, parameterized SystemVerilog module that detects a user-loaded 5-bit target pattern in a serial input stream and pulses a one-cycle 'seen' flag on each match. | [chi, 2025] |
| NS_PA_SS_09 | Create a fully synthesizable, parameterized SystemVerilog module that converts an N-bit Gray code input into its binary equivalent using a combinational XOR cascade. | [chi, 2025] |
| NS_PA_SS_10 | Implement a synthesizable SystemVerilog finite-state machine that reads a serial bitstream, tracks the value modulo 5, and raises dout when the running value is divisible by 5. | [chi, 2025] |
| Topic: Analog Integrated Circuit Design | | |
| TB_01 | Size the devices in a folded-cascode OTA netlist by assigning concrete W, L, m, and R values to meet given performance specs under a fixed testbench. | [ECE Department, University of Illinois Urbana-Champaign, 2025c] |
| TB_02 | Design and size a folded-cascode OTA from scratch in SPICE format to meet specific performance specs using the tsmc18 process and a given testbench setup. | [ECE Department, University of Illinois Urbana-Champaign, 2025c] |
| TB_03 | Size a 5-transistor differential-to-single-ended OTA by assigning W, L, and m values to meet gain, bandwidth, CMRR, and ICMR specs using tsmc18 models in a fixed testbench. | [ECE Department, University of Illinois Urbana-Champaign, 2025b] |
| TB_04 | Design and size a 5-transistor OTA along with its bias circuit in a single SPICE netlist to meet performance specs under a unity-gain testbench. | [ECE Department, University of Illinois Urbana-Champaign, 2025b] |

| Task-id | Task Summary | Reference |
|---------|-------------|-----------|
| TB_05 | Size a common-source amplifier with resistive load in SPICE by tuning NMOS, resistor, and bias parameters to meet gain, bandwidth, and output voltage specs. | [ECE Department, University of Illinois Urbana-Champaign, 2025a] |
| | Topic: Robotics | |
| Ziheng_02 | Determine the six joint angles of a 6-DOF serial robot that achieve a specified end-effector pose given the robot's screw axes and home transformation. | [Weng et al., 2018] |
| AM_02 | Compute two time-parameterized, collision-free trajectories for 2 by 2 robots on a 30 by 30 grid from given start to end points in 20 time steps, avoiding static rectangular obstacles and dynamically moving pedestrians under a maximum per-step speed of 2 units in each axis. | |
| AM_03 | Compute a time-stamped, grid-based, collision-free trajectory for a 2 by 2 robot on a 30 by 30 map with static rectangles and three moving pedestrians, starting at (17,2) and visiting goals A and B in any order without exceeding a per-step speed of 2 in x or y. | |
| HJ_01 | Select refresh rate, acceleration, maximum velocity, and look-ahead distance settings to complete a 56 m lap in under 10 s with a maximum 1 m track-off error on a $300 budget. | |
| XZ_01 | Compute the shortest collision-free grid path for a small inspection robot across a $50 \times 40$ m construction map with specified walls and obstacles. | [mat, 2025] |
| XZ_02 | Compute a smooth, collision-free trajectory for an autonomous construction vehicle from $(0,0,\pi)$ to $(49,39,\pi/2)$ on a $50 \times 40$ m site with fixed obstacles, while respecting a 4 m turning radius and 0.5 m vehicle clearance. | [mat, 2025] |
| XZ_03 | Generate a collision-free shortest path on an integer grid in a Webots world from (0,0,0) to (49,39,0) around defined obstacles. | [mat, 2025] |
| XZ_04 | Generate a time-parameterized, collision-free 3D trajectory in a Webots world from a start to a goal point while respecting zone-based speed constraints. | [mat, 2025] |
| YX_01 | Compute the world-coordinate corners of a 2D sparse costmap and then use a Hybrid A* planner with nonholonomic constraints to find collision-free paths for given start and goal poses. | [mat, 2025] |
| ZH_03 | Determine wheel diameter, gear ratio, battery capacity, and robot mass for an electric ground robot to meet specified speed, incline, and endurance targets. | |
| | Topic: Signal Processing | |

| Task-id | Task Summary | Reference |
|---------|--------------|-----------|
| qjlim2_01 | Design a rectangular microstrip patch antenna on a Rogers laminate to resonate at 1.537 GHz with $\geq$ 50 MHz bandwidth, $\geq$ 3 dBi gain, and fit within a $100 \times 100 \times 10$ mm volume using a $50\Omega$ coaxial feed. | [Antenna-Theory.com, 2011b], [Kashwan et al., 2011], [mat, 2025] |
| qjlim2_02 | Design a free-space strip dipole antenna centered at 3 GHz with at least 100 MHz bandwidth, $\geq$ 2 dBi gain, and fitting within a $100 \times 100$ mm footprint. | [Antenna-Theory.com, 2011a], [mat, 2025] |
| qjlim2_04 | Design a cylindrical monopole antenna resonant at 0.487 GHz with S11 $\leq -10$ dB, bandwidth $\geq$ 10 MHz, and gain $\geq$ 0 dBi. | [Antenna-Theory.com, 2011a], [mat, 2025] |
| XG_09 | Tune the parameters of Gaussian and triangular membership functions in a fuzzy inference system to improve edge detection performance (MSE, PSNR, SSIM) on a grayscale image relative to a Canny pseudo-ground truth. | [mat, 2025] |
| YZ_01 | Design a three-stage sample-rate converter—including a Farrow fractional resampler and two decimating FIR filters—to down-convert input rates (e.g., 150 MHz) to 30.72 MHz while meeting LTE passband, stopband, and EVM requirements. | [mat, 2025] |
| YZ_02 | Derive analytic formulas and determine coefficient sets for SG filter cutoff and half-magnitude width, then design four weighted SG filters meeting specified noise-vs-smoothness performance criteria. | [mat, 2025] |
| YZ_03 | Design a circularly polarized helical antenna in MATLAB's Antenna Toolbox to operate from 1.3 to 2 GHz, meeting specified directivity and axial ratio requirements. | [mat, 2025] |
| AB_01 | Compute first-order statistical features (mean, variance, skewness, kurtosis) and the GLCM contrast for pixel intensities strictly inside a given polygonal ROI in a medical image patch. | |
| AB_02 | Implement the OpenCV watershed algorithm in Python to segment the red heart pips and numerals from the white background in the specified image. | [Howse and Minichino, 2020] |
| AB_03 | Load a hard-coded image, find its largest contour, and compute both its approximate polygon and convex hull using OpenCV. | [Howse and Minichino, 2020] |
| AV_02 | Design a digital IIR filter to notch out a 120 Hz resonance in a 2000 Hz-sampled sensor signal while preserving $\pm 15$ Hz around that frequency. | [Zhi-Pei Liang, 2025] |
| AV_03 | Design an FIR anti-aliasing filter to allow downsampling a 48 kHz audio signal to 8 kHz while preserving the 0-3.5 kHz band with less than 3 dB ripple. | [Zhi-Pei Liang, 2025] |
| HC_03 | Estimate the coefficients a, b, and c of a second-order polynomial $y = ax^2 + bx + c$ that best fits a given noisy 2D data set by minimizing mean squared error. | |

| Task-id | Task Summary | Reference |
|---------|--------------|-----------|
| JY_01 | Design and implement an automated test stand using two linear polarizers, a quarter-wave plate, and rotary motors to control and measure changes in the Stokes vector of polarized light. | |
| JY_02 | Design and apply a Gaussian smoothing filter, horizontal and vertical edge-detection kernels, and a thresholding step to produce an edge map from the given image array. | [Soria et al., 2020] |
| JY_03 | Design a convolutional kernel that performs linear interpolation to fill zero-valued pixels in the corrupted Degree0 polarization image and evaluate its effect on AoLP and DoLP outputs. | |
| WJ_01 | Design and implement a flexible OpenCV based filter pipeline choosing and parameterizing one or more denoising filters based on detected noise types. | |
| YX_03 | Compute and combine SNR metrics, integration gains, and various radar losses to evaluate the detectability of a $1\,\text{m}^2$ target at 100 km for an S-band surveillance radar. | [mat, 2025] |

