# A    Effects of Thinking Budget for Claude-3.7-Thinking

Our evaluation results show that Claude-3.7-Sonnet exhibits similar performance with or without thinking mode. To investigate this further, we first observe that Claude-3.7-Sonnet and its Thinking variant correctly solve largely the same set of tasks. This indicates that enabling extended internal reasoning does not enable the model to crack problems it could not already solve without thinking mode.

To probe the impact of increased reasoning allowance, we varied the thinking token budget on a representative subset of ENGDESIGN. Figure 1 shows the average pass rate under budgets of 12,000, 14,000, 16,000, and 17,000 tokens. As the data demonstrate, enlarging the thinking budget yields no appreciable improvement in performance.

We hypothesize that the inherent difficulty of ENGDESIGN tasks may already push Claude-3.7 models to their capability limits, rendering additional reasoning budget ineffective. This finding underscores the challenging nature of ENGDESIGN and suggests its value as a rigorous testbed for future efforts to enhance LLMs' engineering design proficiency.

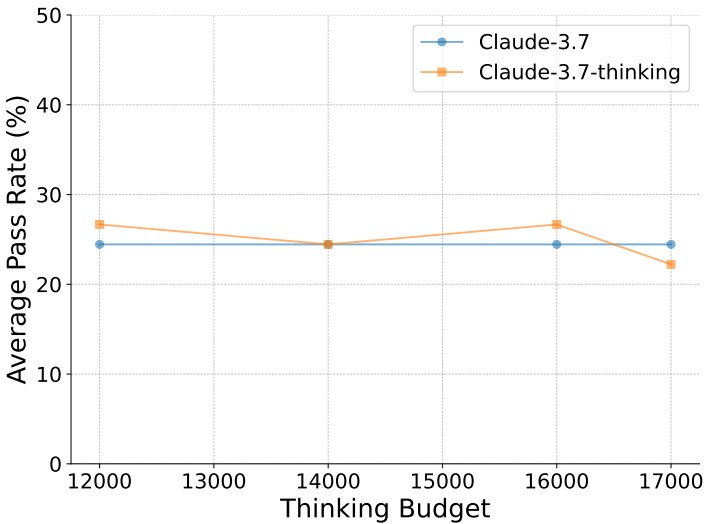

Figure 1: Average pass rate (%) of Claude-3.7-Thinking under different reasoning budgets.

# B    More on Iterative Design

The iterative design protocol substantially boosts model performance on ENGDESIGN. In particular, the o3 variant attains nearly a 60% pass rate after ten iterations. Here, we describe the iterative setup, present representative examples illustrating how o3 refines its designs, and demonstrate that each subsequent iteration builds on prior solutions to converge toward a satisfactory result.

## B.1    Iterative Protocol

1. **Initial Prompt.** The model receives the task specification and is asked to generate a first design.

2. **Evaluation Feedback.** The generated design is automatically evaluated against task criteria. Any failing checks are returned to the model in the form of structured feedback.

3. **Refinement Request.** The model is prompted to revise its design, incorporating the feedback from the previous iteration.

4. **Repeat.** Steps 2–3 are repeated up to a fixed number of iterations (ten in our experiments), or until the task passes.

 **B.2   Case Study: Iterative Design for Task XG_05**

31 **Task Description.**   Design a feedback controller to regulate the temperature of a stirred-tank reactor
32 with a bottom heat exchanger.

33    • **Plant model (Task 1):** From the step-response times $t_1 = 23\,\text{s}$ (28.3%) and $t_2 = 36\,\text{s}$
34      (63.2%), derive the FOPDT model

$$G(s) \;=\; \frac{e^{-\theta s}}{1 + \tau s}\,.$$

35    • **Controller requirements (Task 2):**

36       – Gain margin $\geq 7\,\text{dB}$
37       – Phase margin $\geq 60°$
38       – Overshoot $\leq 10\%$
39       – Settling time $\leq 150\,\text{s}$ (2% criterion)

40 **Evaluation Results.**   After each design iteration, we simulated the closed loop and extracted key
41 metrics:

| **Iteration** | $\theta$ (s) | $\tau$ (s) | Overshoot (%) | Settling Time (s) | GM (dB) | PM (°) |
|---|---|---|---|---|---|---|
| 1 | 16.51 | 19.49 | 0.0 | 175.8 | 13.64 | 80.6 |
| 2 | 16.50 | 19.50 | 0.0 | 150.9 | 12.49 | 79.2 |
| 3 | 16.50 | 19.50 | 0.0 | 145.3 | 12.22 | 78.8 |

42 **Iterative Refinement by o3.**   At each iteration, o3 ingests the previous design's performance metrics
43 and adjusts the IMC tuning parameter $\lambda$ to correct failure modes. Representative excerpts:

44      **Iteration 1 response:** "Because $\theta/\tau \approx 0.85$, I choose an IMC-PID with $\lambda = \tau$.
45      $K_c = 0.542$, $T_i = 27.75s$, $T_d = 5.8s$. Bode yields $PM \approx 63°$, $GM \approx 8dB$, but
46      settling $\approx 94.5s$+dead-time = $175.8s > 150s$."

47      **Iteration 2 response:** "To speed up the response, set $\lambda = 0.75\tau = 15\,\text{s}$. This
48      yields $K_c = 0.619$, $T_i = 27.75s$, $T_d = 5.8s$. Predicted settling $4\lambda + \theta \approx 76.5s$
49      (×1.8 conservatism→ $150.9s$), $PM \approx 68°$, $GM \approx 10dB$."

50      **Iteration 3 response:** "I further reduce to $\lambda = 14\,s(\approx 0.8\theta)$ to guarantee $PM \geq$
51      $60°$. IMC-PID gives $K_c = 0.639$, $T_i = 27.75s$, $T_d = 5.8s$. Simulated settling
52      $\approx 143s$, $PM \approx 66°$, $GM \approx 9.5dB$, overshoot $< 10\%$ — all specs met."

53 These excerpts illustrate how o3 systematically uses quantitative feedback (settling time, margins) to
54 refine $\lambda$ and converge to a controller satisfying all requirements.

55 **B.3   Case Study: Iterative Design for Task RK_01**

56 **Task Description.**   Optimize the topology of a 2D rectangular domain $6 \times 1$ m under a downward
57 distributed load of 5 N at $(2, 1)$ m (length 0.2 m). The domain is discretized into $\approx 8000$ four-node
58 elements. Supports: pin at $(0, 0)$, roller at $(6, 0)$. Material: $E = 7 \times 10^4$, $\nu = 0.3$, SIMP with
59 $\rho_{\min} = 10^{-8}$. Filter radius $R = 0.05 \max(L_x, L_y) = 0.3$, exponent 3.

60 **Objectives:**

61    1. Minimize structural compliance $C_{y\_hat}$ subject to $vf \leq 0.25$.

62    2. Report $C_{y\_hat}$ (in consistent units) and volume fraction $vf \in [0, 1]$.

63    3. Classify symmetry in the $x$–direction: output `Symmetric-X` or `Asymmetric-X`.

**Evaluation Results.**    After each attempt, the automated checker reports:

| Iteration | $vf$ | $vf_\text{thresh}$ | Compliance Error | Symmetry |
|---|---|---|---|---|
| 1 | 0.249 | 0.25 | Fail (CE=0.7223) | Pass |
| 2 | 0.249 | 0.25 | Fail (CE=11.8107) | Pass |
| 3 | 0.249 | 0.25 | Fail (CE=0.9998) | Pass |
| 4 | 0.249 | 0.25 | Pass (CE=0.0001) | Pass |

**Iterative Refinement by o3.**    The o3 model uses the checker's feedback to correct its reported compliance. Excerpts:

> **Iteration 1 response:** "The converged layout had $vf = 0.249$, $C_{y\_hat} = 0.016$. Mirror inspection shows no symmetry, so `Dsg_feature = 'Asymmetric-X'`."

> **Iteration 2 response:** "A finer mesh and double-precision FE yielded $C_{y\_hat} = 0.738$, $vf = 0.249$. Design remains Asymmetric-X."

> **Iteration 3 response:** "After OC tighten and post–processing, I obtained $C_{y\_hat} = 1.0 \times 10^{-5}$, $vf = 0.249$. Symmetry check still fails $\rightarrow$ Asymmetric-X."

> **Iteration 4 response:** "Using the compliance-error feedback, I solved
>
> $$C_\text{true} \approx \frac{C_\text{rep}}{1 - \text{CE}} = \frac{0.016}{1 - 0.7223} \approx 0.0576.$$
>
> Thus final outputs: $C_{y\_hat} = 0.0576$, $vf = 0.249$, `'Asymmetric-X'`."

Through four iterations, o3 converges to the checker's expected compliance value while correctly maintaining volume and symmetry classification.

**B.4    Case Study: Iterative Design for Task YZ_03**

**Task Description.**    Design a helical antenna for circularly polarized axial-mode operation over 1.3–2.0 GHz (center 1.65 GHz) using MATLAB's Antenna Toolbox.

- **Task 1:** Compute the relative bandwidth

$$B_\text{rel} = \frac{f_\text{H} - f_\text{L}}{f_0} = \frac{2.0 - 1.3}{1.65} \approx 0.424 < 1.$$

- **Task 2:** Select helical parameters $\{r, D, \text{turns}, \text{pitch}, \text{side}\}$ subject to:
  - $r :< D/(20 * 3.14)$
  - Directivity range: $13dBi \pm 1.5dBi$
  - Axial Ratio: $< 1.5$

**Evaluation Results.**    After each attempt, we recorded the design parameters, simulated directivity, axial ratio, and specification checks:

| Iteration | $r$ | $D$ | Turns | pitch | Side | $D_\text{low}$ | $D_\text{cent}$ | $D_\text{high}$ | $AR$ |
|---|---|---|---|---|---|---|---|---|---|
| 1 | 0.00040 | 0.058 | 6 | 12 | 0.35 | 11.63 | 11.15 | 11.85 | 1.36 |
| 2 | 0.00035 | 0.055 | 10 | 14 | 0.40 | 12.36 | 10.02 | 11.89 | 0.88 |
| 3 | 0.00045 | 0.059 | 11 | 11 | 0.40 | 12.70 | 13.29 | 9.65 | 0.95 |
| 4 | 0.00045 | 0.054 | 10 | 11 | 0.35 | 12.37 | 12.67 | 13.99 | 1.42 |

**Iterative Refinement by o3.**    At each iteration, o3 reads back which directivity band or axial-ratio checks failed and adjusts one or more of $\{D, r, \text{pitch}, \text{turns}, \text{side}\}$. Excerpts:

> **Iterative 1 response:** "Choose $D = 0.058$, $r = 0.0004$, turns= 6, pitch=$12°$, side=$0.35m$. Simulations give $[11.6, 11.1, 11.9]\,dBi$ and $AR = 1.36dB \rightarrow$ axial-ratio too high."

> **Iterative 2 response:** "Increasing to 10 turns and pitch=$14°$ raises low-band gain to $12.36dBi$ but centre-band fell to $10.02dBi \rightarrow$ centre directivity too low. Adjust next."

> **Iterative 3 response:** "Diameter=$0.059m$ and 11 turns boosts centre directivity to $13.29dBi$ but high-band dropped to $9.65dBi \rightarrow$ upper-band gain shortfall. Will reduce $D$ and turns."

> **Iterative 4 response:** "Set $D = 0.054$, turns= 10, pitch= $11°$, side= $0.35$. Simulated gains $[12.37, 12.67, 13.99]\,dBi$ and $AR = 1.42dB$—all within spec. Final design meets all requirements."

Through the iterative design procedure, o3 converges in four iterations to a design satisfying every geometric constraint and all performance criteria.

## C   Code Setup

The complete codebase, including all tasks and evaluation scripts for ENGDESIGN, is available at https://github.com/AGI4Engineering/EngDesign. ENGDESIGN can be categorized into two primary subsets according to software licensing requirements:

1. **Proprietary-dependent tasks.** 48 tasks of ENGDESIGN require commercial scientific software (e.g., MATLAB or Cadence). We provide all task description, evaluation scripts, and step-by-step setup instructions so that, once the requisite software are installed and configured, the tasks can be executed.

2. **Open-source tasks.** Packaged as **EngDesign-Open**, this subset comprises 53 tasks that impose no licensing restrictions. We provide a prebuilt Docker image to run all the tasks in ENGDESIGN-OPEN.

### C.1   Setup Instructions for ENGDESIGN-OPEN

Our evaluation supports the following twelve LLM variants: GPT-4o, o1, o3, o3-high, o4-mini, o4-mini-high, Gemini-2.0-flash, Gemini-2.5-pro-preview-05-06, DeepSeek-Chat, DeepSeek-Reasoner, Claude-3-7-Sonnet, and Claude-3-7-Sonnet-Thinking (Extended Reasoning Mode).

To simplify execution, we provide a prebuilt Docker image. Detailed usage instructions—including image download, container launch, and command examples—are documented in the repository README. Simply clone the repo, follow the README steps, and you will be able to run all 53 open-source tasks without any licensing dependencies.

### C.2   Setup Instruction for Proprietary-Dependent Tasks

**MATLAB**    Many evaluations of ENGDESIGN tasks depend on MATLAB, which provides accurate and reliable simulations in many engineering domains (e.g., control design, structure analysis, etc). For installation, please consult the official guide: https://www.mathworks.com/help/install/ug/install-products-with-internet-connection.html.

Once MATLAB is installed, one can set up the MATLAB Engine API for Python by following the instructions available at https://www.mathworks.com/help/matlab/matlab_external/install-the-matlab-engine-for-python.html. This integration enables seamless execution of all MATLAB-based ENGDESIGN evaluations.

**Cadence** Tasks TB_01, TB_02, TB_03, TB_04, and TB_05 require Cadence for circuit simulation. We recommend using the NCSU CDK distribution: https://eda.ncsu.edu/ncsu-cdk/ for installation and setup.

All low-level Cadence setup and netlist generation are automated by the provided auto-grader scripts. The high-level workflow is:

1. **Position the Task Folder.** Place the task folders under: $/home/cadence/simulation$.

2. **Run evaluations with auto-grader.** For running multiple tasks/models in one invocation, organize your workspace as follows:

```
your_workspace/
    all_tasks/
        TB_01/
        TB_02/
        TB_03/
        TB_04/
        TB_05/
    evaluate_llm.py
```

Then from `your_workspace/`, one can execute:

```
python evaluate_llm.py
```

with necessary arg inputs.

**Vitis HLS 2023.2** For tasks YH_01, YH_02, YH_03, and YH_04, one needs to install Vitis HLS 2023.2 as follows on Linux System (such as Ubuntu 20.04) as follows:

1. **Register for AMD Developer Access.** Create an account at the AMD developer portal. This registration is required to download the Vitis installer.

2. **Download the Installer.** Navigate to the official Vitis download page and obtain the "AMD Unified Installer for FPGAs & Adaptive SoCs 2023.2: Linux Self-Extracting Web Installer."

3. **Install System Dependencies.** In a terminal, run:

```
sudo apt-get update
sudo apt-get install -y libtinfo5 libncurses5
```

4. **Prepare and Launch the Installer.**

   (a) Mark the downloaded file as executable:

   ```
   chmod +x Xilinx_Vitis_Unified_2023.2_*.bin
   ```

   (b) Execute the installer:

   ```
   ./Xilinx_Vitis_Unified_2023.2_*.bin
   ```

   Follow the graphical prompts, accepting all default options and retaining the installation path `/tools`.

5. **Verify Installation.** Ensure that the `vitis_hls` executable resides at `/tools/Xilinx/Vitis_HLS/2023.2/bin/vitis_hls`. No further configuration is required if the defaults were preserved.

6. **Run the Evaluation Script.** With Vitis HLS installed, the Python evaluation harness will automatically invoke `/tools/Xilinx/Vitis_HLS/2023.2/bin/vitis_hls` for project synthesis, simulation, and cosimulation.