# OpenReview forum: "Toward Engineering AGI: Benchmarking the Engineering Design Capabilities of LLMs"
_NeurIPS.cc/2025/Datasets_and_Benchmarks_Track — NeurIPS 2025 Datasets and Benchmarks Track poster_

### Official Review · Reviewer_FsX6 · 2025-06-29

**Rating:** 5
**Confidence:** 4

**Summary:**

The authors introduce ENGDESIGN, a benchmark meant to test LLMs on engineering design work instead of textbook-style questions. ENGDESIGN spans nine engineering domains and contains 101 problems linked with evaluations (based e.g. on manually-written scripts) that can assess the quality of the proposed solutions rather than just a pass/fail label. Experiments with SotA models show that they get somewhat good results that do not seem to saturate the benchmark.

**Additional Feedback:**

It might be necessary to try to anchor the benchmark with a human-engineer baseline; this may be difficult to do in a "normalized" and cost/time-effective fashion, but it is something I was surprised not to see.

**Dataset Code Accessibility:**

Yes

**Dataset Code Comments:**

Code is availabe on github, and includes evaluation scripts. Dataset is available on huggingface.

**Ethical Considerations:**

No, there are no or only very minor ethics concerns

**Final Justification:**

My concerns were mostly minor, and the authors have provided sufficient information + justification to maintain a positive rating.

**Limitations Weaknesses:**

The most important (potential) problem I see in this work is that evaluation successes hinge on every model emitting JSON that exactly matches the Instructor schema. The paper barely analyzes whether those schemas are question-specific, overly rigid, or silently leak task structure. If the schema hard-codes what parameters exist, the model is reduced to filling in numbers rather than proposing novel designs, capping expressiveness. A concrete discussion of a) how many tasks needed bespoke schemas; b) failure rates caused by minor formatting drift; and c) whether free-form answers could be graded post-hoc, is missing.

The fact that the Analog Integrated Circuit Design problem category has an average pass rate of 0% seems to indicate a difficulty calibration problem. It might be interesting to see how much "simplifying" a single problem from that category is necessary in order for any model to propose a passing solution; doing so may also help identify other problems that may be affected model performance.

The iterative “design-fix-resubmit” protocol (section 3.2) is intriguing but under-specified -- how was feedback phrased? was chain-of-thought retained or regenerated? I have the intuition that this protocol may become a significant way to improve the performance of agents on this benchmark in the future, and therefore it would be important to structure how future authors should be conducting these experiments so that they have a standardized approach to follow.

Another minor point is the writing quality: there are several passages that seem a bit "cheesy" across the paper (e.g. "from starships to Dyson spheres"). Trimming some of those clichés might help boost credibility, even if I myself share the optimism for this kind of work.

**Strengths Contributions:**

The benchmark has an ambitious scope, and the tasks to be solved seem fairly realistic. I was surprised myself by how few tasks I thought I would be able to solve on my own; the covered domains are quite varied.

The fact that evaluations are based on automated verification (using e.g. scripts) makes human judgement (or unreliable LLM-as-a-judge frameworks) unnecessary, which helps us focus on the actual practical value of LLMs in the studied domains.

The results are quite interesting and show (among other points) that modern reasoning models do not necessarily have more robust reasoning compared to their ancestors. There is also a very interesting breakdown and analysis of failure modes for a recent model (o4-mini).

---

> ### Author Rebuttal · Authors · 2025-07-30
>
> We sincerely thank the reviewer for the positive and encouraging feedback. We greatly appreciate the recognition that **EngDesign has an ambitious scope** with realistic tasks spanning a wide range of engineering domains, many of which are challenging even for human experts. The reviewer also highlighted the strength of our **automated verification pipeline**, which eliminates reliance on subjective human judgment or unreliable LLM-as-a-judge frameworks, ensuring that evaluations accurately reflect the **actual practical value of LLMs in the studied domains**. We are also pleased that the reviewer found our **results insightful**, particularly the observation that modern reasoning models do not always exhibit more robust reasoning than their predecessors, and that our **detailed breakdown and analysis of failure modes for o4‑mini** provides valuable insights. Next, we address the reviewer’s concerns point by point below.
>
> ---
>
> ### **Reliance on Instructor Schema and Potential Constraints**
>
> We sincerely appreciate the reviewer’s thoughtful concern regarding our use of the Instructor schema. As described in Section 2.2, we employ **Instructor**, a widely used Python library that enforces structured outputs from LLMs. For every task, we prompt models to generate responses in two parts:
>
> * **Reasoning** – a universal field capturing the model’s open-ended, step-by-step reasoning process.
> * **ConfigFile** – a task-specific field extracting key parameters or code snippets from the reasoning trace for automated evaluation.
>
> Importantly, the use of Instructor does **not** leak task structure. The schema enforces **format consistency** for evaluation purposes but does not dictate the solution path. The model must still generate a **complete reasoning trace** and derive structured outputs from it, preserving expressiveness. Moreover, the output format can be **complex**, often containing nested configurations, making simple “number-filling” insufficient.
>
> We now address the reviewer’s subpoints:
>
> * **(a) How many tasks required bespoke schemas:**
>   All tasks use tailored JSON schemas to ensure compatibility with the evaluation pipeline. This design was essential given the diversity of domains and heterogeneity of outputs (e.g., controller parameters vs. structural designs).
>
> * **(b) Failure rates caused by formatting drift:**
>   Formatting drift was rare. After examine GPT-4o responses, fewer than 1% of responses failed due to formatting errors, and nearly all were corrected by re-running. This demonstrates that Instructor effectively minimizes formatting issues.
>
> * **(c) Whether free-form answers could be graded post-hoc:**
>   Post-hoc grading of free-form answers is theoretically possible but practically unreliable. Existing methods either (1) restrict outputs to short strings for exact matching, limiting expressiveness, or (2) rely on LLM-as-judge frameworks, which introduce additional uncertainty. While using an **LLM-as-extractor** to parse free-form outputs is an option, it adds another fragile layer prone to misinterpretation. Given Instructor’s robustness and wide adoption, we believe it remains the most practical and principled solution for structured evaluation.
>
> ---
>
> ### **Zero Pass Rate in Analog IC Design**
>
> We thank the reviewer for noting the **0% pass rate** in Analog IC Design. This reflects the **inherent complexity** of these tasks.  Analog IC design is widely regarded as one of the most challenging areas of engineering. For analog IC design problems, a solution passes only if **all pre-defined performance requirements** are satisfied, and such a standard may take a human engineer hours or even days to achieve.  Thus, we are not surprised by the low pass rate.
>
> Importantly, our **partial-credit rubric** captures intermediate successes. As reported in Table 5 (Appendix E.4), models such as Claude‑3.7‑Sonnet and Gemini‑2.5‑Pro achieved **average scores of 31/100** in this domain, indicating that they met some requirements even without full success. As the reviewer suggested, simplifying tasks by relaxing certain constraints could make them solvable by current models and reveal **failure boundaries**. We will discuss this idea explicitly in the revised paper.
>
> ---
>
> ### **Iterative “Design-Fix-Resubmit” Protocol**
>
> We thank the reviewer for highlighting the promise of our **iterative design** protocol. We agree that this approach is an important avenue for improving model performance on complex tasks and merits clear standardization.
>
> To clarify, our protocol (Section E.6) uses the following iterative prompt format:
>
> ```
> Task Description: {original task description}
> Previous Responses: {previous responses}
> Evaluation Feedback: {previous evaluation results}
> Please review the feedback above and propose an improved solution based on the evaluation results.
> ```
>
> * **Chain-of-thought is retained** across iterations: previous reasoning traces and feedback remain visible to the model, enabling cumulative reasoning and refinement.
> * **Feedback** is structured to explicitly indicate unmet requirements.
> * **Iteration continues** up to 10 rounds or until the design meets all requirements.
>
> Section B of the supplementary material provides further protocol details and demonstration examples for reproducibility. The process consists of:
>
> 1. **Initial Prompt**: The model receives the task specification and generates a first design.
> 2. **Evaluation Feedback**: The design is automatically assessed, with unmet requirements returned in structured form.
> 3. **Refinement Request**: The model is prompted again with the full task description, previous responses, and feedback, and asked to revise its design.
> 4. **Repeat**: Steps 2–3 are repeated for up to 10 iterations, or until all requirements are met.
>
> This protocol mirrors how human engineers iteratively refine designs. By retaining reasoning and feedback across iterations, the model gains richer signals for self-correction. We will make these details more explicit in the main paper to guide future research in adopting this methodology.
>
> ---
>
>
>
> ### **Writing Style**
>
> We acknowledge the reviewer’s suggestion to remove overly casual phrases (e.g., “from starships to Dyson spheres”). We will revise these passages to maintain a **professional, academic tone**, improving the paper’s overall credibility.
>
> ---
>
> ### **Human Engineer Baselines**
>
> Thanks for this comment. Notice that the tasks in EngDesign require specialized expertise across 9 distinct engineering domains. **Although each task is solvable by the human engineer in that specific domain with sufficient time and required tools, no single human engineer would be expected to solve problems across all 9 domains.** This is similar to Humanity's Last Exam (HLE) [Phan et al., 2025], which does not involve human baselines due to the fact that no single human being is expected to solve the problems across all the areas in HLE. Hence we think that it is not meaningful to evaluate single human engineer's performance on EngDesign.  The purpose of our paper is to evaluate LLMs' capabilities in practical engineering design, and we believe that the main value of EngDesign lies in serving as a diagnostic tool and a unified testbed in: i) Identifying LLM failure modes for practical engineering design, ii) tracking improvements as LLMs advance, iii) providing simulation-based evaluation for the LLM community.
>
>
>
> ---
>
> **References**
>
> * \[Phan et al., 2025] Humanity’s Last Exam. *arXiv preprint arXiv:2501.14249*.
>
> ---
>
> ---
>
> We thank the reviewer once again for the constructive and insightful feedback. We have clarified our methodology, justified our design choices, and outlined revisions to improve clarity and rigor. We believe these updates will strengthen the paper and address all concerns thoroughly.

---

> > ### Comment · Reviewer_FsX6 · 2025-08-04
> >
> > I thank the authors for their detailed response;
> >
> > Extra thought regarding the Instructor schemas: I see your point, and I agree that what you are currently doing is probably the best solution in practice. I have some lingering doubts that *even if* the output formats may contain complex, nested parameter configurations, the fact that output configurations are still "forced" onto the model's answers means that we are limiting their expressiveness. Good designs are sometimes designs that go a little bit outside the box, if the box is wrong: capturing how often this happens would be really interesting, but again, likely impractical for such a benchmark. It might be a nice thing to note for "future works".
> >
> > Everything else looks good to me, thanks again.

---

> > > ### Author Response · Authors · 2025-08-04
> > > **Thank You for the Feedback**
> > >
> > > We sincerely thank the reviewer for their thoughtful feedback. We fully agree that good designs can sometimes go out of the box. Our use of Instructor schemas aims to balance the trade-off between a standardized, systematic framework for automated evaluation and the expressiveness of LLM outputs. Evaluating out-of-the-box LLM designs when responses are provided in free form remains an intriguing future direction. We will clarify this point more clearly in the paper and highlight it as a promising avenue for future research. We thank the reviewer again for their constructive comments!

---

### Official Review · Reviewer_verF · 2025-07-02

**Rating:** 5
**Confidence:** 4

**Summary:**

The authors propose EngDesign, a novel multi-domain benchmark to evaluate the capabilities of LLMs in complex, real-world engineering tasks. Therefore, the authors propose a set of 101 engineering tasks across nine different engineering domains that go beyond classical textbook-level problem solving, specifically assessing LLMs’ ability to generate objective-oriented designs while using domain knowledge, adhering to task-specific constraints, and balancing relevant trade-offs. In addition, the EngDesign benchmark implements a simulation-driven evaluation pipeline, allowing to determine the LLMs’ performance and to validate designs against task requirements using domain-specific simulations. It also extends the binary scoring of conventional benchmarks by providing a metric that quantifies partial progress and allows to reward iterative refinement. Through a comprehensive analysis across 10+ state-of-the-art LLMs, including general-purpose and reasoning models, the authors demonstrate critical limitations, such as over-reliance or constraint violation errors, of state-of-the-art LLMs in real-world engineering tasks.

**Additional Feedback:**

• I would recommend to provide a link/reference to the HuggingFace dataset and the gitlab repository directly in the paper
    • I am wondering how much prompt optimization was done. What about different prompting templates such as Chain-of-Thought prompting and other state-of-the-art prompting strategies? How much do the LLMs’ performances vary regarding different prompt formulations?
    • The authors could comment how users of the benchmark can extend it with novel or own engineering design tasks
    • In Fig. 6 (right), the type “other” with 0.9% should not be in the same color as the “prior knowledge overreliance”
    • Similarly, on could combine the two plots of Fig. 5, thus using a single plot with the same y-axis scale to highlight the significant differences of 4o and o1
    • The authors could comment how the reference designs (exemplary solutions) were generated; by providing multiple exemplary solutions one would also enable the use of approaches such as In-Context Learning
    • Provide additional information on Tiktoken and the o200k_base_encoding; see e.g. Table 1 (caption)
    • Carefully use the term AGI as there is not a clear definition and one could still argue that solving the provided benchmark tasks is not general intelligence as human-level intelligent (even in engineering domains) is more than implementing objective-oriented designs
    • “…, from analog integrated circuits forming the foundation of electronics…”;  not only analog but also digital and mixed-signal circuits form the foundation of modern electronics
    • Rigorously define/introduce abbreviations, e.g., AI in abstract
    • The first references seem to be a bit arbitrary. For example, why is [6] “Formal verification of safety-critical aerospace systems” a good citation for the sentence “From the electrical grids powering our cities to the aerospace systems launching us beyond Earth, from analog integrated circuits forming the foundation of electronics to the structural designs supporting our skylines, engineering disciplines have woven themselves into the fabric of human existence”? There would be much better/general choices.
    • Line 241 contains one dot too much
    • Why are not all 101 test cases in the same format? Not all contain a “## Task Description”. Furthermore, the descriptions contain errors. E.g. case JY_01 ends with “Notes: Are angles are given with respect to the horizontal axis” which should likely read “All angles”, as guessing the right task was not part of the test.

**Dataset Code Accessibility:**

Yes

**Dataset Code Comments:**

The benchmark EngDesign is hosted on HuggingFace and thus can be easily accessed using dataset loaders and copy-pasteable Python snippets. Moreover, the authors include a croissant file with all relevant data available, such as the metadata, dataset structure, and dataset summary.

**Ethical Considerations:**

No, there are no or only very minor ethics concerns

**Final Justification:**

I continue to support the publication of this paper.

**Limitations Weaknesses:**

• Initial LLM-based filtering/pre-testing could introduce biases, may be sensitive to different prompt formulations, and could dismiss tasks well-suited tasks (the opposite case is most likely detected in one of the review stages)
    • The composition of the group proposing the benchmark tasks could have introduced a selection bias.
    • Due to issues with the licenses of proprietary software tools such as MATLAB or Cadence, only 53 (approximately 50% of the benchmark) remain fully open-source
    • An evaluation in how many cases a LLM fail to adhere to the required output format enforced using instructor would be interesting/beneficial. For example, in code generation tasks a non-negligible number of generated codes can not be successfully extracted due to incorrect output formats and thus negatively impact the overall performance of LLMs

**Strengths Contributions:**

• Novel benchmark with complex, real-world examples of engineering tasks across multiple domains
    • Multi-stage reviewing process to construct a high-quality dataset that integrated automated validation in the initial filtering/pre-testing phase and human expertise  in the first- and second-round review process
    • Simulation-driven evaluation pipeline that  allows to verify functional feasibility based on engineering standard and best practices, ensuring objectivity and reproducible scoring
    • The partial-credit scores is particularly valuable for iterative refinement and optimizations, which are getting popular in LLM-driven frameworks, e.g., in code generation, especially since LLMs are often fine-tuned using Reinforcement Learning
    • Thorough analysis of challenges and limitations of using LLMs for complex, real-world engineering tasks
    • The authors carefully address limitations, including the potential issues that arise from enforcing a specific output structure using the instructor framework. This is particularly important as the cases in which the LLM generates a suitable design, but does not follow the output format, impact the overall performance analysis
    • Extensive analysis of common failure types regarding different state-of-the-art LLMs, including one or multiple examples for each type of failure; see Appendix F
    • Reference design (exemplary solution) for each of the task that sully satisfies all requirements, ensuring real-world feasibility
    • Well-written paper.

---

> ### Author Rebuttal · Authors · 2025-07-30
>
> We sincerely thank the reviewer for the careful review and the highly positive assessment of our work. We appreciate the recognition that **EngDesign is a novel benchmark** featuring **complex, real-world engineering tasks across multiple domains**. The reviewer highlights the strength of our **multi-stage review process**, which combines automated validation with expert review to ensure a **high-quality dataset**, and our **simulation-driven evaluation pipeline**, which verifies functional feasibility based on engineering standards and ensures **objective, reproducible scoring**. We are pleased that the reviewer values our **partial-credit scoring**, which supports iterative refinement and acknowledges our **thorough analysis of challenges, limitations, and common failure types** across state-of-the-art models. The reviewer also commends the inclusion of **reference designs** that fully meet task requirements, the **careful consideration of output structure constraints**, and the overall **clarity and quality of writing**. We are grateful for these encouraging remarks. Next, we will address the reviewer's concerns point by point below.
>
> ---
>
> ### **LLM-Based Filtering/Pre-Testing Could Introduce Bias**
>
> As described in Section 2.3, LLM filtering and pre-testing serve only as an **initial screening layer** in the dataset curation process. If the model flags missing or unclear details, we will **work directly with the original contributor to address these gaps**. Therefore, no well-suited tasks are dropped at this stage. Tasks are removed only if they are clearly **off-topic, low-quality, or do not reflect true engineering design**. Furthermore, any rare false negatives would likely be caught during the subsequent multi-stage expert review process.
>
> ---
>
> ### **Potential Selection Bias in Benchmark Task Composition**
>
> We agree with the reviewer that **some selection bias is inevitable** in any benchmark construction. We mitigated this through a **multi-layer data curation process** involving diverse contributors and two rounds of expert review. This process helps ensure that the tasks are both **representative** and **high-quality**, minimizing the impact of individual biases.
>
> ---
>
> ### **Use of Proprietary Software**
>
> We appreciate the reviewer’s concern and have taken concrete steps to improve accessibility. We realize that for several MATLAB-dependent tasks that do not require advanced toolboxes, we can directly replace Matlab simulations with GNU Octave (which is open-source) and produce equivalent evaluation results. This aligns with the well-known fact that GNU Octave is the open-source alternative with high compatibility for MATLAB when advanced toolboxes are not involved. Now only **34 out of 101** tasks (instead of 48 out of 101) still require commercial software such as MATLAB toolboxes, Cadence, or Vitis HLS, and the resultant domain-wise breakdown is as follows:
>
> | Domain                       | Total | OS | Arch | Ctrl | Mech | Stru | DHD | AICD | Robo | SigP |
> | ---------------------------- | ----- | -- | ---- | ---- | ---- | ---- | --- | ---- | ---- | ---- |
> | **# of open-source tasks**   | 67    | 8  | 5    | 7    | 6    | 7    | 13  | 0    | 10   | 11   |
> | **# of closed-source tasks** | 34    | 0  | 0    | 11   | 1    | 6    | 4   | 5    | 0    | 7    |
>
> Consequently, the majority of tasks (**67 out of 101**) are now fully open-source, significantly improving accessibility.  The remaining commercial tools (MATLAB toolboxes, Cadence, Vitis HLS) reflect industry standard practice essential for authentic evaluation, particularly in specialized domains like robust control and Analog IC Design where open-source alternatives lack the maturity and accuracy required for reliable assessment. While we explored open-source alternatives (e.g., the Python Control Systems Library), our preliminary experiments showed that these packages were insufficiently mature for evaluating some of our tasks.
>
>
> We emphasize that including commercial scientific software reflects **the practical reality of modern engineering practice**, where such tools are widely recognized as industry standards. This choice ensures that EngDesign faithfully captures real-world workflows and evaluates AI systems under realistic conditions. **There is an inherent trade-off between authentic engineering practice and open-source accessibility**. In industry, practicing engineers routinely rely on commercial software (e.g., MATLAB toolboxes for robust control, SPICE for circuit design) because these tools represent the state-of-the-art in their respective domains. Several of our authors have extensive prior industry experience and can confirm this is standard practice. To provide a meaningful evaluation of AI systems intended for real-world engineering applications, our benchmark must capture this reality.
>
> ---
>
> ### **Adherence to Output Format (Instructor Framework)**
>
> We thank the reviewer for this insightful comment. After closer examination on the responses from GPT-4o, we found that the **instructor package is very reliable**:
>
> * Across one trial, only **1 out of 101 tasks** failed to generate the required output format in one trial.
> * In such rare cases, **re-running the task produces the desired output**.
>
> Thus, we believe that formatting issues have **minimal impact** on overall performance analysis.
>
> ---
>
>
> ### **Prompt Optimization and Variants**
>
> Our evaluation prompt already incorporates **Chain-of-Thought (CoT)** reasoning, instructing the model to solve tasks step by step. Specifically, we employ the following prompt:
>
> ```
> Detailed reasoning process to accomplish the task, please solve all the tasks step by step.
> ```
>
> To address the reviewer’s question on the effects of different prompt, we additionally tested GPT-4o with and without CoT on EngDesign-Open. The results (below) show that **CoT in general improves performance** on both evaluation metrics (with some exception on signal processing and the score evaluation case of operating system design).
>
> | Domain                  | Pass Rate w/ CoT |Pass Rate w/o CoT | Score w/ CoT | Score w/o CoT |
> | ----------------------- | ---------------- | ----------------- | ------------ | ------------- |
> | Overall                 | **14.29%**           | 12.70%            | **42.89**        | 37.50         |
> | Operating System Design | **12.50%**           | 0%                | 33.25        | **36.94**         |
> | Control System Design   | **25%**              | **25%**               | **66.00**        | 57.50         |
> | Mechanical Systems   | **0%**              | **0%**              | **31.67**        | 25.00         |
> | Structural Design   | **0%**              | **0%**               | **32.86**        | 30.00         |
> | Digital Hardware Design   | **15.38%**              | 7.69%               | **50.54**        | 35.54         |
> | Robotics                | **22.22%**           | **22.22%**            | **59.78**        | 50.61         |
> | Signal Processing       | 27.27%           | **36.36%**            | 43.52        | **45.52**         |
>
> These results confirm that prompt structure influences model performance, and our current prompt formulation is reasonable. We will add the above new results and related discussion to our revised paper.
>
> ---
>
> ### **Extending the Benchmark**
>
> EngDesign was deliberately designed to be **highly modular**. Users can easily add new tasks by:
>
> 1. Defining a task description and output format,
> 2. Specifying evaluation scripts, and
> 3. Using our **existing evaluation pipeline** to automatically assess model performance.
>
> Existing tasks are equally easy to extend (e.g., by changing a dynamical systems or updating performance criteria in control tasks). This modularity encourages community participation and continual expansion.
>
> ---
>
> ### **Figure Improvements**
>
>
> We will revise **Figure 6** to differentiate the “other” category clearly and merge the plots in **Figure 5** into a single figure with a unified y-axis scale.
>
> ---
>
> ### **Reference Designs and In-Context Learning**
>
> Reference designs were provided by initial contributors to ensure they fully satisfy all task requirements and reviewed by reviewers. While demonstration examples could be used for in-context learning, we note that **EngDesign tasks are highly diverse**, and even within the same domain, tasks require **different reasoning paths**. Therefore, for in-context learning, it is unclear how to choose the right demonstrations for the purpose of improving the overall performances on the entire EngDesign  dataset.
>
> ---
>
> ### **Additional Minor Comments**
>
> * **Adding a link to the HuggingFace dataset and the gitlab repository directly in the paper**: We will do this for our revised paper.
> * **Information on o200k_base_encoding**: We will add background information on this tokenization method in **Table 1’s caption** and cite relevant references.
> * **Use of AGI terminology**: We will clarify our language and avoid ambiguous claims regarding AGI.
> * **Analog vs. Digital Circuits**: We will revise the sentence according to the reviewer's suggestion.
> * **Abbreviations**: We will rigorously define all abbreviations (e.g., AI) upon first use.
> * **References**: We will refine early citations to use more appropriate and general references.
> * **Typographical errors**: We will correct minor typos (e.g., the extra period in Line 241).
> * **Dataset formatting**: We will clean the dataset to ensure all 101 tasks follow a **consistent format** and correct errors such as the one in case JY\_01.
>
> ---
>
>
> We thank the reviewer for their thoughtful and constructive feedback. We have addressed all concerns through clarifications, additional analyses, and planned revisions, which we believe will significantly strengthen the manuscript.

---

> > ### Comment · Reviewer_verF · 2025-08-05
> >
> > Thanks a lot for the great reply and the clarifications.
> >
> > In particular the comparison with and without CoT is helpful, as guiding the LLM in the right direction is still key to good results. I am a bit surprised, that CoT leads to worse results for Operting System Design and Signal Processing. Do you have an explanation for that?
> >
> > Thank you for replacing some of the Matlab dependencies by Octave. I agree that in this realm it is merely impossible to get rid of all closed source software (e.g. for SPICE), but at least Matlab can often be replaced.

---

> > > ### Author Response · Authors · 2025-08-06
> > > **Analysis of CoT Performance**
> > >
> > > We sincerely thank the reviewer for such an insightful question. We have taken a closer look of our experiment results, and we observe that Chain‑of‑Thought (CoT) prompting techniques sometimes degrades performance, especially in **coding implementation** tasks. Below is the detailed breakdown for three demonstration examples:
> > >
> > > | Task Domain | Task ID | Score w/ CoT | Score w/o CoT | Performance Gap | Key Issues with CoT |
> > > |-------------|---------|-----------|--------------|-----------------|-------------------|
> > > | **Operating System** | CY_03 (Virtio Block Driver) | 1 | 55.5 | -54.5 | Incorrect API assumptions, over-thinking |
> > > | **Operating System** | XW_04 (File System) | 0 | 80 | -80 | Analysis paralysis, no implementation |
> > > | **Signal Processing** | WJ_01 (Image Denoising) | 69.37 | 73.00 | -3.63 | Over-engineering |
> > >
> > > Specifically, our observations suggest that CoT often induces **over‑engineering** or **over-thinking**, where the model focuses excessively on theoretical reasoning instead of producing practical, efficient code.
> > >
> > > For example, in task WJ_01, the CoT‑generated solution introduced an unnecessary bilateral filtering step, resulting in worse performance:
> > >
> > > ```
> > > # CoT approach – THREE steps (over‑engineered)
> > > median_filtered = cv2.medianBlur(img, ksize=3)
> > > bilateral_filtered = cv2.bilateralFilter(
> > >     median_filtered,
> > >     d=9, sigmaColor=75, sigmaSpace=75
> > > )
> > > filtered_img = cv2.fastNlMeansDenoisingColored(
> > >     bilateral_filtered,
> > >     None,
> > >     h=10,
> > >     templateWindowSize=7,
> > >     searchWindowSize=21
> > > )
> > > ```
> > >
> > > By contrast, the solution without CoT chose a simpler two‑step process that achieved better results:
> > >
> > > ```
> > > # Without CoT approach – TWO steps (optimal)
> > > denoised_img = cv2.medianBlur(noisy_img, 3)
> > > denoised_img = cv2.fastNlMeansDenoisingColored(
> > >     denoised_img,
> > >     None,
> > >     10, 10, 7, 21
> > > )
> > > ```
> > >
> > > Overall, for GPT‑4o, we observed that CoT encourages deeper theoretical reasoning in general. But this **does not always translate into correct or efficient implementations**. Some recent studies also have similar observations such as [zhu2025uncertainty], [wu2025more]. We thank the reviewer again for raising this point and we will incorporate this analysis into the revised paper.
> > >
> > > ---
> > >
> > > ### References
> > >
> > > [zhu2025uncertainty] Zhu, Y., Li, G., Jiang, X., Li, J., Mei, H., Jin, Z. and Dong, Y., 2025. Uncertainty-guided chain-of-thought for code generation with llms. arXiv preprint arXiv:2503.15341.
> > >
> > > [wu2025more] Wu, Y., Wang, Y., Ye, Z., Du, T., Jegelka, S. and Wang, Y., 2025. When more is less: Understanding chain-of-thought length in llms. arXiv preprint arXiv:2502.07266.

---

> > > > ### Comment · Reviewer_verF · 2025-08-06
> > > >
> > > > Many thanks for the extra look and the example. Very interesting. If space permits, I would encourage you to include at least a short remark, if not the table above, in the final version. On a second thought, the explanation with over-thinking for coding tasks sounds really convincing.

---

> > > > > ### Author Response · Authors · 2025-08-08
> > > > > **Thank you for the feedback!**
> > > > >
> > > > > Thanks so much for this suggestion. We will include a remark on this in the final version of our paper, and we sincerely appreciate your time and effort in reviewing our work!

---

### Official Review · Reviewer_2Gmh · 2025-07-03

**Ethics Flags:** Safety and security
**Rating:** 5
**Confidence:** 3

**Summary:**

The paper introduces EngDesign, a benchmark designed to evaluate Large Language Models (LLMs) on engineering design tasks across 9 domains. Unlike traditional QA benchmarks that focus on factual recall, EngDesign evaluates LLMs' ability to synthesize domain knowledge, reason under constraints, and generate functional designs. The benchmark comprises 101 tasks, evaluated through domain-specific simulations (SPICE, MATLAB, etc.) rather than simple answer matching. Results show that even SOTA models achieve barely 35% pass rate, highlighting the significant challenge of engineering design for current LLMs.

**Additional Feedback:**

1. I found that there is mroe information about dataset colelction than the actual dataset itself. Cosider for example: Sec. 2.3 is entirely unnecessary. This space could be used to expand related wok.

2. Related work discussion is very small and I would've preferred to see how this benchmark is different from top benchmarks in each subfield?

3. There is no comparison with humans. Also, were the LLMs finetuned for the specific domains? If not, then is it a fair expectation to have a single LLM that solves all these engineering tasks successfully?

4. How do you ensure that the 101 tasks truly represent the breadth of engineering design rather than just computational problems with engineering context?

5. With only 3 trials per model, how confident are you in the statistical significance of the reported differences?

6. Minor corrections:
6.1: Line 241: iterations.3. (extra full stop)

**Dataset Code Accessibility:**

Yes

**Dataset Code Comments:**

I was able to access the dataset and see the files. I found the documentatiom for code a bit hasty and feel that more of it could be added for most functions in the code,

**Ethical Comments:**

If the systems are built based on testing using this benchmark, then would the author ensure that this does not lead to a safety critical settings?

**Ethical Considerations:**

Yes, there are ethics concerns that require attention by the authors

**Final Justification:**

I believe the paper makes an important contribution in terms of benchmarks for engineering tasks. I had a few confusions about the work which were resolved during the author response and interaction.

**Limitations Weaknesses:**

1. Insufficient Task Detail in the paper: Many tasks lack sufficient detail about requirements, constraints, and expected outputs, and Figure 2 examples are too brief: The sample tasks shown don't provide enough context to understand what's actually being evaluated

2. There is no comparison with human engineer performance or existing engineering benchmarks. How dow e know that LLMs perform better than humans on these tasks?

3.  Figure 4 lacks explanation: No clear definition of what "reasoning robustness" means in practice is provided.

4.Some domains have only 5 tasks while others have 18, potentially skewing results.

**Strengths Contributions:**

1. The shift from static answer checking to dynamic simulation-based evaluation is a novel and more realistic for engineering assessment.

2. Spanning 9 engineering domains provides broad coverage of engineering disciplines.

3. Tasks represent actual engineering problems with constraints, trade-offs, and performance requirements rather than textbook exercises.

---

> ### Author Rebuttal · Authors · 2025-07-30
>
> We appreciate the reviewer’s recognition that our benchmark **reflect real engineering problems with constraints, trade‑offs, and performance requirements** rather than textbook exercises. Next, we address the reviewer’s concerns.
>
> ---
>
> ### **Insufficient Task Detail in the Paper**
>
> Thanks for this comment. As noticed by the reviewer, we have provided a **Dataset URL** which is a Hugging Face repository. Complete task descriptions with all requirements, constraints, and expected outputs are publicly available at that Dataset URL. Originally, Figure 2 only serves as a high-level illustration of task formats, not complete descriptions. **To address the reviewer's comment, we will copy complete task descriptions from our Dataset URL for the 4 examples in Figure 2 of our paper to the appendix of our revised version to give readers a clearer understanding of how tasks are structured and evaluated**. This will balance paper space constraints with transparency.
>
> ---
>
> ### **Human Engineer Baselines**
>
> Thanks for this comment. Notice that the tasks in EngDesign require specialized expertise across 9 distinct engineering domains. **Although each task is solvable by the human engineer in that specific domain with sufficient time and required tools, no single human engineer would be expected to solve problems across all 9 domains.** This is similar to HLE [Phan et al., 2025], which does not involve human baselines due to the fact that no single human being is expected to solve the problems across all the areas in HLE. Hence it is not meaningful to evaluate single human engineer's performance on EngDesign. We also note that the purpose of our paper is not to conclude that LLMs perform better than humans on EngDesign. Similarly, the main purpose of EngDesign is to serve as a diagnostic tool and a unified testbed in: i) Identifying LLM failure modes for practical engineering design, ii) tracking improvements as LLMs advance, iii) providing simulation-based evaluation for the LLM community.
>
> ---
>
> ### **Definition of Reasoning Robustness**
>
> We thank the reviewer for requesting clarification of the **Reasoning Robustness** metric. A high-level description appears in **Section 3 (Evaluation Metrics)**, with the full derivation in **Appendix E.3**. For completeness, we restate the formal definition here:
>
> Let $T$ denote the set of all evaluated tasks. For a given model, each task $t \in T$ is evaluated over $N$ independent trials. Define:
>
> * $N_{\text{all}}$: the number of tasks where **all** $N$ trials passed,
> * $N_{\text{any}}$: the number of tasks where **at least one** trial passed.
>
> The **Reasoning Robustness** $R \in [0, 1]$ is then $R = \frac{N_{\text{all}}}{N_{\text{any}}}$.
> This metric quantifies a model’s **consistency in reasoning**:
>
> * $R = 1$ indicates **perfect robustness**, meaning the model consistently succeeds whenever it can solve a task.
> * Lower values reveal **higher variance** or instability in reasoning across repeated attempts.
>
> ---
>
> ### **Uneven Task Distribution**
>
> This concern was also raised by Reviewer yeM4. Please see our comprehensive response under "**Uneven Task Distribution**" in our yeM4 response, which addresses domain complexity differences, evaluation weighting methodology, and benchmark precedents.
>
> ---
>
> ### **Potentially Skewing Results**
>
> Our results are not skewed since we report both:
>    * **Overall metrics** (average pass rate and score), and
>    * **Domain-wise metrics** (per-domain pass rates and scores).
>
> Our main metric uses **sample-level aggregation** with weights inversely proportional to the number of samples in each domain. For domain $d$ with $S_d$ samples, each sample weight is $w_i = \frac{1}{S_d}$. This ensures that no single domain dominates the overall score and allows users to identify strengths and weaknesses by domain. We also emphasize that domain-wise metrics are equally important as the overall metrics.
>
> ---
>
> ### **Necessary of Sec. 2.3 and the Breadth of EngDesign**
>
> We think that **Section 2.3 is critical** because it documents our **data collection methodology**, ensuring **transparency and credibility**. Section 2.3 outlines our **multi-stage expert review process**, which guarantees that tasks are **not merely computational problems with an engineering context**, but **realistic engineering design tasks** with genuine constraints and trade-offs, ensuring EngDesign truly represent the breadth of engineering design. Faculty experts and their students with expertise spanning all 9 domains are involved to contribute diverse expertise to ensure that EngDesign represents the breadth of engineering design.
>
> ---
>
> ### **Expanded Related Work**
>
>
> After expanding our literature review, we realize that most engineering benchmarks in subdomains do not focus on practical design. We will add the table below and discuss how EngDesign differs from sub-domain engineering benchmarks: EngDesign requires **end‑to‑end design and simulation‑based validation**, whereas prior work focuses on narrower subtasks such as code snippets, spec-writing, analysis, or Q\&A. For example, OSVBench evaluates formal specification and verification of simplified OS kernels, but it does not involve designing new operating systems. In contrast, the OS part of EngDesign focuses on designing subsystems.
>
> | Domain   | Sub-domain Benchmarks | Key Difference vs. EngDesign |
> | -------- | --------------------------------- | ---------------------------------------------------------------------- |
> | **OS**   | OSVBench | Spec/Veri (no design tasks)→ **design & simulate new subsystems** |
> | **Arch** | FIXME, ChatCPU | Verification / patching → **design & optimize new micro‑architecture** |
> | **Ctrl** | ControlBench, ControlEval  | Static Q\&A /simple PID tasks→ **diverse types of control design & validate closed‑loop**|
> | **Mech** | Mechanical‑agent case studies  | One‑off demos → **multiple mechanism designs with physics sim** |
> | **Stru** | SAWP, beam‑sets  | Analyze given beams → **size new structures under trade-offs**  |
> | **DHD**  | VerilogEval, LAD  | Snippet/verify RTL → **full‑module design with resource trade‑offs** |
> | **AICD** | AnalogCoder, ChipExpert, PICBench | Block synthesis → **topology design + sizing (more complex)** |
> | **Robo** | VL‑Manip, SafeAgentBench    | High-level Planning → **low level design and control sim** |
> | **SigP** | SensorBench, ASR‑Diag     | Coding / inference → **diverse topics on design for signal processing**  |
>
> ---
>
> ### **Do 101 tasks represent the breadth of engineering design rather than just computational problems with engineering context?**.
>
> At the heart of EngDesign is a clear distinction between genuine design problems and simple computations. Our task construction guidelines require that every task involve synthesis rather than analysis. Contributors must propose problems that call for multiple interdependent design decisions, fundamental design trade-offs, and demonstration of working solutions in simulation. An automated pre-check and two rounds of human review filter out "trivial" or misaligned submissions, with faculty experts in each discipline verifying that surviving tasks mirror real engineering practice. The contributors for EngDesign have engineering expertise spanning all 9 domains, and our authors include faculty experts with extensive prior industry experiences in various domains such as control, analog IC design, computer architecture, etc.
>
> ---
>
> ### **Statistical Significance of Our Results**
>
> To address this, now we add new results conducting **five independent trials** for each OpenAI model, and the results remain consistent with those reported in the main paper. Furthermore, we computed **95% confidence intervals** using sample variance to confirm statistical significance.
>
> | Model        | Avg. Pass Rate (3 trials) | Avg. Score (3 trials) | Avg. Pass Rate (5 trials) | Avg. Score (5 trials) |
> | ------------ | ------------------------- | --------------------- | ------------------------- | --------------------- |
> | gpt‑4o       | 14.74  $\pm$ 0.16        | 36.03   $\pm$ 0.17            | 15.30 $\pm$ 0.85                  | 40.75 $\pm$ 0.55                |
> | o1           | 36.54  $\pm$ 0.44          | 51.64   $\pm$ 0.26            | 35.52  $\pm$ 0.40                   | 52.82   $\pm$ 0.33              |
> | o3           | 35.90   $\pm$ 0.34         | 47.48  $\pm$ 0.38             | 34.43 $\pm$ 0.95                    | 50.14    $\pm$ 0.63             |
> | o4‑mini      | 36.54  $\pm$ 0.10       | 52.08   $\pm$ 0.26            | 39.89  $\pm$ 0.55                   | 59.36 $\pm$ 0.26                |
> | o3‑high      | 36.77   $\pm$ 0.23     | 49.61  $\pm$ 0.38              | 37.16    $\pm$ 0.40                 | 53.47   $\pm$ 0.09              |
> | o4‑mini‑high | 40.52    $\pm$ 0.10  | 54.02    $\pm$ 0.21          | 45.90  $\pm$ 0.26                   | 61.38   $\pm$ 0.21              |
>
> These results confirm that our findings are **statistically robust**, and our conclusions remain well supported.
>
> ---
>
> ### **Ethical Considerations: Safety and Security**
>
> EngDesign is specifically designed as a **benchmark for the engineering design stage**, not for deployment in safety-critical settings. The tasks focus on **design proposals and simulations**, not on direct control or execution in real-world environments. Any AI system evaluated on EngDesign would still require **extensive downstream validation, rigorous testing, and certification** before its outputs could be used in practice. We will reinforce these points in the **Broader Impact** section of the paper to ensure that readers understand the **scope and limitations** of EngDesign and its safe use in research settings.
>
> ---
>
> **References**
>
> * \[Phan et al., 2025] Humanity’s Last Exam. *arXiv preprint arXiv:2501.14249*.
>
> ---
>
> We hope our response addressed the reviewer’s concerns, and we would be sincerely grateful if the reviewer could reevaluate our work in light of our new results and responses.

---

> > ### Comment · Reviewer_2Gmh · 2025-08-04
> >
> > Thanks a lot for your response. I would like to clarify the following regarding my comment on adding human baselines:
> > 1. I did not mean performing user studies with the same human on each domain. I meant conducting user studies with experts in each domain, so an IC engineer can focus solely on IC design tasks.
> > 2. I also did not mean to show that LLMs are better or worse compared to humans on these tasks. I just wanted to evaluate how good humans are at these tasks as a baseline human performance.
> >
> > This is also important because you mentioned in your introduction that "our benchmark requires models to function as practicing engineers,..." However, it lacks a baseline for evaluating the current performance of human practicing engineers on these tasks that are part of the benchmark.
> >
> > Also, Phan et al.'s lack of human baselines cannot be used as a justification in this context, as that is a non-peer-reviewed paper available only on arXiv.
> >
> > I agree with your other responses. Thanks a lot for such a detailed response.

---

> > > ### Author Response · Authors · 2025-08-04
> > > **Response to Reviewer's Comment**
> > >
> > > We appreciate the reviewer’s clarification. We will address the reviewer's comments next.
> > >
> > > ---
> > >
> > > ### **Human engineer baselines:   how good humans are at these tasks as a baseline human performance.**
> > >
> > > We want to clarify that human solvability is actually already established for EngDesign. Firstly, all 101 tasks in EngDesign come with reference solutions crafted and verified by humans (there are typically infinitely many design solutions for each task and the reference solution is one of them; see our dataset). Of course, the reference solutions for different tasks are not from the same human, but this already established the human solvability for EngDesign. Secondly, faculty experts on our team have expertise spanning all nine domains, and can be viewed as proxy for expert performance. Each task has been confirmed solvable by at least one faculty member (for example, one of the faculty experts on our team has extensive consulting experience at major IC companies and can consistently solve all the analog IC design problems in EngDesign). This further established the human solvability for EngDesign.
> > >
> > > ---
> > >
> > > ### **Phan et al.'s lack of human baselines cannot be used as a justification in this context, as that is a non-peer-reviewed paper available only on arXiv.**
> > >
> > > While *Humanity’s Last Exam (HLE)* has not undergone formal peer review, it is widely discussed and has influenced how benchmarks are designed and evaluated. Regardless, our argument now does not hinge on HLE, and EngDesign stands on its own with reference solutions from humans and the established human solvability.

---

> > > > ### Comment · Reviewer_2Gmh · 2025-08-04
> > > >
> > > > Thanks a lot for the clarification. I suggest you incorporate the clarifications in the final version of the paper if accepted. Especially, adding a clarification for the uneven task distribution, explaining the reasoning you provided in the response to the other reviewer, and an extended related work.
> > > >
> > > > Based on the interaction, I will increase my score. Thanks again for the detailed responses.

---

> > > > > ### Author Response · Authors · 2025-08-04
> > > > > **Thank You for the Support**
> > > > >
> > > > > We sincerely thank the reviewer for their support of EngDesign. We are grateful for the fruitful discussion and constructive feedback, which have greatly strengthened our paper. We will ensure that all the points mentioned by the reviewer are thoroughly addressed and incorporated in the revised version.

---

### Official Review · Reviewer_yeM4 · 2025-07-04

**Rating:** 4
**Confidence:** 3

**Summary:**

This submission introduces ENGDESIGN, a novel benchmark designed to evaluate the engineering design capabilities of Large Language Models (LLMs). The benchmark moves beyond traditional question-answering formats by presenting complex, open-ended design tasks across nine distinct engineering domains: Operating System Design, Computer Architecture, Control Systems, Mechanical Systems, Structural Design, Digital Hardware, Analog IC Design, Robotics, and Signal Processing. A key innovation is the use of a simulation-based evaluation pipeline, where LLM-generated designs are rigorously tested for functional correctness using domain-specific tools like SPICE, MATLAB, and finite element analysis. The authors conduct a comprehensive evaluation of over 10 state-of-the-art LLMs, analyzing their performance, robustness, and failure modes. The work also explores an iterative design protocol, allowing models to refine their solutions based on feedback, mimicking a real-world engineering workflow.

**Dataset Code Accessibility:**

Yes

**Ethical Considerations:**

No, there are no or only very minor ethics concerns

**Final Justification:**

The authors have addressed almost all of my previous concerns, clarifying key methodological choices and strengthening the interpretation of the results. I have adjusted my score (4).

**Limitations Weaknesses:**

1. Heavy Reliance on Proprietary Software: A major limitation to the benchmark's accessibility is that 48 of the 101 tasks require expensive, licensed commercial software like MATLAB or Cadence for evaluation. This creates a significant financial barrier for many researchers, hindering the full reproducibility of the paper's results and limiting the benchmark's widespread adoption. While the authors provide an open-source subset (ENGDESIGN-OPEN), nearly half of the complete benchmark remains behind a paywall.

2. Uneven Task Distribution: While the benchmark covers nine engineering domains, the number of tasks per domain is significantly imbalanced. For example, Control Design and Signal Processing each contain 18 tasks, whereas Computer Architecture and Analog IC Design have only five each. This disparity, which the authors acknowledge reflects contributor expertise, could compromise the fairness of cross-domain comparisons and the overall assessment of a model's general engineering skills.

3. Superficial Analysis of Some Key Claims: The paper asserts that "reasoning models" consistently outperform chat models , and notes that they consume substantially more tokens to do so. However, the analysis does not sufficiently investigate why this is the case, leaving the connection between longer outputs and higher-quality reasoning underexplored. Furthermore, the detailed failure analysis was conducted on only a single model (o4-mini), missing an opportunity to provide a comparative analysis of the types of errors made by different classes of models.

**Strengths Contributions:**

1. The First Benchmark for Practical Engineering Design: The paper's most crucial and innovative contribution is establishing ENGDESIGN as the first-ever benchmark specifically targeting the practical design capabilities of LLMs. The authors correctly identify and address a critical void in the evaluation landscape; while other benchmarks test factual recall or textbook-style problems, ENGDESIGN is pioneering in its focus on the holistic engineering process: synthesizing domain knowledge, managing complex trade-offs, and generating functional solutions. This novelty makes the benchmark a groundbreaking and essential tool for measuring true, applicable engineering intelligence in AI.

2. Exceptional Thoroughness in Benchmark Curation: The completeness of the work that went into creating the benchmark is a significant strength. The paper details a large-scale, meticulous development process involving over 60 contributors, multiple stages of expert and faculty review, and automated pre-testing. Each of the 101 tasks is structured with impressive detail, including clear objectives, partial-credit rubrics, reference designs, and a robust, automated evaluation pipeline. This comprehensive and rigorous curation effort ensures the benchmark is not just novel in its concept, but also high-quality, reliable, and realistic in its execution.


3. Exhaustive and Insightful Empirical Validation: The paper is a complete research contribution, as the authors not only built this novel benchmark but also demonstrated its value through an exhaustive empirical study. They deployed ENGDESIGN to test a wide range of cutting-edge LLMs, providing a rich, multi-faceted analysis of their capabilities. The work goes beyond simple pass/fail scores to introduce insightful metrics like "Reasoning Robustness" and explore advanced concepts like iterative design, offering a deep and nuanced perspective on current AI limitations. This demonstrates the benchmark’s immediate utility and provides a solid foundation for future research.

---

> ### Author Rebuttal · Authors · 2025-07-30
>
> We thank the reviewer for the thoughtful feedback and recognition of ENGDESIGN as a **groundbreaking and essential tool** that addresses a **critical void in the evaluation landscape**. Below, we address the reviewer’s concerns point by point.
>
> ---
>
> ### **Reliance on Proprietary Software**
>
> We appreciate the reviewer’s concern and have taken concrete steps to improve accessibility. We realize that for several MATLAB-dependent tasks that do not require advanced toolboxes, our MATLAB.m files can be directly run on GNU Octave (which is open-source) and produce equivalent evaluation results. This aligns with the well-known fact that GNU Octave is the open-source alternative with high compatibility for MATLAB when advanced toolboxes are not involved. Now only **34 out of 101** tasks (instead of 48 out of 101) still require commercial software such as MATLAB toolboxes, Cadence, or Vitis HLS, and the resultant domain-wise breakdown is as follows:
>
> | Domain                       | Total | OS | Arch | Ctrl | Mech | Stru | DHD | AICD | Robo | SigP |
> | ---------------------------- | ----- | -- | ---- | ---- | ---- | ---- | --- | ---- | ---- | ---- |
> | **# of open-source tasks**   | 67    | 8  | 5    | 7    | 6    | 7    | 13  | 0    | 10   | 11   |
> | **# of closed-source tasks** | 34    | 0  | 0    | 11   | 1    | 6    | 4   | 5    | 0    | 7    |
>
> Consequently, the majority of tasks (**67 out of 101**) are now fully open-source, significantly improving accessibility.  The remaining commercial tools (MATLAB toolboxes, Cadence, Vitis HLS) reflect industry standard practice essential for authentic evaluation, particularly in specialized domains like robust control and Analog IC Design where open-source alternatives lack the maturity and accuracy required for reliable assessment. While we explored open-source alternatives (e.g., the Python Control Systems Library), our preliminary experiments showed that these packages were insufficiently mature for evaluating some of our tasks.
>
>
> We emphasize that including commercial scientific software reflects **the practical reality of modern engineering practice**, where such tools are widely recognized as industry standards. This choice ensures that EngDesign faithfully captures real-world workflows and evaluates AI systems under realistic conditions. **There is an inherent trade-off between authentic engineering practice and open-source accessibility**. In industry, practicing engineers routinely rely on commercial software (e.g., MATLAB toolboxes for robust control, SPICE for circuit design) because these tools represent the state-of-the-art in their respective domains. Several of our authors have extensive prior industry experience and can confirm this is standard practice. To provide a meaningful evaluation of AI systems intended for real-world engineering applications, our benchmark must capture this reality.
>
>
>
>
> ---
>
> ### **Uneven Task Distribution**
>
> Thanks for this comment. We emphasize that task counts correspond to the natural scope of problems in each field, and the small number of the tasks in certain domain (e.g., analog IC design) naturally reflects the effort required to make them realistic and gradable, not an intent to underrepresent the domain. Importantly, fewer analog IC design tasks $\neq$ underrepresentation or lower effort!  We elaborate on our points as follows:
>
> 1. **Quality and realism:**  Designing high‑fidelity simulation-based tasks is labor‑intensive and depends on domain expertise.
> For example, each analog‑IC task (TB_01–TB_05) encapsulates a multi‑dimensional design challenge, requires solving a complex sizing and topology problem under multiple constraints (e.g., gain, bandwidth, etc) and needs to be verified through detailed SPICE simulation sweeping across various operating points. Such tasks are much more parameter‑rich and labor‑intensive to curate than, for example, tuning a PID controller. The small number of analog IC tasks reflects the effort required to make them realistic and gradable, not an intent to underrepresent the domain.
>
> 2. **Domain‑appropriate complexity:**  Some disciplines naturally organize into discrete design challenges, e.g. individual filter designs and controller synthesis. Each problem in these domains is complex but can be evaluated independently. Other domains, like analog IC design and computer architecture, typically require integrated, system‑level approaches coupling multiple design challenges. Hence task counts correspond to the natural scope of problems rather than arbitrary balance.
>
> 3. **Precedent in benchmark literature:** Many widely used benchmarks (e.g., **GPQA** \[Rein et al., 2024] and **HLE** \[Phan et al., 2025]) are naturally imbalanced, yet remain valuable because each task is **carefully curated** and **domain-authentic**. For example, the Humanity’s Last Exam (HLE) benchmark spans a broad range of disciplines but does not have equal numbers of questions per area; instead, it focuses on domain‑authentic, high‑quality problems (there are many more math problems in HLE than in other areas).
>
> 4. **Mitigating bias with evaluation metrics:** To ensure fair cross-domain evaluation, we report both:
>    * **Overall metrics** (average pass rate and score), and
>    * **Domain-wise metrics** (per-domain pass rates and scores).
>
>     Our main metric uses **sample-level aggregation** with weights inversely proportional to the number of samples in each domain. For domain $d$ with $S_d$ samples, each sample weight is $w_i = \frac{1}{S_d}$. This ensures that no single domain dominates the overall score and allows users to identify strengths and weaknesses by domain. We also emphasize that domain-wise metrics are equally important as the overall metrics.
>
> 5. **A foundation for future work:** EngDesign is the **first** benchmark for simulation-based evaluation of general engineering design capabilities. Although current task counts vary by domain, we believe that this framework provides a **solid foundation for the community** to add new, high-quality engineering design tasks over time.
>
> We will add these clarifications, along with details about our weighting scheme, into **Section 3 (Evaluation Metrics)** and the discussion of results.
>
> ---
>
> ### **Superficial Analysis of Some Key Claims**
>
> We agree with the reviewer that a deeper understanding of why longer reasoning traces correlate with higher accuracy—and how error patterns vary across model classes—adds important context. To address this, we have added (i) a concrete case study contrasting a chat model with a reasoning model, and (ii) an expanded cross-model failure analysis.
>
> #### 1. **Case Study: Task XG\_05**
>
> **Goal:** Identify a first-order-plus-dead-time (FOPDT) model for a stirred-tank reactor from its step response, then design a feedback controller that meets specified gain/phase margins, overshoot, and settling time.
>
> |                           | GPT‑4o                                      | o3                                              |
> | ------------------------- | ------------------------------------------- | ----------------------------------------------- |
> | **Token Consumption**     | 669                                         | 7 763                                           |
> | **System Identification** | $\theta$ = 19.36, $\tau$ = 13 **(wrong)**               | $\theta$ = 16.5, $\tau$ = 19.5 **(correct)**                |
> | **Controller Design**     | Ad‑hoc PID, no performance check → **fail** | IMC‑tuned PID, validated performance → **pass** |
>
> **Observation:** GPT‑4o uses incorrect parameters and never revisits them, resulting in failure. In contrast, o3 derives the FOPDT equations symbolically, solves them exactly, and validates closed-loop margins before finalizing the controller. Although o3’s reasoning trace is not visible via API calls, the **quality of its end-to-end reasoning**—evident in correct parameter derivation and structured checks—demonstrates its superior performance compared to non-reasoning models.
>
> ---
>
> #### 2. **Expanded Fine-Grained Failure Analysis**
>
> We extended our fine-grained failure analysis to two additional frontier models (Gemini‑2.5‑Pro and Claude‑3.7‑Sonnet). The results are summarized below:
>
> | Model             | Domain-knowledge error | Constraint-violation error | Prior-knowledge over-reliance | Hallucination | Computation error | Other |
> | ----------------- | ---------------------- | -------------------------- | ----------------------------- | ------------- | ----------------- | ----- |
> | o4‑mini           | 33.3 %                 | 25.2 %                     | 18.8 %                        | 12.6 %        | 9.0 %             | 0.9 % |
> | Gemini‑2.5‑Pro    | 31.9 %                 | 31.9 %                     | 15.9 %                        | 12.4 %        | 6.2 %             | 1.7 % |
> | Claude‑3.7‑Sonnet | 30.7 %                 | 36.0 %                     | 10.5 %                        | 13.2 %        | 7.0 %             | 2.6 % |
>
> ##### **Key Takeaways**
>
> 1. **Dominant failure modes:** Insufficient domain knowledge and constraint violations account for **55–67%** of failures.
> 2. **Second-tier issues:** Over-reliance on prior knowledge and hallucinations contribute an additional **25–30%**, showing that misapplication of known facts remains a major hurdle.
> 3. **Low arithmetic fragility:** Pure computation errors are rare (≤9%), indicating that numerical routines are **not the primary bottleneck** for current frontier models.
>
> These patterns are consistent with our initial observations.
>
> ---
>
> **References**
>
> * \[Rein et al., 2024] GPQA: A Graduate-Level Google-Proof Q\&A Benchmark. *First Conference on Language Modeling*.
> * \[Phan et al., 2025] Humanity’s Last Exam. *arXiv preprint arXiv:2501.14249*.
>
> ---
> We hope these clarifications address the reviewer’s concerns, and we would be sincerely grateful if the reviewer could reevaluate our work in light of our new results and detailed responses.

---

### Decision · Program_Chairs · 2025-09-18

**Decision:**

Accept (poster)

**Comment:**

This paper introduces an approach to evaluating large language model performance on realistic engineering tasks.

Strengths:

> I agree that “the shift from static answer checking to dynamic simulation-based evaluation is a novel and more realistic for engineering assessment” (As highlighted by Reviewer 2Gmh) -   in fact, I consider the “rigorous testing through executable, domain-specific simulations” to be the main contribution of this paper - engineering problems are implemented and tested in compiled settings, which is much more robust and realistic an approach than simple typical QA formats.

Limitations:

> I have some concern about an over-reliance on LLMs. The design scenarios are LLM generated, and do not actually reflect real prompts used by engineers during a realistic design workflow. None of the questions are real and there are limited task details so there is opportunity for at least a subset of the tasks to be contrived (as noted by verF, 2Gmh). Proper and complete task documentation would be necessary for full confidence in this benchmark - though the authors do mention in the rebuttal a willingness to include more task design and description details in the Hugging Face dataset documentation.

> Motivation & Domain Coverage: The motivation here is a bit unclear? What is the value of having a single unspecialized model perform well across this particular set of engineering tasks? Beyond topical distinctions, are the characteristics of the tasks that the model is performing that are especially notable and/or challenging and/or representative? In general, there is a high risk involved in any benchmark that claims "general" competence or coverage [1]. That being said - the role of the consulted experts in deriving the taxonomy of tasks is also a bit unclear. In particular, the risk of selection bias (as noted by Reviewer 2Gmh), could meaningfully skew how overall benchmark performance should be interpreted.

> There were some raised concerns around related work completeness and comparison to prior state of the art across specific task categories and implementations.

> Several reviewers raised the issue of openness, as the benchmark needs to be compiled by various proprietary tools in order to be validated. Being explicit about this limitation in the main paper is a must; also being upfront about how this changes the scale of the number of available tasks and representation across task categories is important.

> “Reasoning robustness” as a concept was unclear to various reviewers, though I feel the authors adequately provided a clarification of the concept in their rebuttal. This clarification should make it into the final version of the paper.

> Lack of human baseline (2Gmh) is concerning - I also do not agree with the author response; why not recruit an ensemble of human engineers that can execute on the individual tasks? There’s not expectation that a single human provide the universal baseline across every task in the suite. GPQA, for example, takes this decentralized human validation approach [2]. It could also be worthwhile to consider not just comparisons to human performance but also human baselines across other important factors, such as robustness [3].

Despite these weaknesses, given the strength of the author rebuttal and promises to address various of the raised issues above, this paper sits just above the acceptance threshold for this conference. I truly do hope, however, that the authors strengthen their work by integrating the feedback, clarifications and additional documentation as recommended explicitly by the reviewers.

[1] Raji, Inioluwa Deborah, et al. "AI and the everything in the whole wide world benchmark." (2021).
[2] Rein, David, et al. "Gpqa: A graduate-level google-proof q&a benchmark." First Conference on Language Modeling. 2024.
[3] Firestone, Chaz. "Performance vs. competence in human–machine comparisons." Proceedings of the National Academy of Sciences 117.43 (2020): 26562-26571.

===== FINAL UPDATE FROM DB Track PCs ====

The final decision for this paper has been taken by the program chairs after consultation with the SACs. All Senior Area Chairs have ranked papers according to the feedback from the AC during the review process. We decided to leave the original meta-review to reflect the opinion of the AC in light of the initial discussions with reviewers and SAC.